# A global historical twice-daily (daytime and nighttime) land surface temperature dataset produced by AVHRR observations from 1981 to 2021

Jia-Hao Li[1,2], Zhao-Liang Li[3,1], Xiangyang Liu[3], and Si-Bo Duan[3]

[1]State Key Laboratory of Resources and Environment Information System, Institute of Geographic Sciences and Natural Resources Research, Chinese Academy of Sciences, Beijing 100101, China
[2]University of Chinese Academy of Sciences, Beijing 100049, China
[3]Key Laboratory of Agricultural Remote Sensing, Ministry of Agriculture and Rural Affairs/Institute of Agricultural Resources and Regional Planning, Chinese Academy of Agricultural Sciences, Beijing 100081, China

*Correspondence to*: Zhao-Liang Li (lizhaoliang@caas.cn)

**Abstract.** Land surface temperature (LST) is a key variable for monitoring and evaluating global long-term climate change. However, existing satellite-based twice-daily LST products only date back to 2000, which makes it difficult to obtain robust long-term temperature variations. In this study, we developed the first global historical twice-daily LST dataset (GT-LST), with a spatial resolution of 0.05°, using Advanced Very High Resolution Radiometer (AVHRR) Level-1b Global Area

Coverage (GAC) data from 1981 to 2021. The GT-LST product was generated using four main processes: (1) GAC data reading, calibration, and pre-processing using open-source Python libraries; (2) cloud detection using the AVHRR-Phase I algorithm; (3) land surface emissivity estimation using an improved method considering annual land cover changes; and (4) LST retrieval based on a nonlinear generalized split-window algorithm. Validation with in situ measurements from Surface Radiation Budget (SURFRAD) sites and Baseline Surface Radiation Network sites showed that the overall root-mean-square

errors of GT-LST varied from 1.6 K to 4.0 K, and nighttime LSTs were typically better than daytime LSTs. Inter-comparison with the Moderate Resolution Imaging Spectroradiometer LST products (MYD11A1 and MYD21A1) revealed that the overall root-mean-square-difference (RMSD) was approximately 3.0 K. Compared with MYD11A1 LST, GT-LST was overestimated, and relatively large RMSDs were obtained during the daytime, spring and summer. Whereas the significantly smaller positive bias was obtained between GT-LST and MYD21A1 LST. Furthermore, we compared our newly generated dataset with a

global AVHRR daytime LST product at the selected measurements of SURFRAD sites (i.e., measurements of these two satellite datasets were valid), which revealed similar accuracies for the two datasets. However, GT-LST can additionally provide nighttime LST, which can be combined with daytime observations estimating relatively accurate monthly mean LST, with RMSE of 2.7 K. Finally, we compared GT-LST with a regional twice-daily AVHRR LST product over continental Africa in different seasons, with RMSDs ranging from 2.1 to 4.3 K. Considering these advantages, the proposed dataset provides a

better data source for a range of research applications. GT-LST is freely available at https://doi.org/10.5281/zenodo.7113080 (1981-2000) (Li et al., 2022a), https://doi.org/10.5281/zenodo.7134158 (2001-2005) (Li et al., 2022b), and https://doi.org/10.5281/zenodo.7813607 (2006-2021) (Li et al., 2023).

# 1 Introduction

Land surface temperature (LST) is one of the key physical variables of land surface processes (Li et al., 2013). As an indicator
of the regional and global surface energy and water balance (Duan et al., 2018; Liu et al., 2019; Ma et al., 2020; Zhang et al.,
2022), LST has been used to detect climate change (Bright et al., 2017; Hansen et al., 2010; Jin and Dickinson, 2002; Li et al.,
2015), estimate surface soil moisture (Bai et al., 2019; Song et al., 2022; Zhao et al., 2021), monitor vegetation (Duveiller et
al., 2018; Sim et al., 2008; Weng et al., 2004), assess drought (Sánchez et al., 2018; Zhang et al., 2017), and study the urban
thermal environment (Phan and Kappas, 2018; Si et al., 2022). Many of these applications require long-term observations
made at regular temporal revisit intervals over large spatial scales (Hong et al., 2022). Compared to traditional ground
observations, which are sparse, unevenly distributed, and able to obtain LST only at a specific point, satellite observations
offer a valid opportunity to obtain LST data with a large and continuous spatial coverage.

LST cannot be measured directly by satellite but can be estimated from satellite-based thermal infrared (TIR) data (Li et al.,
2013). To date, several methods for LST retrieval have been developed in accordance with TIR data, such as mono-window
algorithm (Qin et al., 2001), split-window algorithms (Becker and Li, 1990; Wan and Dozier, 1996), temperature–emissivity
separation algorithm (Gillespie et al., 1998), and physical day and night algorithm (Wan and Li, 1997). Currently, a number
of publicly available LST products exist that are based on various TIR instruments on board satellite platforms and derived
from different LST retrieval algorithms (Li et al., 2023). These LST products can be divided into three approximate categories
according to their spatial–temporal resolutions and time periods: (1) global LST products with low temporal resolution but
high spatial resolution, such as Landsat LST product (16-day and 30-m) and the Advanced Spaceborne Thermal Emission and
Reflection Radiometer (ASTER) LST product (16-day and 90-m) (Gillespie et al., 1998; Malakar et al., 2018); (2) global LST
products with medium spatial resolution (1-km) and medium temporal resolution (twice daily), such as the Advanced Very
High Resolution Radiometer (AVHRR) LST product, the (Advanced) Along-Track Scanning Radiometer LST product, the
Moderate Resolution Imaging Spectroradiometer (MODIS) LST product, and the Visible and Infrared Imagery Radiometer
Suite LST product (Hulley and Hook, 2018a; Hulley and Hook, 2018b; Prate, 2002; Trigo et al., 2011; Wan, 2006;); and (3)
regional LST products with relatively low spatial resolution but high temporal resolution, such as the Advanced Baseline
Imager LST product (America, 1-h and 2-km), the Spinning Enhanced Visible and InfraRed Imager LST product (Africa, 15-
min and 3-km), the Advanced Geosynchronous Radiation Imager LST product (China, 1-h and 4-km) and the Advanced
Himawari Imagers LST product (Japan, 1-h and 2-km) (Trigo et at., 2008; Yamamoto et al., 2018; Yang et al., 2017; Yu et al.,
2008). In summary, the number of regional and global LST products derived from TIR data has increased, but global daily
satellite-derived LST products with medium and high spatial resolution only date back to the year 2000. However, many
application fields, including climate change, environmental monitoring, and meteorology, urgently require global LST
products with twice-daily observations that include more than 40 years of available data (IPCC, 2014; Liu et al., 2019; Ma et
al., 2020). Notably, AVHRR is the only sensor that has the advantages of frequent revisits (twice per day), relatively high
spatial resolution (4-km at the nadir), global coverage, and easy access prior to 2000.

Several LST products have been generated from AVHRR TIR measurements before 2000 (Table 1). These products can be broadly classified into two categories. The first includes regional products with relatively high spatial or temporal resolution. For example, the European Space Agency produced the World Land Surface Temperature Atlas dataset, which provides monthly LST data over Europe at 1-km and 0.5° spatial resolution from 1992 to 1993 (Kerr et al., 1998). Moreover, Pinheiro

et al. (2006) developed a regional daily, 8-km resolution, daytime and nighttime LST dataset over Africa for the NOAA-14 AVHRR from 1995 to 2000 (denoted as RT-LST). Khorchani et al. (2018) generated a long-term AVHRR LST dataset with a spatial resolution of 1 km for Peninsular Spain at annual and seasonal time scales for 1981–2015. Furthermore, a long-term study by the TIMELINE project of the Earth Observation Center at the German Aerospace Center provided a long time series of almost 40 years of daily AVHRR LST at 1-km spatial resolution over Europe and North Africa (Frey et al., 2012; Frey et

al., 2017; Holzwarth et al., 2021; Reiners et al., 2021). The second category includes global products with low temporal resolution. For example, Ouaidrari et al. (2002) generated a global monthly average LST dataset at 8-km spatial resolution for January and July 1989, based on the AVHRR Land Pathfinder II project framework. Moreover, Jin (2004) provided a monthly global 8-km, 0.5° and 5° resolution LST dataset based on the diurnal temperature cycle model, which spans a 17-year period (i.e., 1981–1998). A more recent study by Ma et al. (2020) generated a global historical daytime 0.05°×0.05° LST product

(denoted as GD-LST) by reprocessing the daytime AVHRR dataset (including reflectance data and brightness temperatures data) provided by the Land Long Term Data Record (LTDR) for 1981–2000. In summary, these efforts are limited by covering only certain regions (e.g., Europe or Africa) or their coarse temporal resolutions (e.g., daytime or monthly). To develop a long-term (> 40 years) satellite-derived LST product, it is necessary to generate a twice-daily AVHRR LST product that covers a period of more than 40 years or can be combined with the existing satellite-derived twice-daily LST product (e.g., MODIS)

after 2000. Moreover, global long-term meteorology and climatology-related applications also demand global and instantaneous AVHRR LST data with two observations each day, which can be used to estimate relatively accurate climate change indices such as the mean LST, extreme LST, and diurnal LST range.

In this study, we aim to fill this research gap by developing a standard global historical twice-daily (daytime and nighttime) LST product (GT-LST) at 0.05° spatial resolution. GT-LST is derived from original long time series AVHRR Level-1b Global

Area Coverage (GAC) data spanning a 41-year period (1981–2021). Section 2 introduces the data used in this study, including data for LST generation and validation. Section 3 describes the methodology for GT-LST generation and validation. Section 4 presents and discusses the results. Section 5 summarizes the main conclusions.

## 2 Datasets

### 2.1 AVHRR datasets

The GT-LST product is derived from AVHRR sensors installed aboard the NOAA-series of polar-orbiting environmental satellites (POES) (Cracknell, 1997). According to the operational time of different POES satellites, NOAA-7/9/11/14/16/18/19 were selected to generate a global long-term LST from 1981 to 2021 (Fig. 1). The orbital period was about 102 min, producing

14 orbits per day (Kidwell, 1991). The AVHRR sensor has six spectral bands with a spatial resolution of 1.1 km at the nadir and scan angles of approximately ±55° off the nadir (Table 2). Although the AVHRR sensors measure the same infrared bands,

their spectral responses are not completely identical. Fig. 2 shows the spectral response of the two infrared band of NOAA-7/9/11/14/16/18/19.

The commonly used AVHRR Level-1b GAC data are reduced-resolution data, which take the first one scan line out of every three, average four of each five consecutive samples along the scan line, and are processed onboard the satellite in real-time. Therefore, AVHRR Level-1b GAC data are generally treated as having a coarse resolution of 4 km at the nadir, and the pixel

size increases with the satellite zenith angle (VZA). Furthermore, as the VZA increases, the geolocation accuracy of the AVHRR GAC scene become lower, particularly when VZAs larger than 40° (Wu et al., 2020). However, the AVHRR Level-1b GAC dataset is the only dataset in which every place on Earth has been sampled at least twice per day (daytime and nighttime) since 1981(Kidwell, 1991). Thus, AVHRR Level-1b GAC data are available for generating global daytime and nighttime LST data from 1981 to 2021. AVHRR GAC data were archived in Level-1b format with 10-bit precision. Then the

data were assembled into discrete datasets using full orbits with quality control. Each file contains video data for the six channels, as well as time codes, quality indicators, Earth location, calibration information, and solar zenith angles (SZA). AVHRR Level-1b GAC data were obtained from the NOAA Comprehensive Large Array-Data Stewardship System (https://www.avl.class.noaa.gov/saa/products/search?datatype_family=AVHRR).

## 2.2 Datasets for generating simulations

To obtain the nonlinear generalized split-window (GSW) algorithm coefficients, it is necessary to establish a comprehensive simulation dataset. In this study, we used the latest version of Thermodynamic Initial Guess Retrieval 2000 dataset, which is a reliable atmospheric profile dataset, and the ASTER spectral library, which is a collection of the Jet Propulsion Laboratory spectral library, Johns Hopkins University spectral library, and United States Geological Survey spectral library.

The Thermodynamic Initial Guess Retrieval 2000 dataset (V1.2) contains 2,311 representative atmospheric situations that were

carefully selected from 8,000 global radiosonde reports (Chedin et al., 1985). Each situation consists of temperature, ozone concentrations, and water vapor values at a given pressure level from the surface to the top of the atmosphere. Finally, we obtained 946 globally representative and clear-sky atmospheric conditions by removing cloudy atmospheric conditions, i.e., removing the relative humidity at any pressure level exceeding 90% or two adjacent pressure levels exceeding 85%. The range of WVC and near-surface air temperature values is 0.06–6.5 g cm$^{-2}$ and 230–310 K under these atmospheric conditions.

The ASTER spectral library version 2.0 includes over 2,300 spectra of natural and man-made materials covering the wavelength range from 0.4 μm to 15.4 μm. In this study, we used 54 land surface emissivity spectra to represent different land surface types, including 41 soil types, four vegetation types, four water body types and five ice/snow types. The emissivity values of the AVHRR TIR channels were estimated by convolving the emissivity spectra with the relative spectral response functions of AVHRR bands 4 and 5.

## 2.3 Datasets for emissivity estimation

For nonlinear GSW, emissivity is an essential parameter in LST retrieval, and its accuracy directly affects LST accuracy. Four datasets were used for emissivity estimation, except for the Level-1b reflectance dataset of the GT-LST product: ASTER Global Emissivity Dataset (GED), Global Soil Regions map (GSRM), and global yearly land cover dynamics of the Global Land Surface Satellite (GLASS-GLC) and MODIS land cover product (MCD12C1).

The ASTER GED product, which provides the global mean land surface emissivity in five ASTER TIR spectral bands with a spatial resolution of 100 m and 1 km on 1°×1° grids, was generated by the National Aeronautics and Space Administration's Jet Propulsion Laboratory (Hulley et al., 2015). The emissivity of ASTER GED was developed from all clear-sky ASTER data acquired over 2000–2008 using temperature-emissivity separation algorithms and the water vapor scaling atmospheric correction algorithm. This product also provides the mean LST, mean normalized difference vegetation index (NDVI), global digital elevation model, land-water mask and other data. In this study, we used the ASTER GED mean emissivity and mean NDVI at 1-km spatial resolution.

The GSRM product provides the global distribution of 12 major soil types with a 2′ spatial resolution. It was generated by the United States Department of Agriculture using a reclassification of the FAO-UNESCO Soil Map of the World, combined with a soil climate map (https://www.nrcs.usda.gov/wps/portal/nrcs/detail/soils/use/?cid=nrcs142p2_054013).

The GLASS-GLC product provides the first record of the 1982–2015 global yearly land cover dynamics with a spatial resolution of 5 km (Liu et al., 2020). It forms part of the global land surface satellite products and is generated using the Google Earth Engine platform. This land cover product contains seven types of land cover: barren land, tundra, cropland, grassland, shrubland, forest, and snow/ice. The average overall accuracy of each land cover type from 1982 to 2015 according to 2,431 test sample units is 82.81 %. The GLASS-GLC product from 1982 to 2005 was used in this study. The MCD12C1 product provides global maps of land cover at a spatial resolution of 0.05° and an annual time step, starting from 2001 and continuing to the present (Sulla-Menashe and Friedl, 2018). In this study, the Collection-6 MCD12C1 product from 2006 to 2020 was employed. In the absence of available global land cover datasets for 1981 and 2021, the land cover data for 1982 and 2020 were used as a substitute in this study.

To match the GT-LST pixels, these global surface datasets were mosaicked and resampled to 0.05° spatial resolution in terms of their geographic longitude and latitude.

## 2.4 Atmospheric water vapor content dataset

The ancillary dataset used for LST retrieval was the Modern-Era Retrospective Analysis for Research and Applications Version 2 Reanalysis dataset, tavg1_2d_slv_Nx, which provides an hourly time-averaged WVC (the variable name is TQV in this dataset) at 0.5°×0.625° spatial resolution (https://disc.gsfc.nasa.gov/datasets?project=MERRA-2). The TQV dataset was corrected to match the spatial resolution and overpass time of AVHRR prior to LST retrieval.

## 2.5 Validation datasets

Validation of product accuracy is necessary before applying a new LST product. In this study, ground-based validation, satellite products inter-comparison, and comparison with existing AVHRR LST data were used to assess the accuracy of the retrieved product.

In situ measurements from the Surface Radiation Budget (SURFRAD) network and the Baseline Surface Radiation Network (BSRN) were used to validate GT-LST. The SURFRAD network was established in 1993 to support accurate, continuous, and long-term measurements of climate research in the United States (Augustine et al., 2000). In this study, we selected seven stations of the SURFRAD network representing various land cover types and providing in situ data between 1994 and 2005 (Table 3). SURFRAD sites provide quality-controlled measurements of solar/infrared upwelling/ downwelling radiation. Upwelling and downwelling TIR radiances are the primary measurements used to retrieve in situ LST. The instrumental error of the SURFRAD station gives rise to uncertainty in the retrieved LST value of less than 1 K (Guillevic et al., 2012). Therefore, the LST from SURFRAD has been widely used to evaluate ASTER, MODIS, and VIIRS LST products (Wang et al., 2008; Wang and Liang, 2009). The BSRN has 76 stations that detect important changes in the Earth's radiation field at the Earth's surface since 1992. These stations provide high-quality surface and upper-air meteorological observations, which are important in supporting the validation and confirmation of satellite. We selected four sites with measurements of upwelling and downwelling TIR radiances before 2000 (Table 3). In situ LST measurements were estimated using Stefan–Boltzmann's law as follows:

$$LST_s = \sqrt[4]{\frac{R\uparrow - (1 - \varepsilon_b)R\downarrow}{\sigma \varepsilon_b}} \tag{1}$$

where $LST_s$ is the in situ LST; $\sigma$ is the Stephan-Boltzmann constant; $R\uparrow$ and $R\downarrow$ are the upwelling and downwelling longwave radiation, respectively; and $\varepsilon_b$ is the broadband emissivity, which was derived from Duan et al. (2019).

The MODIS LST products (MYD11A1 and MYD21A1) were used to evaluate the accuracy of GT-LST. MYD11A1 LST is a daily level 3 LST product, which is a typical operational and standard LST product with a 1-km spatial resolution from 2002 to the present. MYD11A1 observations were obtained by the MODIS sensor onboard the Aqua satellites, which pass through the equator at approximately 13:30/1:30 local solar time. Every pixel has quality flags containing cloud contamination, emissivity, input data, and calibration. In this study, Collection-6.1 MYD11A1 of 2004 was selected for sensor-to-sensor comparison. MYD21A1 LST product, which uses the same observations with MYD11A1 but uses temperature–emissivity separation method to dynamically retrieve LST and emissivity, was also selected to make an intercomparison with GT-LST in this study. This inter-comparison was conducted on 4 months in 2004 (January, April, July, and October) which cover different seasons.

Globally and regionally historical AVHRR LST products, GD-LST and RT-LST, were used to compare to GT-LST. Especially, GD-LST is the only currently available global daytime AVHRR LST, with a spatial resolution of 0.05°×0.05° from 1981 to 2000. Compared to GT-LST, GD-LST is not derived from the original AVHRR Level-1b GAC datasets, but from LTDR datasets that reprocess daytime AVHRR data such as the reflectance, top-of-atmosphere brightness temperature of TIR bands,

and NDVI. RT-LST is a twice-daily LST product at 8-km resolution over continental Africa from 1995 to 2000, which is based
on GAC data. Auxiliary data of RT-LST only include cloud mask and observation time without satellite zenith angles (VZA).

## 3 Methodology

### 3.1 LST generation

This study developed an AVHRR LST processing system to produce a global historical twice-daily (daytime and nighttime)
LST dataset with a 0.05° spatial resolution from 1981 to 2021 (Fig. 3). The system includes four steps: (1) data reading,
calibration, and pre-processing; (2) cloud detection; (3) land surface emissivity estimation; and (4) LST retrieval. In the
following subsections, we describe each major component of the processing system.

#### 3.1.1 Data reading, calibration, and pre-processing

The first step in our framework includes reading, decoding, performing quality control, and calibrating packed 10-bit AVHRR
Level-1b GAC data (Fig. 3). In this study, we used an open-source and community-driven package, Pygac, to process the 41-
year AVHRR Level-1b GAC data record. Pygac is a Python package used for reading, calibrating, and navigating data from
the AVHRR instrument in GAC and Local Area Coverage (LAC) formats (Devasthale et al., 2017). Many studies have
processed AVHRR GAC/LAC data using this package (Karlsson et al., 2017; Pareeth et al., 2016; Wu et al., 2021). By
inputting the AVHRR Level-1b GAC data and two-line elements of a satellite into the Pygac program, we can obtain calibrated
quality control (QC) flags, sun-satellite position, reflectance and brightness temperature data. The complete details of the
package are provided at https://github.com/pytroll/pygac.
We then remapped and rebinned the data into the World Geodetic System 1984 projection with 0.05° grid cells. Owing to the
wider scan angles of NOAA satellites, panoramic bow-tie effects were apparent at the edges of the images (Pareeth et al.,
2016). Thus, we used the Pyresample package to resample the AVHRR Level-1b GAC data and correct for bow-tie effects.
Further details of the package are explained at https://github.com/pytroll/pyresample. In areas where multiple AVHRR
observations were available for a given grid cell, especially in polar latitudes, we selected and stored only one observation per
grid cell with the maximum brightness temperature from channel 4 (Pinheiro et al., 2006; Salelous et al., 2000). We assumed
that this observation had a lower possibility of including cloud. Then, we distinguished daytime and nighttime observations
using SZA to ensure compatibility with the cloud detection algorithm (Stowe et al., 1999). If the SZA of a pixel was less than
85°, the pixel and its observations were assigned to the daytime class; otherwise, they were assigned to the nighttime class.

#### 3.1.2 Cloud detection

Currently, no global daytime and nighttime cloud mask datasets are available for AVHRR Level-1b GAC data before 2000.
Therefore, to obtain global daytime and nighttime cloud-free pixels from 1981 to 2021, we adapted the Clouds from AVHRR-
Phase I (CLAVR-1) algorithm, which classifies each 2×2 AVHRR Level-1b GAC pixel array into clear, mixed, and cloudy

classifications (Stowe et al., 1999). The CLAVR-1 algorithm used three different tests to perform the classification: contrast,

spectral, and spatial signature threshold tests. This algorithm is a more generic approach that detects cloud/clear observations over both day and night, and land and ocean via the day-land algorithm, day-ocean algorithm, night-land algorithm, and night-ocean algorithm. Further details of the algorithm are provided by Stowe et al. (1999). In this study, we used the day-land and night-land algorithms of CLAVR-1 to identify clear and cloudy pixels and create a cloud mask dataset (Fig. 3).

### 3.1.3 Land surface emissivity estimation

To retrieve LST using nonlinear GSW, the land surface emissivity must be known a priori. The NDVI threshold method is an operationally simplified emissivity estimation method that is widely used to estimate emissivity from AVHRR observations (Liu et al., 2019; Ma et al., 2020; Sobrino et al., 2008). However, previous studies have combined this method with a fixed land cover dataset to determine the long-term emissivity (Frey et al., 2017; Ma et al., 2020; Reiners et al., 2021). As an intrinsic property of the surface, land surface emissivity predominantly depends on the land cover type, which is highly temporally

dynamic because of phenological changes and human activities. Therefore, to obtain relatively accurate emissivity values, we developed an improved method that considers annual changes in land cover from the GLASS-GLC dataset and combines ASTER GED data with the NDVI threshold method to estimate the emissivity (Fig. 3).

First, we assumed that the emissivity of an AVHRR pixel can be described as the weighted ensemble of bare soil emissivity and vegetation emissivity, where the weights are determined by the vegetation cover fraction:

$$\varepsilon_i = \varepsilon_{i,v} P_v + \varepsilon_{i,s}(1 - P_v) \tag{2}$$

Here, $\varepsilon_i$ is the emissivity in channel $i$, $\varepsilon_{i,v}$ is the vegetation emissivity in channel $i$, $\varepsilon_{i,s}$ is the bare soil emissivity in channel $i$, and $P_v$ is the fraction of vegetation cover, calculated as follows:

$$P_v = \frac{NDVI - NDVI_{min}}{NDVI_{max} - NDVI_{min}} \tag{3}$$

where $NDVI_{max}$ and $NDVI_{min}$ are the thresholds for pure vegetation and pure bare soil pixels, respectively. According to

Sobrino et al. (2001), $NDVI_{max}$ and $NDVI_{min}$ were set to 0.5 and 0.2, respectively. When NDVI is no more than 0.2, the pixel is assumed as pure bare soil with no vegetation cover; when NDVI is no less than 0.5, the pixel is assumed as pure dense vegetation.

Following Eq. (2), the bare soil component emissivity of ASTER channels 10–14 can be calculated as follows:

$$\varepsilon_{i,s}^{AST} = \frac{\varepsilon_i^{AST} - P_v \varepsilon_{i,v}^{AST}}{1 - P_v} \tag{4}$$

where $\varepsilon_{i,s}^{AST}$ is the bare soil emissivity in ASTER channel $i$ (i=10, …, 14), and $\varepsilon_{i,v}^{AST}$ is the emissivity of dense vegetation in ASTER channel $i$. Because the emissivity spectra of dense vegetation are similar and vary slightly in the TIR region, we used the dense vegetation emissivity of ASTER channel $i$ provided by Meng et al. (2016). $\varepsilon_i^{AST}$ is the emissivity of the ASTER GED product in channel $i$. $P_v$ is calculated from the NDVI of the ASTER GED product according to Eq. (3). For long-term cloud cover pixels and dense vegetation pixels ($P_v$=1), the bare soil emissivity of these ASTER pixels are null values. To

generate a global gap-free bare soil emissivity map of ASTER, we used the average emissivity of the same soil type within 5×5 neighborhood pixels to fill these null values. Because of some pixels with no valid neighbor pixels for averaging we needed to enlarge the neighborhood until all null values are filled. Soil-type data are described in Section 2.3.

Fig. 2 shows that the spectral range of ASTER channels 10–14 covers AVHRR channels 4 and 5. A linear regression relationship was used to convert the bare soil emissivity values from ASTER channels to AVHRR channels.

$$\varepsilon_{j,s}^{AVH} = b_0 + b_1\varepsilon_{10,s}^{AST} + b_2\varepsilon_{11,s}^{AST} + b_3\varepsilon_{12,s}^{AST} + b_4\varepsilon_{13,s}^{AST} + b_5\varepsilon_{14,s}^{AST} \qquad (5)$$

where $\varepsilon_{j,s}^{AVH}$ is the bare soil emissivity in AVHRR channel $j$ (j=4, 5), and $b_0$ to $b_5$ are the coefficients provided by Ma et al. (2020).

The emissivity of each vegetation type in the GLASS-GLC dataset was obtained from Ma et al. (2020). Specially, the vegetation type of a pixel was determined from the annual global land cover dataset (see Section 2.3). NDVI values were derived from the reflectance data of AVHRR channels 1 and 2 (see Section 3.1). In addition, the emissivity values of water pixels and ice/snow pixels were used to distinguish non-vegetated pixels. We then produced a daily dynamic global emissivity map for AVHRR channels 4 and 5. Further details can be found in Ma et al. (2020).

### 3.1.4 LST retrieval

To obtain the LST, we adopted the nonlinear GSW algorithm proposed by Wan (2014) because of its simplicity, efficiency, and high accuracy. The algorithm can be formulated as follows:

$$LST = a_0 + \left(a_1 + a_2\frac{1-\varepsilon}{\varepsilon} + a_3\frac{\Delta\varepsilon}{\varepsilon^2}\right)\frac{T_4+T_5}{2} + \left(a_4 + a_5\frac{1-\varepsilon}{\varepsilon} + a_6\frac{\Delta\varepsilon}{\varepsilon^2}\right)\frac{T_4-T_5}{2} + a_7(T_4 - T_5)^2 \qquad (6)$$

with $\varepsilon = (\varepsilon_4 + \varepsilon_5)/2$ and, $\Delta\varepsilon = \varepsilon_4 - \varepsilon_5$,

where $T_4$ and $T_5$ are the brightness temperatures measured in AVHRR channels 4 and 5, $\varepsilon_4$ and $\varepsilon_5$ are the land surface emissivity values in channels 4 and 5, $\varepsilon$ is the average emissivity for these two channels, $\Delta\varepsilon$ is the emissivity difference between these two channels, and $a_n$ (n = 0, 1, …, 7) are coefficients related to the WVC and satellite zenith angles.

The coefficient simulation for the nonlinear GSW algorithm is based on the radiative transfer theory in a cloud-free atmosphere (Fig. 3). The channel radiance received at the top of the atmosphere in the TIR channel of the sensor can be described using the radiative transfer theory:

$$L_i = \varepsilon_i B_i(T_s)\tau_i + R_i^{atm\uparrow} + (1 - \varepsilon_i)R_i^{atm\downarrow} \qquad (7)$$

where $L_i$ is the top-of-atmosphere radiance in channel $i$, $\varepsilon_i$ is the emissivity in channel $i$, $B_i$ is the Planck function, $T_s$ is the LST, $\tau_i$ is the total atmospheric transmittance in channel $i$, and $R_i^{atm\uparrow}$ and $R_i^{atm\downarrow}$ are the thermal path atmospheric upwelling and downwelling radiances in channels $i$, respectively.

To estimate the coefficients, the VZA sensor was set to 0°, 33.56°, 44.42°, 51.32°, 56.25°, and 60°. A moderate spectral resolution atmospheric transmittance algorithm and a computer model (MODTRAN, version=5.2) were run using 946 clear-sky atmospheric profile data to simulate the atmospheric parameters. By convolving these parameters with the spectral response functions of the two AVHRR TIR channels, we obtained the channel atmospheric parameters of each VZA, including

the total atmospheric transmittance, thermal path atmospheric upwelling, and downwelling radiances. To ensure that the simulation experiments were representative, the bottom air temperature ($T_{bat}$) of the profiles was adopted as the LST. Specifically, LST varies from $T_{bat}$–5 to $T_{bat}$+15 K in 5-K intervals for $T_{bat} \geq$290 K, and from $T_{bat}$–5 to $T_{bat}$+5 K in 5-K intervals for $T_{bat}$<290 K (Tang, 2018). In a subsequent step, we converted the LST, channel atmospheric parameters ($\tau_i$, $R_i^{atm\uparrow}$ and $R_i^{atm\downarrow}$), and channel emissivity mentioned earlier to brightness temperature using the radiative transfer theory (Eq. (7)). The brightness temperatures and LST were then used for coefficient estimation according to Eq. (6). To improve the fitting accuracy for each VZA mentioned above, the averaged emissivity values, WVC, and LST were divided into two, six, and five subranges, respectively. More details can be found in Tang et al. (2008) and Liu et al. (2018). The coefficients $a_0$ to $a_7$ in Eq. (6) were obtained using the least-squares method for each subrange.

Finally, the LST product was retrieved in two steps. In the known subranges of emissivity and WVC, the initial LST was estimated with coefficients derived for the entire range of LST, whereas the ultimate LST was estimated using coefficients for a suitable LST subrange determined by the initial LST (Tang, 2018).

## 3.2 LST validation

To assess the quality of the GT-LST product, two classical LST validation approaches were used in this study: ground-based validation (Göttsche et al., 2016; OuYang et al., 2017; Wang and Liang, 2008) and satellite product inter-comparison (Guillevic et al., 2014; Trigo et al., 2008). To further demonstrate the preponderance of this product, we also compared GT-LST with historical AVHRR LST products (i.e., GD-LST and RT-LST).

Ground-based validation was performed between in situ LST obtained at six stations in the SURFRAD network and GT-LST from 1994 to 2005. Four criteria were used to guarantee the validation results: (1) the two LST datasets were accurately matched under the condition of geolocation; (2) time differences between in situ LST and GT-LST acquisition of less than 3-min were permitted, as measurements were provided by the SURFRAD network every 3 min; (3) we only used high-quality data of GT-LST (QC=0) and in situ data with the quality flag corresponding to high-quality data; and (4) to further minimize the effect of cloud contamination, a popular method, "$3\sigma$-Hampel identifier", was employed to further remove cloudy samples (Duan et al., 2019).

$$S = 1.4628 \times median\{|x_k - x_m|\} \tag{8}$$

Here, $x_k$ is the differences between GT-LST and in situ LST, and $x_m$ is the median of the dataset $\{x_k\}$. Matchups with difference of less than $x_m - 3\sigma$ or greater than $x_m + 3\sigma$ were regarded as cloudy contamination.

In this study, satellite product inter-comparison was performed between GT-LST and the MODIS LST products (MYD11A1 and MYD21A1). Because these two MODIS LST products have provided daily LST since 2002, the comparisons were limited to data in 2004 (see Section 2.1). Five criteria were used to guarantee the validation results: (1) MODIS LST matched GT-LST in space; (2) because MODIS LST has a finer spatial resolution than GT-LST, MODIS LST was spatially aggregated to the GT-LST pixel scale with a simple arithmetic mean and a rigorous standard that all MODIS pixels within a GT-LST pixel

must be valid; (3) differences in the acquisition time between MODIS LST and GT-LST of less than 15-min were permitted; (4) differences in VZA between MODIS and GT-LST were not more than 15°; and (5) we only use high-quality LST values of MODIS (QC=0, i.e., good quality data with no need to examine more details) and GT-LST (QC=0).

In contrast to the ground-based validation and satellite product inter-comparison mentioned above, the comparisons for AVHRR LST products were performed using different strategies. Concretely, GT-LST during daytime was compared with that of GD-LST using a strategy that compares GT-LST and GD-LST with same SURFRAD measurements concurrently with the satellite overpass, to evaluate the difference in the absolute accuracy of these two products. GT-LST was compared with RT-LST using two strategies: (1) Two days, January 15 and July 15, 1997 were selected to implement the comparison over continental Africa because they represent the median time of different seasons (winter and summer, respectively); (2) because RT-LST has a coarser spatial resolution, the closest GT-LST LST values were extracted based on longitude and latitude of each pixel of RT-LST.

## 4. Results and discussion

### 4.1 Comparison with in situ LST

We first compared GT-LST data with in situ LST data at BND, DRA, FPK, GWN, PSU, SXF, and TBL sites from SURFRAD network for 1994–2005 (Fig. 4). Each scatterplot shows the overall validation count, root-mean-square error (RMSE), bias, standard deviation and coefficient of determination ($R^2$). First, the GWN site had the most data points matching the GT-LST, which meant that more data passed the validation criteria shown in Section 3.2 at this site than at other sites. The stations with the next highest number of matching data points were BND, DRA, FPK, and TBL. The stations PSU and SXF had the least valid points because the time period for these two sites was smaller (1998–2005 and 2003–2005). The overall RMSE range was approximately1.6–4.0 K (Fig. 4), 1.8–4.8 K for daytime observations and 1.0–3.3 K for nighttime observations (Table 4). The RMSEs of all sites except PSU for nighttime observations were less than 3.0 K. Compared to daytime observations, nighttime observations of all sites except GWN and PSU had better accuracy with lower RMSE. This is because in situ LST measurements during the daytime do not necessarily have good spatial representativeness for the satellite sensor footprint (Duan et al., 2019; Göttsche et al., 2016). In contrast, the LST was more spatially homogeneous at night. The BND site exhibited low accuracy with the largest RMSE and bias values; this result was also confirmed by previous studies (Liu et al., 2019; Ma et al., 2020; Reiners et al., 2021). A positive bias (GT-LST–in situ LST) was found for all SURFRAD sites except for daytime observations at the GWN stations. Furthermore, $R^2$ values between the retrieved LST and in situ LST ranged from 0.94 to 0.99, indicating a high correlation between these data. We further compared GT-LST data with in situ LST data at BAR, NYA, PYA, and TAT sites from BSRN network for 1995–2005. Fig. 5 shows the scatterplots between GT-LST and in situ LST at four BSRN sites. The accuracy of GT-LST product at BSRN sites is relatively worse than that at SURFRAD sites, with RMSE (bias) ranges from 3.1 K (-2.7 K) to 4.0 K (2.5 K). It should be noted that relatively poor accuracy at BSRN sites possible due to large spatial heterogeneity of LST at these sites.

Many studies have obtained similar results. For example, Duan et al. (2019) evaluated the accuracy of the Collection-6 MODIS LST data based on in situ LST observations and obtained large RMSE values (>2 K) during the daytime. Moreover, Martin et al. (2019) evaluated the accuracies of several LST products (AATSR, GOES, MODIS, and SEVIRI) based on multiple years of in situ LST observations, and concluded that the average daytime and nighttime accuracies over the entire time span were within ±4 K and ±2 K, respectively. Furthermore, Ma et al. (2020) and Liu et al. (2019) compared AVHRR LST with in situ LST during the daytime, and revealed RMSE variations of 2.3–3.9 K and 2.2–4.1 K, respectively. Therefore, the accuracy of GT-LST is encouraging.

## 4.2 Comparison with MODIS LST

An inter-comparison between GT-LST and MYD11A1 LST was performed on a global scale for 2004 (see Section 3.2). Specifically, Fig. 6 shows the daytime and nighttime root-mean-square-difference (RMSD) values of 3.4 K and 3.1 K and that of positive bias of 1.4 K and 2.4 K between GT-LST and MYD11A1 LST for 2004, respectively. This result is similar to that of Reiner et al. (2021), who compared a regional 1-km AVHRR LST product of the TIMELINE project with MODIS LST for 2003–2014, and reported RMSD and bias values of approximately 2.7 K and 2.2 K, respectively. However, as can be seen in the red box of Fig. 6, there are some considerable scattered samples (111 samples) which perform large LST differences (more than 20 K). Fig. A1 shows that all scattered samples are barren land cover type and arid climate type. About two-third of all samples (77 samples) happened in Haiya, Sudan on March 31, 2004. The samples of rest (34 samples) happened in Taif, Saudi Arabia on April 2, 2004. For these samples, we double-checked variables that are essential in GT-LST retrieval. The result showed that values of all variables are reasonable except BTs of TIR bands. Abnormal high BTs at these nighttime samples were found on March 31 and April 2, 2004 (Fig. A2), which leaded to extreme high LSTs. The possible reasons for abnormal high BTs may be instrument failure on these two days.

Fig. 7 shows the RMSD and bias between GT-LST and MYD11A1 LST for 2004 over various land cover types. The RMSD varied from 2.1 K to 4.2 K and the bias varied from approximately 0.6 K to 3.3 K. Specifically, savannas and cropland/natural vegetation mosaics had an RMSD of larger than 4 K. The permanent snow and ice and water bodies land cover types had an RMSD of less than 2.5 K, with the water bodies exhibiting the lowest RMSD of 2.1 K. We further analyzed the land cover types of different groups. Forests except deciduous broadleaf forests, including evergreen needleleaf forests, evergreen broadleaf forests, deciduous needleleaf forests and mixed forests, had an RMSD of less than 3K. Shrublands, including open shrublands and closed shrublands, had a similar RMSD of 3.3 K. Savannas and croplands, including woody savannas and savannas, croplands and cropland/natural vegetation mosaics, respectively, had the largest RMSD. The possible reason is that the fraction of vegetation cover of savannas and croplands vary greatly due to the influence of natural and human factors, which leads to the underestimation of emissivity comparing with fixed emissivity of MYD11A1, resulting in an overestimation of LST. Snow and ice and water bodies had the smallest RMSD.

Spring (March–May), summer (June–August), autumn (September–November) and winter (December–February) of 2004 were used to perform a seasonal inter-comparison at a global scale. Fig. 8 shows the GT-LST versus MYD11A1 LST during

different seasons. The plot shows a strong correlation, with $R^2$ values greater than 0.97, and a positive bias between GT-LST
and MYD11A1 LST in each season. The RMSDs of each season varied from approximately 3.0 K to 3.5 K. Moreover, we
observed a seasonal pattern, with a higher RMSD and bias in spring and summer and a lower RMSD and bias in autumn and
winter.

As noted above, these validation results are encouraging. However, GT-LST was overestimated when compared with
MYD11A1 LST. A reasonable explanation could be that the emissivity used for the retrieval of AVHRR LST was lower than
that of MYD11A1 LST. Specifically, the emissivity of MYD11A1 LST was derived from the classification-based method,
whereas that of GT-LST was derived from the NDVI threshold method, which considers annual changes in land cover and
dynamically retrieve daily emissivity. As a result, the dynamic emissivity of GT-LST is typically lower than that of MYD11A1,
which leads to overestimation of the LST (Hulley et al., 2016; Guillevic et al., 2014; Reiners et al., 2021; Ren et al., 2011).
Fig. A3 shows that the mean biases (GT-LST – MYD11A1) for LSTs calculated with emissivity differences less than -0.05,
between -0.05 and -0.03, between -0.03 and -0.01, between -0.01 and 0.01 and more than 0.01 are 7.0, 4.3, 2.3, 0.8 and 0.7 K,
respectively. To further demonstrate this point, we compared GT-LST with MYD21A1 LST. Fig. 9 shows the daytime and
nighttime RMSD values of 3.2 K and 2.5 K and that of bias of 0.1 K and 1.3 K between GT-LST and MYD21A1 LST for 4
months in 2004. Compared to the result of MYD11A1, the significantly smaller bias was obtained for MYD21A1. The possible
reason is attributed to the fact that the MYD21A1 LST uses the same observations with MYD11A1 but uses a physics-based
method to dynamically retrieve emissivity.

## 4.3 Comparison with existing AVHRR LST data

A recent study by Ma et al. (2020) generated a global historical daytime 0.05°×0.05° LST product from NOAA AVHRR data
for 1981–2000 (see Section 2.5). To further validate the GT-LST product, we compared these two LST products at the selected
SURFRAD sites (see Section 3.2). The results of the daytime comparison, shown in Fig. 10, were as follows. First, comparing
these two AVHRR LST products to the same in situ LSTs showed that both GT-LST and GD-LST obtained approximately
similar accuracies, with an overall RMSE of 3.0 K. Except for the BND and FPK stations, GT-LST showed higher accuracy
for all sites, especially GWN and PSU stations which had RMSE values of less than 2 K. All sites showed positive biases for
GT-LST other than GWN, whereas only BND and FPK had positive biases for GD-LST.

However, GD-LST data are limited in that they are only obtained during the daytime, which somewhat limits its practical
applications. Meteorology- and climatology-related applications require at least two instantaneous LSTs (i.e., one daytime
LST and one nighttime LST) to estimate temperature-based climate change indices such as the mean LST, extreme LST, and
LST range for different temporal scales. In contrast, the GT-LST product significantly improved the generation of the two
instantaneous LSTs per day (Fig. 11). Furthermore, many studies have shown that two satellite observations that are separated
by approximately 12 h can be used to estimate a relatively accurate daily and monthly mean LST (i.e., DMLST and MMLST)
(Chen et al., 2017; Liu et al., 2023; Xing et al., 2021). Therefore, it was possible to derive an estimate of the global accurate
DMLST and MMLST based on the average value of daytime and nighttime overpasses of the AVHRR sensors (Fig. 12). To

estimate MMLST, first obtain the mean instantaneous clear-sky LST at daytime and nighttime, and then use these mean values to estimate MMLST according to the simple linear regression method (see Appendix B). In order to validate the accuracy of MMLST results, we compared MMLST based on GT-LST with that of in situ LST observations from SURFRAD sites for 1994–2005. All in situ LST measurements are all-sky and complete on a certain month, which means that the in situ MMLST is true MMLST. Fig. 13 showed that MMLST derived from GT-LST are related to the true MMLST, with an $R^2$ value of 0.94 and an RMSE value of 2.7 K. This result is similar to that of Chen et al. (2017), who compared MMLST from MODIS day and night instantaneous clear-sky LST with actual MMLST from 156 flux tower stations, and reported RMSE values of approximately 2.7 K. However, it should be noted that a positive bias of 1.3 K between GT-LST MMLST and in situ MMLST. One possible reason is that in situ MMLST of some sites does not represent the MMLST over the 0.05°×0.05° pixel.

Moreover, a comparison between GT-LST and RT-LST was performed during daytime and nighttime over continental Africa on January 15 and July 15, 1997 (Fig. 14). As can be seen, GT-LST and RT-LST had an RMSD of more than 2.1 K and a bias of more than 1.1 K. A likely explanation is that the emissivity of GT-LST is lower than that of RT-LST, which leads to overestimation of the LST. Compared to daytime LST, nighttime LST had an improvement with lower RMSD due to the comparatively spatially homogeneous LST during night. Furthermore, the RMSD of the July 15 is distinctly higher than January 15 because the atmospheric condition is hot and wet in July 15, cool and dry in January 15.

### 4.4 Benefits, limitations, and future prospects

To the best of our knowledge, a global historical twice-daily LST dataset for the period 1981–2021 has never before been generated because of the limitations of large amounts of original Level-1b data handling (i.e., approximately 14 TB), huge amounts of process variable data generation (i.e., approximately 10 TB), and complicated data processing flow design. Based on the experience of other research institutions and scholars, we generated the GT-LST product based on AVHRR observations, which showed advantages in spatial coverage and temporal resolution compared to existing studies. Moreover, to obtain a relatively accurate emissivity, we used an improved method that considers annual changes in land cover to estimate the emissivity. The GT-LST product, with two observations every day, can provide daily, monthly and yearly mean LST datasets. This can reduce the number of gaps and uncertainty in instantaneous LST data. Furthermore, the mean LST is more valuable than the instantaneous LST for global climate change. Although many LST products can provide global twice-daily LST after 2000, we still extend the time span of GT-LST to 2021. In this way, users can obtain a relatively homogeneous twice-daily LST product over a long time series. Furthermore, the overlapping observations between the GT-LST product and other LST datasets during the extension period can be used to calibrate the bias when combining these datasets. In conclusion, the GT-LST product is suitable for detecting climate changes over the past 40 years, such as global extreme LST changes and trends of global mean LST.

However, it should be noted that observations of equatorial crossing time for NOAA afternoon satellites become progressively later after launch (Fig. 15). As the orbit drifts, the AVHRR sensors change the illumination conditions and local solar time of observations. Users are therefore urged to be cautious when using the AVHRR LST product, especially in the LST range. The

timing of the occurrences of maximum and minimum LST is approximately 13:30 local solar time and 01:30 local solar time, respectively, which corresponds to the initially observed time of NOAA afternoon satellites. However, the overpass time of these satellites gradually drifts backward because of drift in the satellite orbits over time. For example, the initial NOAA-14 overpass time was 13:30 local solar time (descending) in 1994, but had shifted to 16:30 local solar time by the end of 2000. Although several studies have proposed correction methods for this problem, the accuracy of the AVHRR LST after orbital

drift correction is lower than that without orbital drift correction (Liu, et al., 2019). Although the GT-LST product extends the time span of LST data, it has a number of missing values (Fig. 11). For MMLST, it still has a few gaps (Fig. 12(b)). A variety of factors such as cloud cover, orbital gaps, and instrument failure are responsible for this limitation. And finally, the geolocation accuracy of GT-LST product basically meets the demand of global applications at 0.05° spatial resolution. However, if users need very high geolocation quality GT-LST data, we suggested that the GT-LST data with VZAs less than

40° should be preferred.

In summary, future work should focus on the following: (1) to alleviate the orbit drift effect, researchers should develop a new orbit drift correction method based on two observations every day; (2) to fill in the missing values, the product could be combined with microwave sensors or an annual temperature cycle model could be employed; (3) to further analyze climate change, it is essential to generate mean and extreme LST datasets based on the GT-LST product; and (4) to combine the GT-

LST product with other LST datasets, it is necessary to research how to calibrate the bias between the GT-LST and other LST datasets.

## 5 Data Availability

The global historical twice-daily (daytime and nighttime) LST product (GT-LST) at 0.05° spatial resolution from 1981 to 2021 is available at https://doi.org/10.5281/zenodo.7113080 (1981-2000) (Li et al., 2022a), https://doi.org/10.5281/zenodo.7134158

(2001-2005) (Li et al., 2022b), and https://doi.org/10.5281/zenodo.7813607 (2006-2021) (Li et al., 2023).

## 6 Conclusions

In this study, we developed a global historical twice-daily LST product with two observations per day for 1981–2021, which was designed to fill the gap in long-term global observations. First, we proposed a framework for generating an AVHRR historical instantaneous LST dataset with two observations every day from 1981 to 2021. The framework contains four major

segments: (1) data reading, calibration and pre-processing using open-source Python packages; (2) cloud detection based on the published CLAVR-1 algorithm; (3) land surface emissivity estimation using the NDVI threshold algorithm considering annual land cover changes; and (4) LST retrieval based on a nonlinear generalized split-window algorithm. We used the proposed method to generate a global 0.05°×0.05° twice-daily (daytime and nighttime) AVHRR LST product from 1981 to

2021, which also contained helpful ancillary products, including the recorded UTC time of observations, VZA, cloud mask, and latitude and longitude data.

To assess the accuracy of this product, we employed three evaluation methods. Ground-based validation, which involved a comparison between the GT-LST product and multi-year SURFRAD and BSRN in situ measurements from 1994 to 2005, showed that the $R^2$ values of all selected data were greater than 0.92 and the overall RMSE range was approximately 1.6–4.0 K; 1.8–4.8 K for daytime observations and 1.0–4.2 K for nighttime observations. These results suggested competitive accuracy with other satellite-derived LST products. Inter-comparison with the satellite products MYD11A1 and MYD21A1 LST showed that: (1) in 2004, the overall RMSD was 3.2 K and the bias was 1.8 K between GT-LST and MYD11A1 LST; (2) according to RMSD values between GT-LST and MYD11A1 LST, nighttime data were more accurate than daytime, as LST is more spatially homogeneous at night; (3) a higher RMSD and bias between GT-LST and MYD11A1 LST were observed in spring and summer, whereas a lower RMSD and bias were observed in autumn and winter; and (4) compared to the result of MYD11A1, the significantly smaller bias was obtained for MYD21A1. Comparisons with existing AVHRR LST products (i.e., GD-LST and RT-LST) showed that: (1) GT-LST and GD-LST products at the selected measurements of SURFRAD sites exhibited similar accuracies, with an overall RMSE of 3.0 K; (2) GT-LST showed a substantial improvement from GD-LST that is only obtained during the daytime, because it generates two instantaneous LST values (daytime and nighttime) every day and then can estimate the global DMLST and MMLST; (3) daytime and nighttime observations of GT-LST can provide relatively accurate MMLST under all-sky conditions, with RMSE of 2.7 K; and (4) compared with RT-LST over continental Africa in different seasons, the results showed that the RMSD range was 2.1–4.1 K and the bias range was 1.1–3.4 K.

## Appendix A: Supplementary tables and figures

**Table A1 Statistics for the relationship between the regressions of the eight combinations and actual monthly mean LST.**

| Case | Combinations (daytime/nighttime) | $a_1$ | $a_2$ | b | RMSE | $R^2$ | Number |
|------|----------------------------------|-------|-------|---|------|-------|--------|
| 1 | 13:30/01:30 | 0.3844 | 0.5783 | 10.3446 | 2.0 | 0.97 | 12095 |
| 2 | 14:00/02:00 | 0.4010 | 0.5621 | 10.2042 | 1.9 | 0.98 | 12241 |
| 3 | 14:30/02:30 | 0.4235 | 0.5451 | 8.6172 | 1.9 | 0.98 | 12381 |
| 4 | 15:00/03:00 | 0.4490 | 0.5211 | 8.2652 | 1.8 | 0.98 | 12303 |
| 5 | 15:30/03:30 | 0.4816 | 0.4840 | 9.5710 | 1.8 | 0.98 | 12165 |
| 6 | 16:00/04:00 | 0.5250 | 0.4349 | 11.2284 | 2.0 | 0.97 | 11818 |
| 7 | 16:30/04:30 | 0.5663 | 0.3884 | 12.8572 | 2.2 | 0.96 | 10992 |
| 8 | 17:00/05:00 | 0.6040 | 0.3621 | 9.7302 | 2.4 | 0.96 | 9765 |

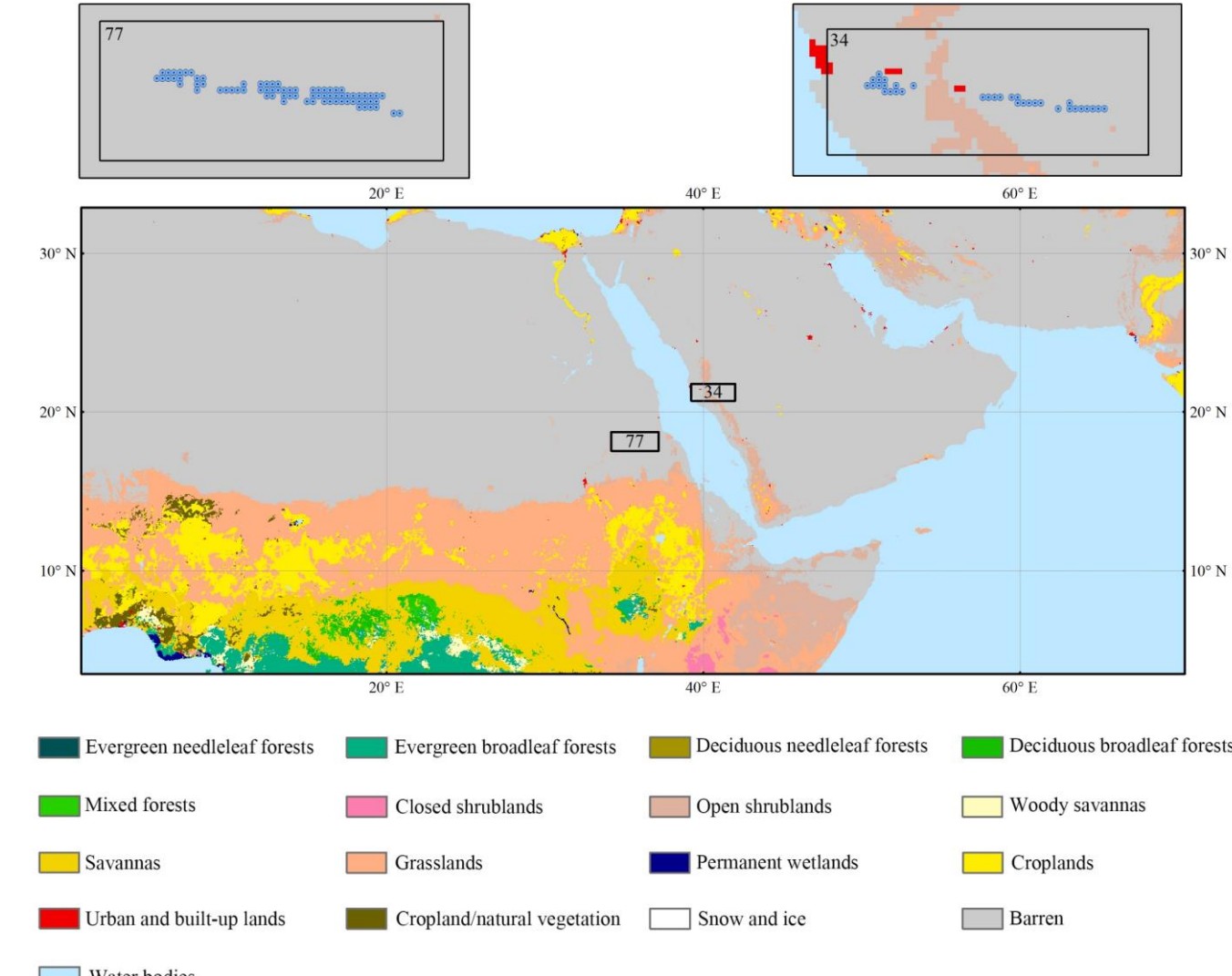

**Figure A1. Distribution of the 111 scattered samples.**

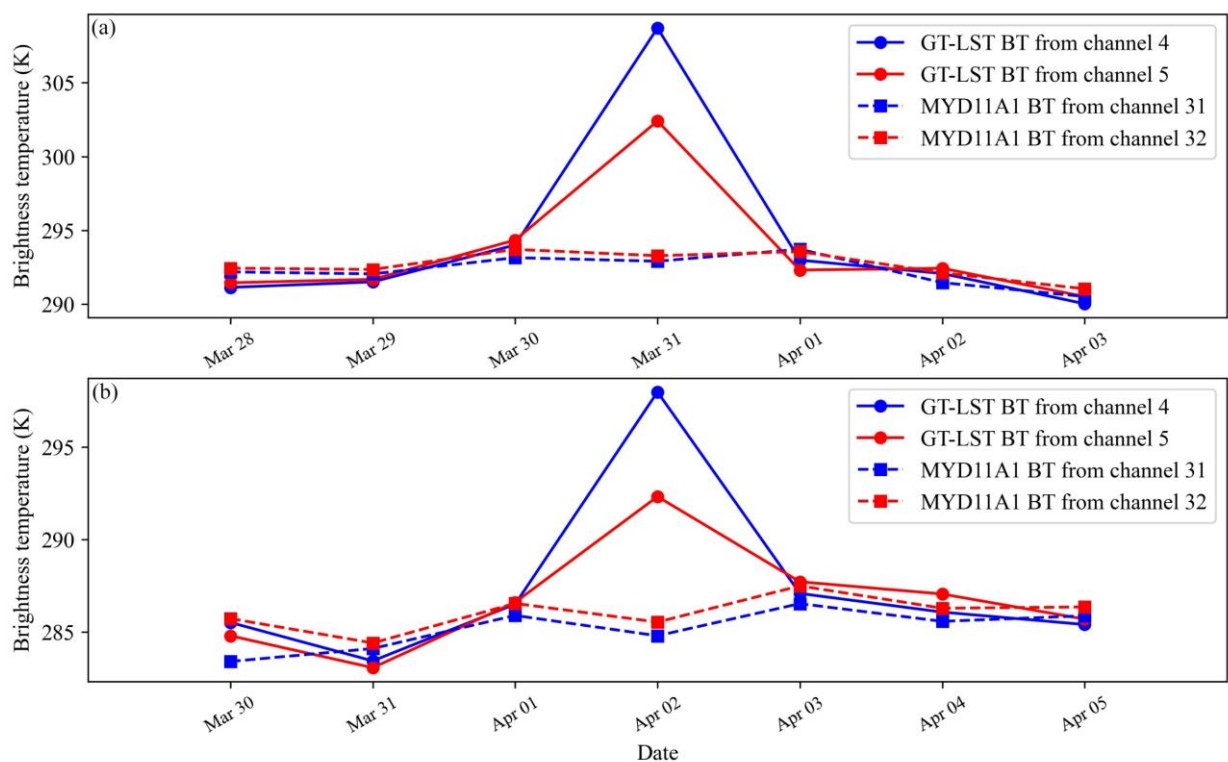

**Figure A2. An example of abnormal high BTs on (a) March 31, 2004 and (b) April 2, 2004.**

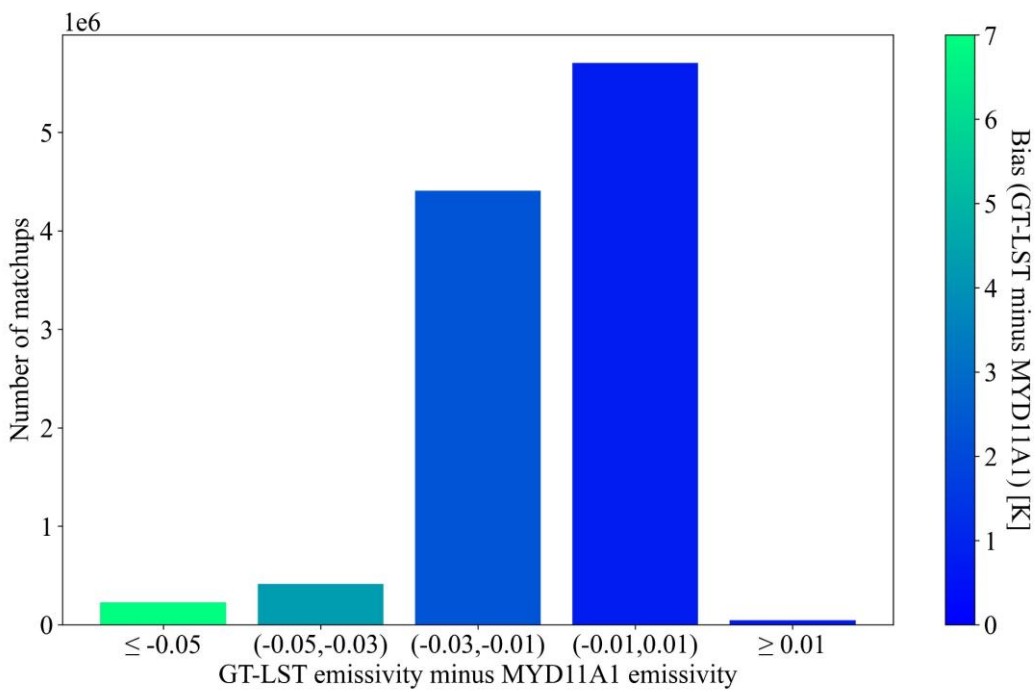

**Figure A3: Difference between GT-LST and MYD11A1 LST stratified by the difference between GT-LST and MYD11A1 emissivity (water vapor content < 5 g cm⁻²; satellite zenith angle < 50°).**

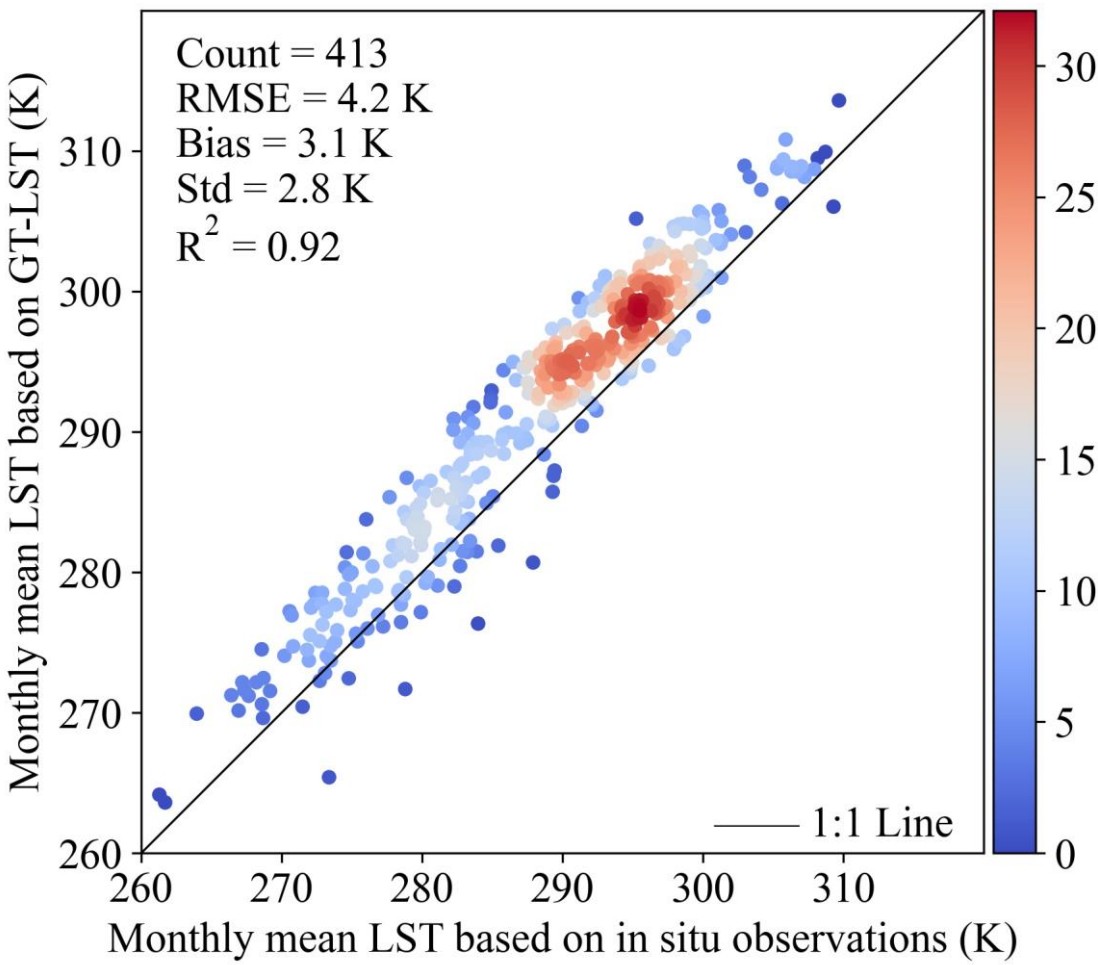

**Figure A4: Monthly mean LST based on GT-LST using simple average method versus monthly mean LST based on in situ LST from 1994 to 2005.**

**Appendix B: Detailed description of the monthly mean LST**

Impacting of the NOAA satellite orbital drift, daytime and nighttime observations of NOAA afternoon satellites cannot represent maximum and minimum temperatures well. Therefore, the monthly mean LST (MMLST) according to the simple
average method has a significantly lower accuracy than other studies (Fig. A4). Xing et al. (2021) proposed to use 9 combinations of two to four MODIS instantaneous retrievals of which at least one daytime LST and one nighttime LST to

estimate mean LSTs, and determined the weight for every moment. Inspired by the work of Xing et al. (2021), we determined to use simple linear combinations of monthly mean daytime and nighttime LST values that were observed at observation times for NOAA to estimate MMLST with ground-based measurement. For the combinations of two valid monthly mean LSTs (one daytime and one nighttime LST), the regression models can be written as follows:

$$MMLST = a_1 * MMLST_{day} + a_2 * MMLST_{night} + b \tag{B1}$$

where $MMLST$ is the ground-based monthly mean LST, $a_1$, $a_2$ and $b$ are the fitting coefficients, $MMLST_{day}$ is the monthly mean in situ LST at the NOAA daytime observation, $MMLST_{night}$ is the monthly mean in situ LST at the NOAA nighttime observation.

Taking into account the observed times of NOAA satellites with orbital drift effect since 1981, combinations of two observations from these satellites contain eight cases: 13:30–17:00/01:30–05:00 local solar time in 0.5-hour interval. Based on the in situ LST measurements during the period 2003 to 2018 at 227 flux stations operating in globally diverse regions, we determined the fitting coefficients (listed in Table A1). Subsequently, we used these coefficients along with GT-LST monthly mean daytime and nighttime LSTs and Eq. (B1) to calculate the MMLST of GT-LST.

## Author Contributions

JHL, XL and ZLL contributed to designing the research; JHL implemented the research and wrote original draft; XL, ZLL and SBD supervised the research; all co-authors revised the manuscript and contributed to the writing.

## Competing Interests

The authors declare that they have no conflict of interest.

## Acknowledgments

This work was supported by the National Natural Science Foundation of China (Grant No. 41921001 and 42101371), the Major S&T project (Innovation 2030) of China (Grant No. 2021ZD0113701), and the China Postdoctoral Science Foundation (Grant No. 2020M680774). We would like to thank Zenodo for publishing the dataset. We also acknowledge NOAA Comprehensive Large Array-Data Stewardship System for providing the AVHRR GAC data, National Aeronautics and Space Administration for providing the MODIS data, SURFRAD and BSRN for providing the in situ measurement. We acknowledge the valuable comments and suggestions from four anonymous referees. Thanks are also extended to Wenhui Du, Menglin Si, Cheng Huang, and Zihao Wu for downloading and processing the data.

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

**Tables**

**Table 1 Characteristics of LST products generated with AVHRR data.**

| Dataset | Spatial coverage | Time span | Temporal resolution | Spatial resolution | References |
|---|---|---|---|---|---|
| The World Land Surface Temperature Atlas dataset | Europe | 1992–1993 | Monthly | 1-km and 0.5° | Kerr et al. (1998) |
| RT-LST | Africa | 1995–2000 | Daytime and Nighttime | 8-km | Pinheiro et al. (2006) |
| Annual and seasonal LST dataset over Peninsular Spain | Peninsular Spain | 1981–2015 | Annual and Seasonal | 1.1-km | Khorchani et al. (2018) |
| TIMELINE LST dataset | European and North Africa | 1981–2021 | Daytime and Nighttime | 1-km | Frey et al. (2012); Frey et al. (2017); Reiners et al. (2021); Holzwarth et al. (2021) |
| ALP-II LST dataset | Global | 1989 | Monthly | 8-km | Ouaidrari et al. (2002) |
| LSTD | Global | 1981–1998 | Monthly | 8-km, 0.5° and 5° | Jin (2004) |
| GD-LST | Global | 1981–2000 | Daytime | 0.05° | Ma et al. (2020) |


**Table 2 Spectral band widths (μm) of AVHRR sensors.**

| Channel | AVHRR-2 (NOAA-7,9,11,14) | AVHRR-3 (NOAA-15 to19, Metop-A,B) | Main application |
|---|---|---|---|
| 1 | 0.58–0.68 | 0.58–0.68 | ice/snow, daytime clouds |
| 2 | 0.725–1.10 | 0.725–1.10 | vegetation cover, land/water boundaries |
| 3A | NA | 1.58–1.64 | dust monitoring, snow/ice detection |
| 3B | 3.55–3.93 | 3.55–3.93 | nighttime clouds, volcanic eruptions |
| 4 | 10.3–11.3 | 10.3–11.3 | sea/land surface temperature, daytime/nighttime imagery |
| 5 | 11.5–12.5 | 11.5–12.5 | sea/land surface temperature, daytime/nighttime imagery |

Note: NA means the channel not available

**Table 3 Details of the validation sites used in this study.**

| | Name | Elevation(m) | Land cover type | Latitude | Longitude | Valid period |
|---|---|---|---|---|---|---|
| SURFRAD | BND | 230 | Croplands | 40.0519 | −88.3731 | 1995–2005 |
| | DRA | 1007 | Open shrublands | 36.6237 | −116.0195 | 1998–2005 |
| | FPK | 634 | Grasslands | 48.3078 | −105.1017 | 1994–2005 |
| | GWN | 98 | Cropland/natural vegetation mosaic | 34.2547 | −89.8729 | 1994–2005 |
| | PSU | 376 | Cropland/natural vegetation mosaic | 40.7201 | −77.9309 | 1998–2005 |
| | TBL | 1689 | Grasslands | 40.1250 | −105.2368 | 1995–2005 |
| | SXF | 473 | Croplands | 43.7343 | −96.6233 | 2003–2005 |
| BSRN | BAR | 8 | tundra | 71.3230 | -156.6070 | 1995–2005 |
| | NYA | 11 | tundra | 78.9227 | 11.9273 | 1999–2005 |
| | PAY | 491 | cultivated | 46.8123 | 6.9422 | 1995–2005 |
| | TAT | 25 | grass | 36.0581 | 140.1258 | 1996–2005 |

**Table 4. GT-LST versus in situ LST during the daytime and nighttime**

| | Site | Day | | | Night | | |
|---|---|---|---|---|---|---|---|
| | | Count | RMSE(K) | Bias(K) | Count | RMSE(K) | Bias(K) |
| SURFRAD | BND | 760 | 4.8 | 3.6 | 565 | 2.6 | 1.8 |
| | DRA | 747 | 2.7 | 1.0 | 533 | 1.9 | 0.6 |
| | FPK | 731 | 3.2 | 2.2 | 435 | 2.0 | 1.0 |
| | GWN | 1193 | 1.9 | -0.5 | 840 | 2.1 | 1.5 |
| | PSU | 431 | 1.9 | 0.1 | 331 | 3.3 | 1.8 |
| | SXF | 250 | 1.8 | 0.1 | 146 | 1.0 | 0.1 |
| | TBL | 631 | 2.2 | 0.7 | 488 | 2.1 | 1.4 |
| BSRN | BAR | 166 | 3.4 | -1.8 | 163 | 3.1 | -0.5 |
| | NYA | 125 | 3.9 | -2.7 | 53 | 4.2 | -2.6 |
| | PAY | 607 | 3.9 | 3.3 | 249 | 3.7 | 2.2 |
| | TAT | 599 | 3.3 | -1.6 | 530 | 2.9 | 2.2 |


**Figures**

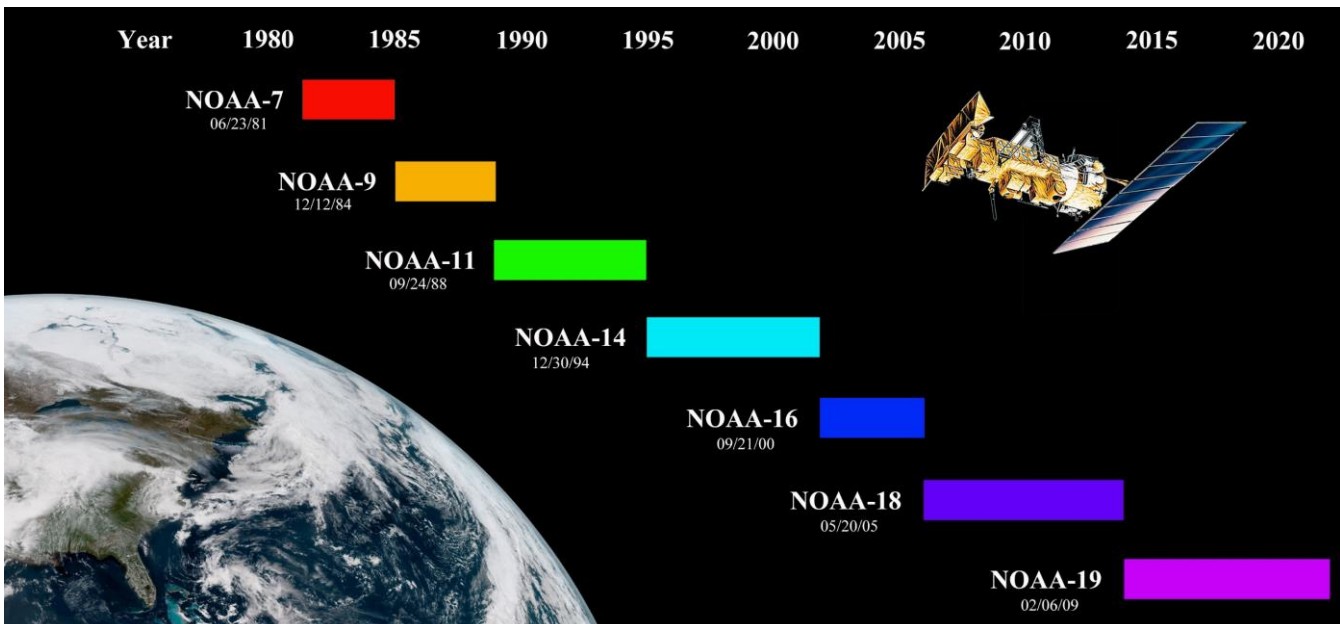

**Figure 1: Coverage period of NOAA satellites used in this study (adapted from http://www.nasa.gov/pdf/111742main_noaa_n_booklet.pdf).**

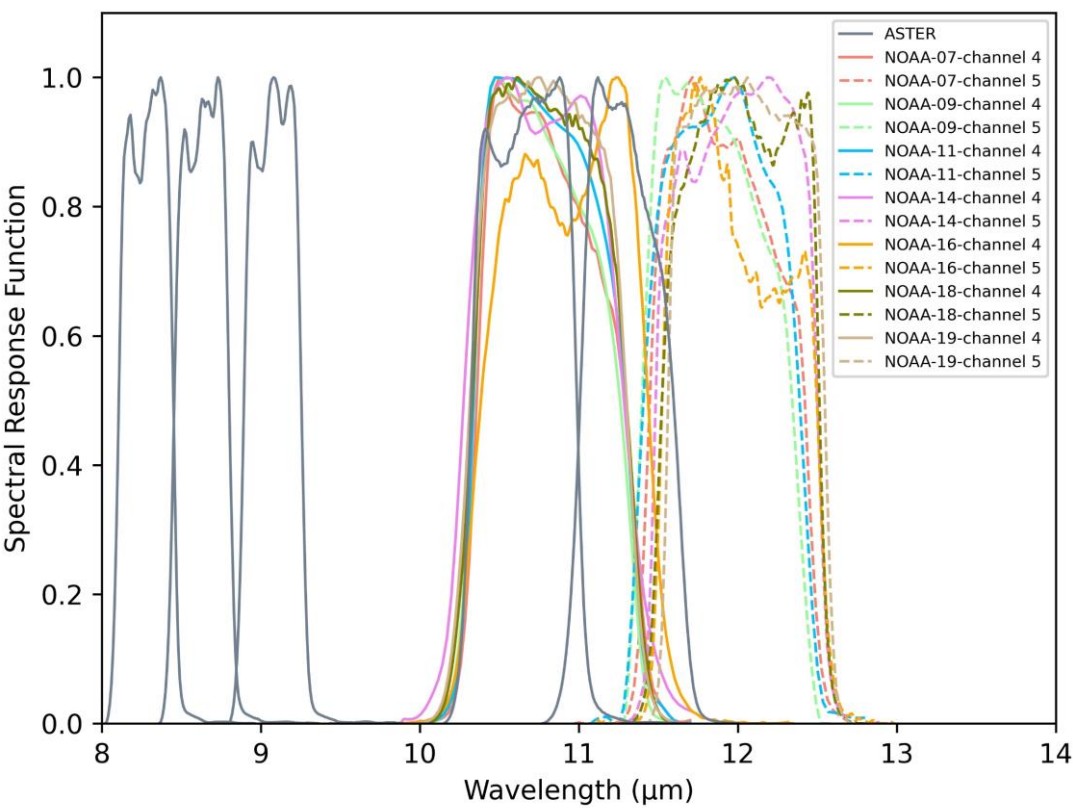


**Figure 2: Spectral response functions of NOAA-7/9/11/14/16/18/19 channel 4 and 5 and ASTER channel 10 to 14.**

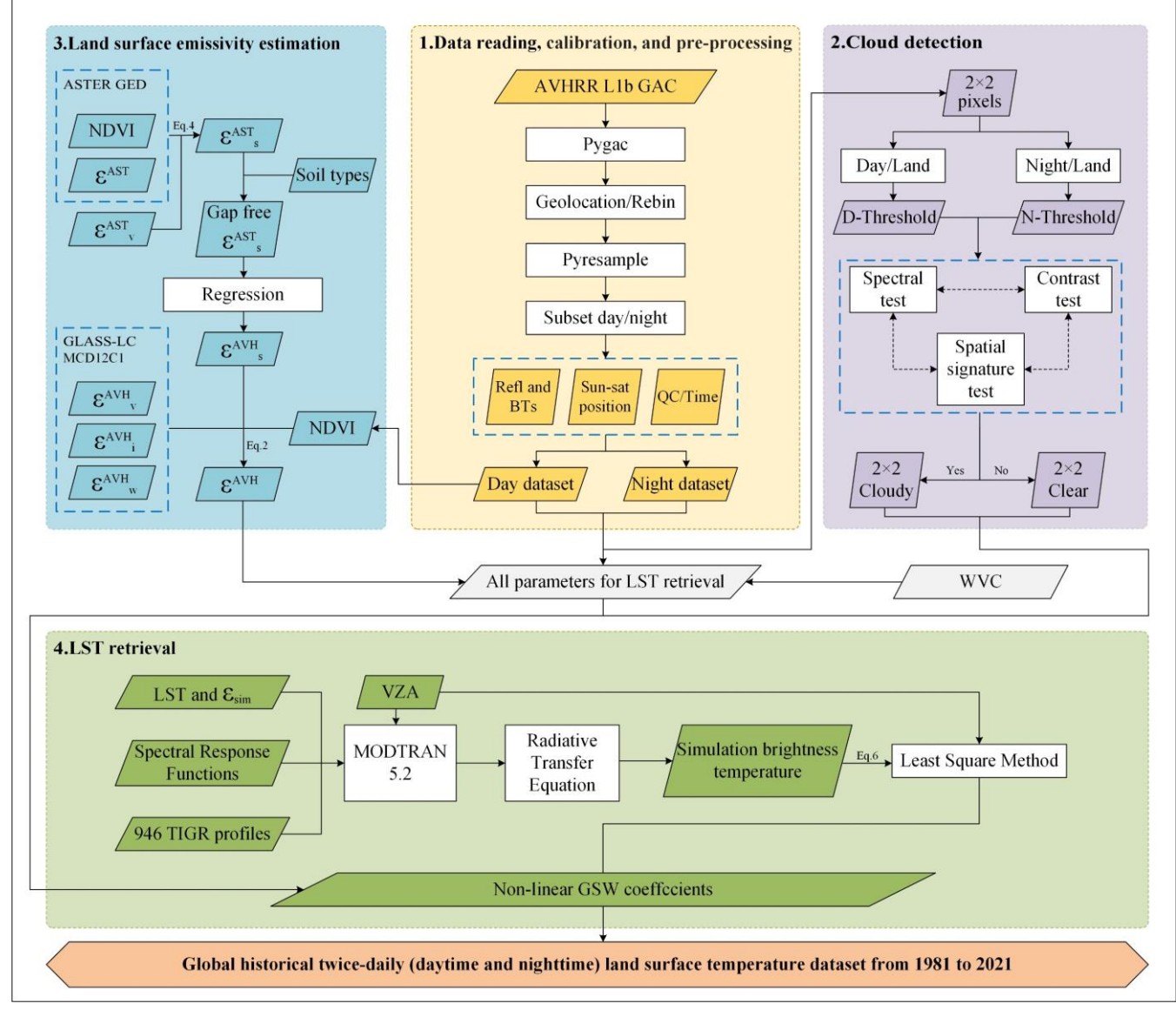

**Figure 3: Schematic of the workflow used to generate the GT-LST product.**

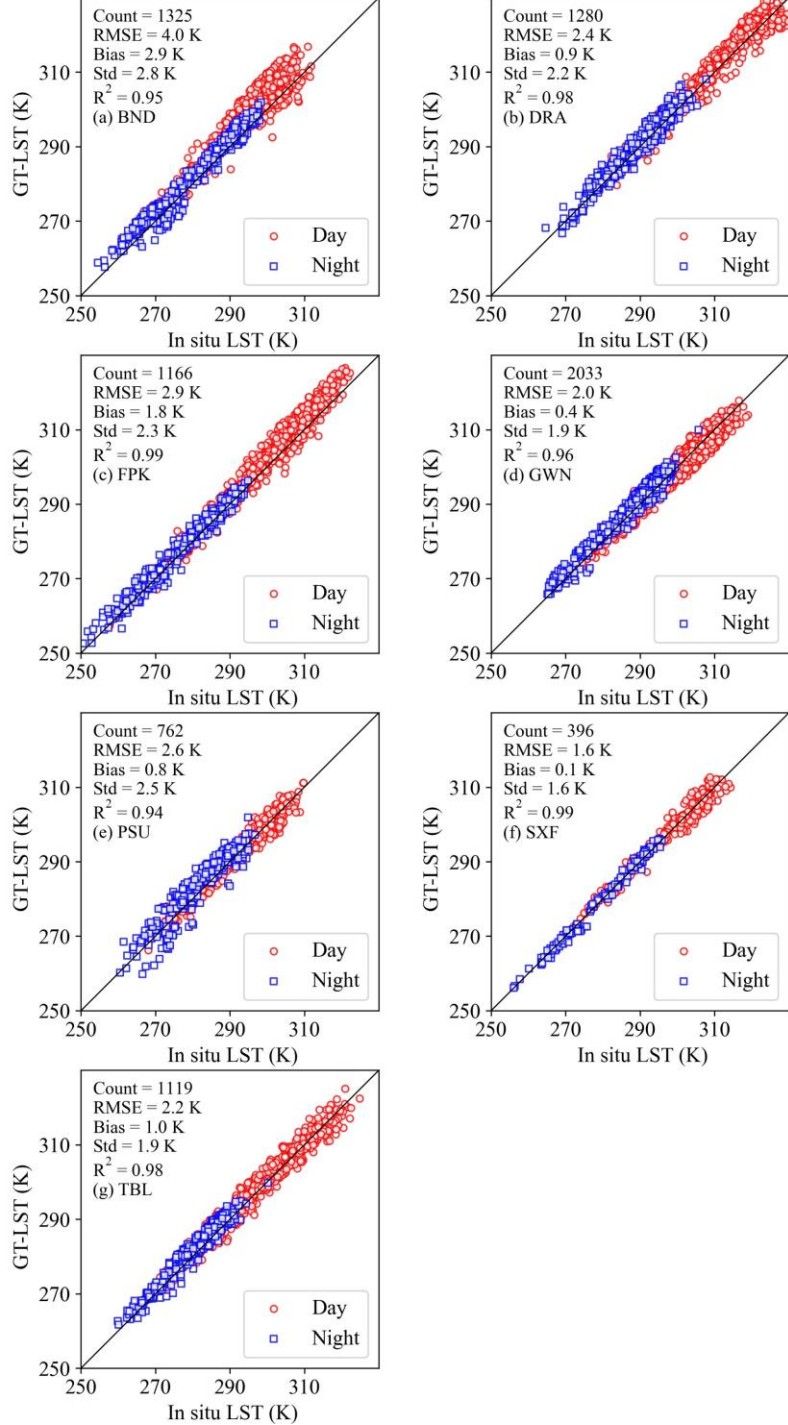

**Figure 4. GT-LST versus in situ LST for 1994–2005 at (a) BND, (b) DRA, (c) FPK, (d) GWN, (e) PSU, (f) SXF, and (g) TBL sites.**

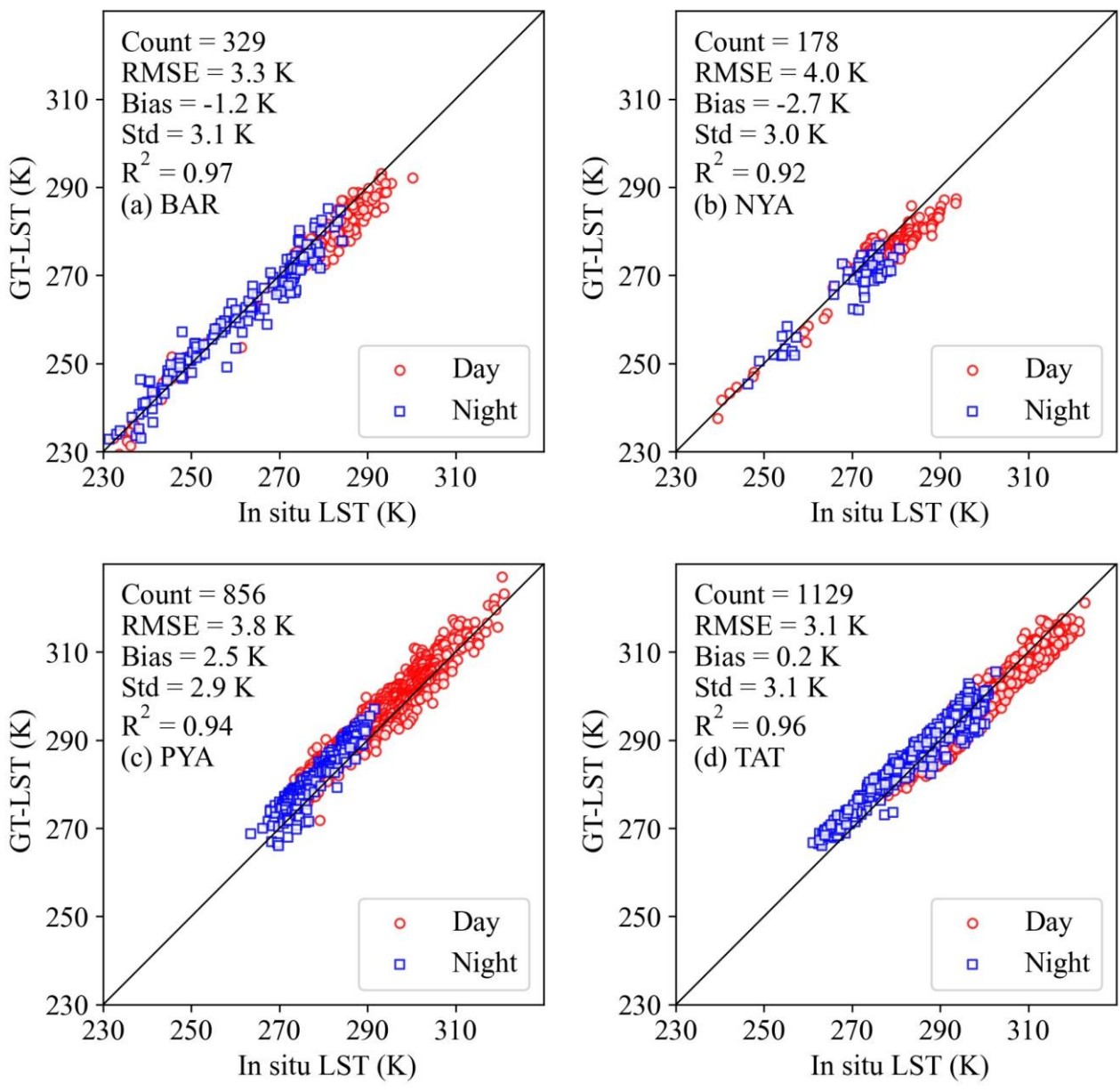

**Figure 5: Scatterplots between GT-LST and in situ LST at (a) BAR, (b) NYA, (c) PYA, and (d) TAT.**


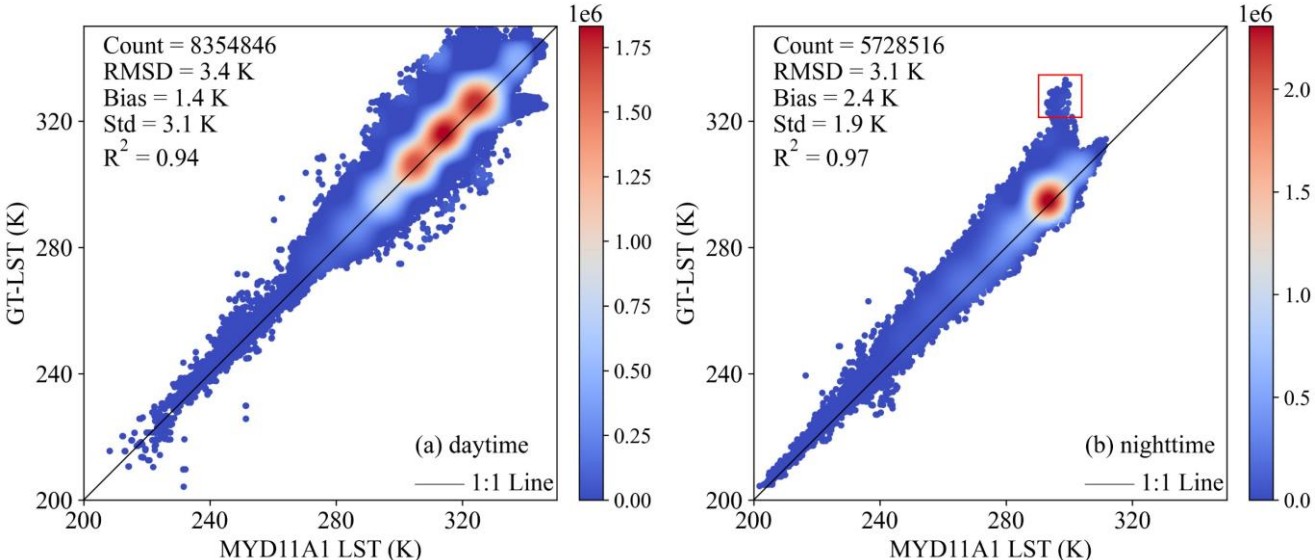

Figure 6: Inter-comparison of GT-LST and MYD11A1 LST in 2004: (a) daytime; (b) nighttime.

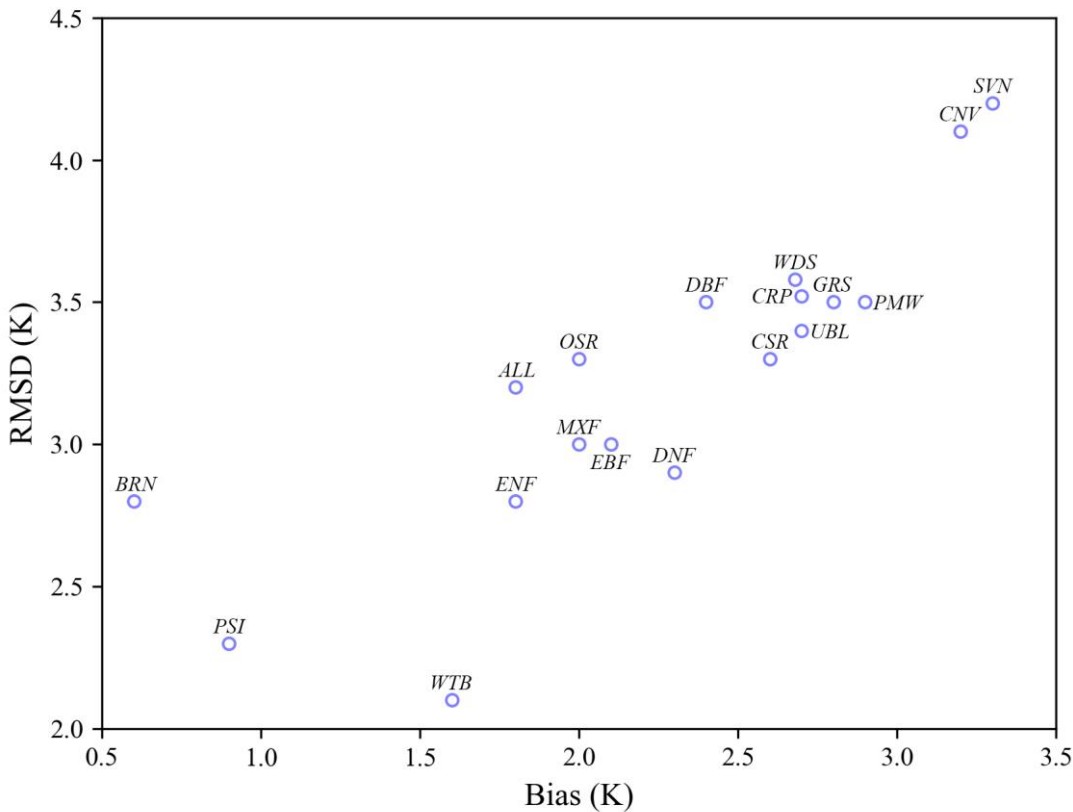


**Figure 7: RMSD and bias between GT-LST and MYD11A1 LST in 2004 for various land cover types. ENF: evergreen needleleaf forests, EBF: evergreen broadleaf forests, DNF: deciduous needleleaf forests, DBF: deciduous broadleaf forests, MXF: mixed forests, CSR: closed shrublands, OSR: open shrublands, WDS: woody savannas, SVN: savannas, GRS: grasslands, PMW: permanent wetlands, CRP: croplands, UBL: urban and built-up lands, CNV: cropland/natural vegetation mosaics, PSI: permanent snow and**
**ice, BRN: barren, WTB: water bodies, and ALL: all land cover types.**

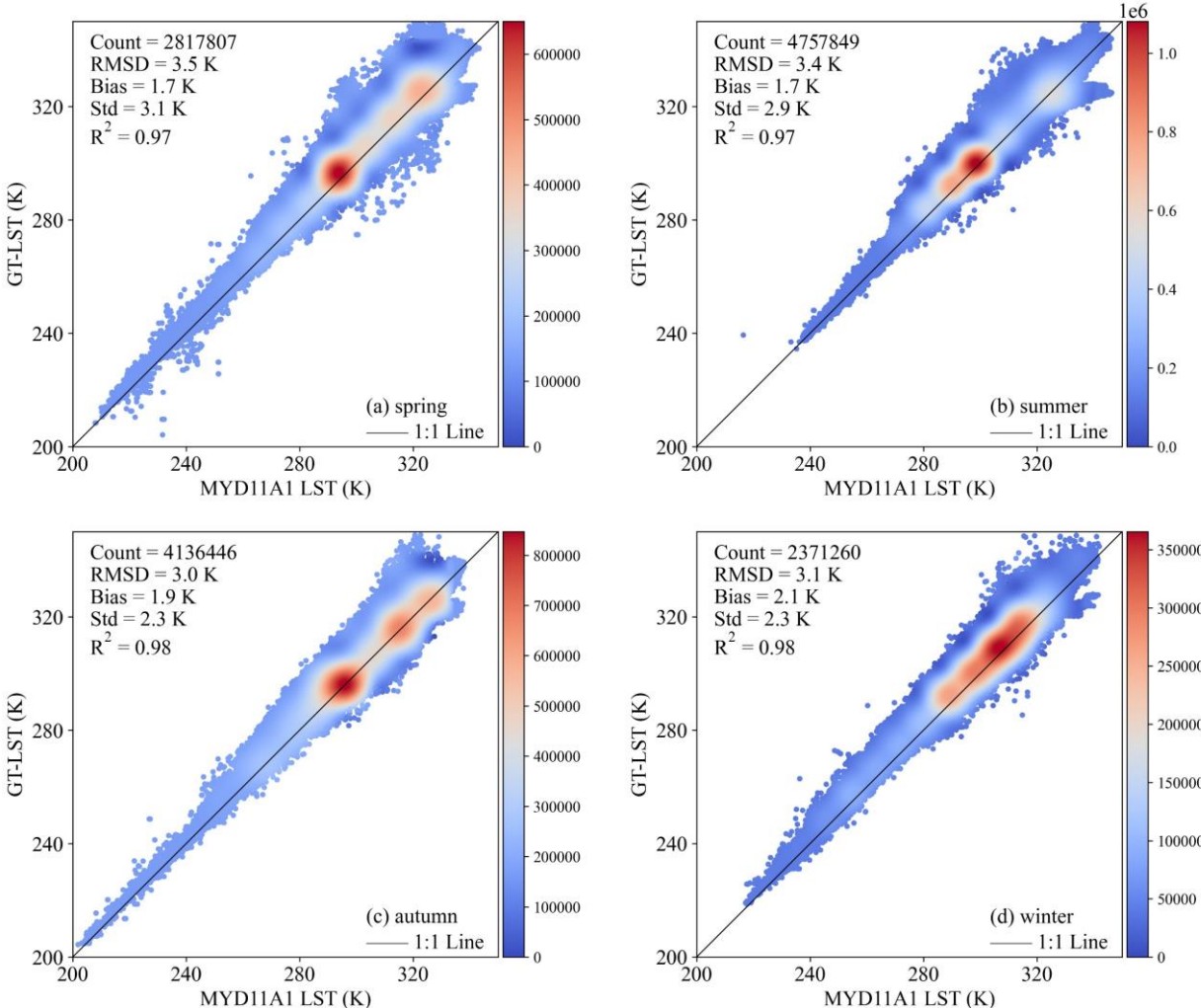

**Figure 8: Inter-comparison of GT-LST and MYD11A1 LST in 2004: (a) spring; (b) summer; (c) autumn; (d) winter.**

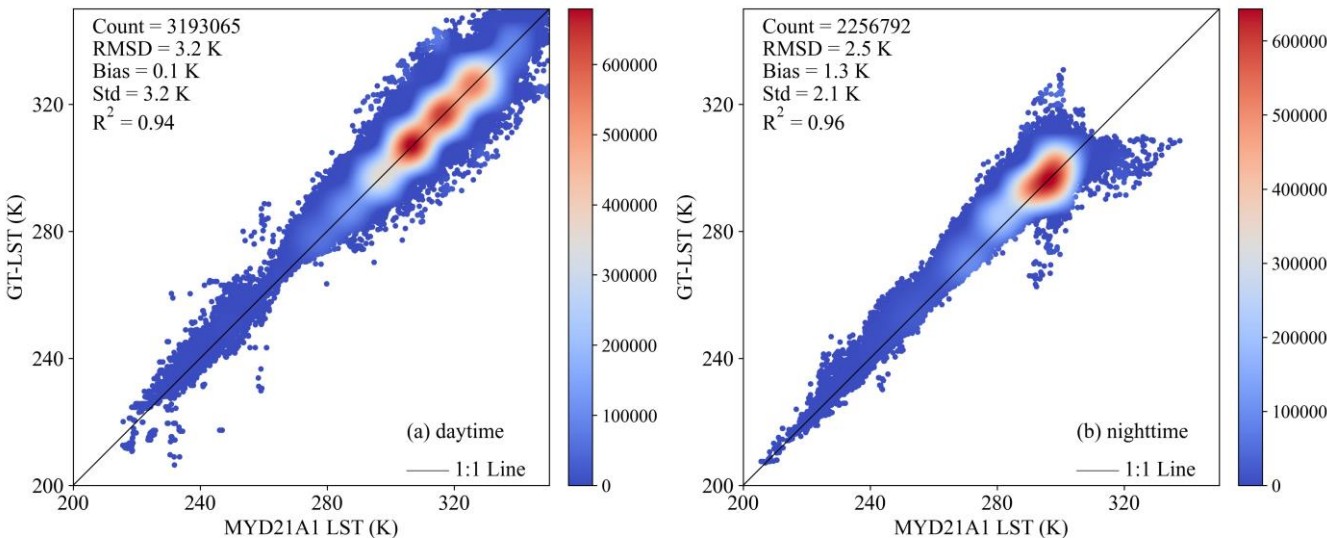

Figure 9: Inter-comparison of GT-LST and MYD21A1 LST in January, April, July, and October 2004: (a) daytime; (b) nighttime.

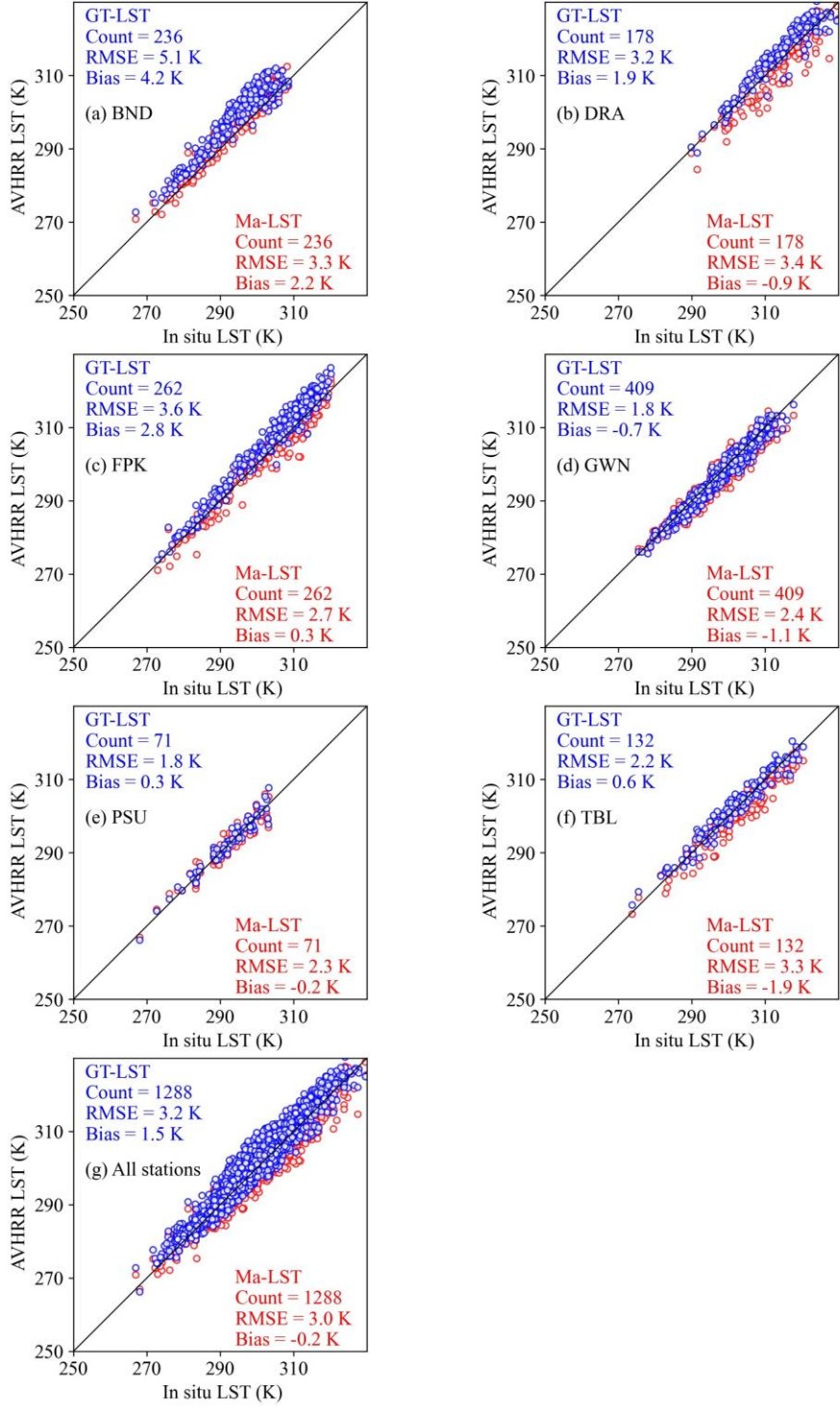

**Figure 10: Inter-comparison of GT-LST and GD-LST with in situ LST during the daytime at (a) BND, (b) DRA, (c) FPK, (d) GWN, (e) PSU, (f) TBL, and (h) all stations.**

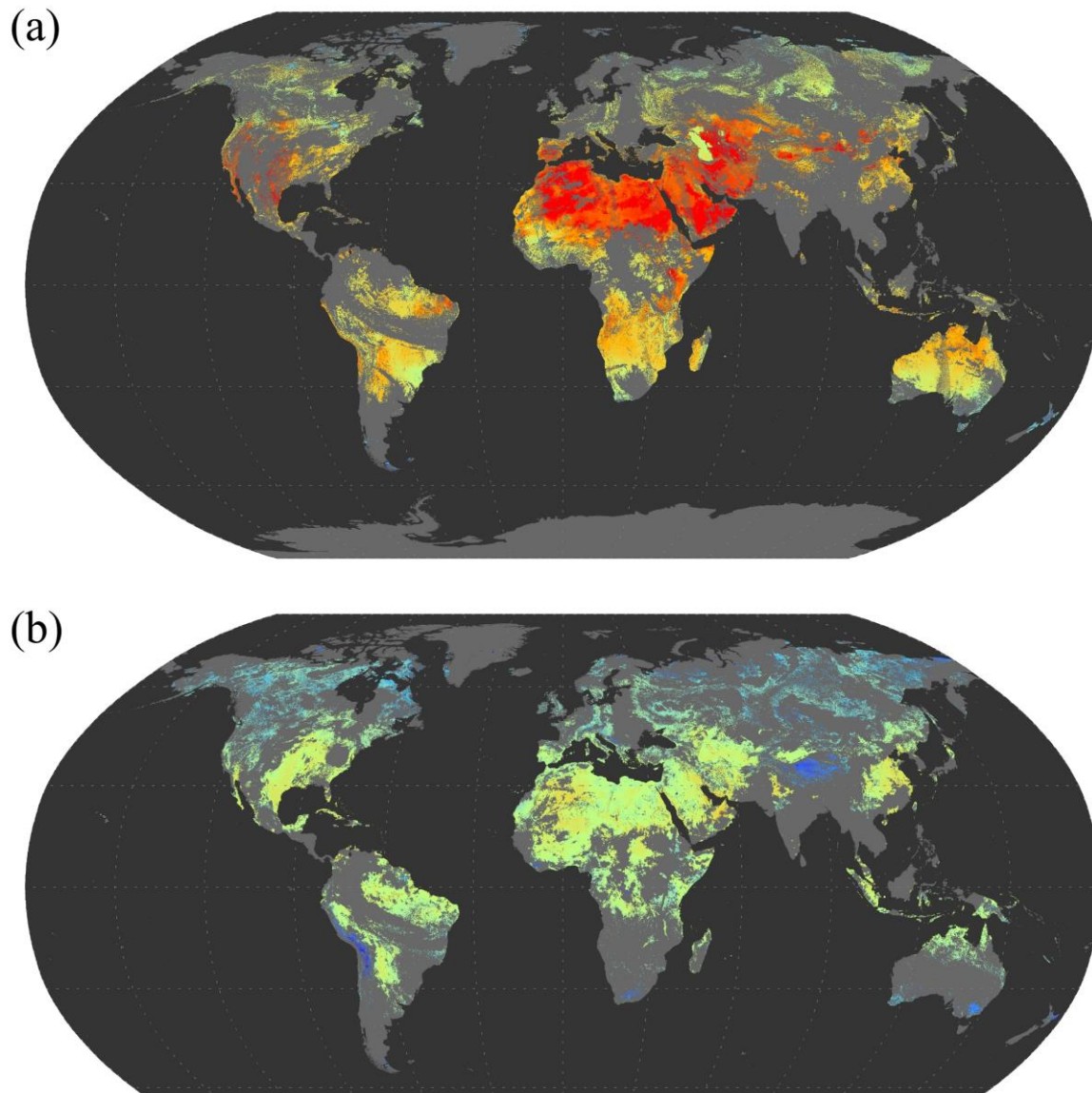


**Figure 11: GT-LST data for July 27, 1997: (a) daytime; (b) nighttime.**


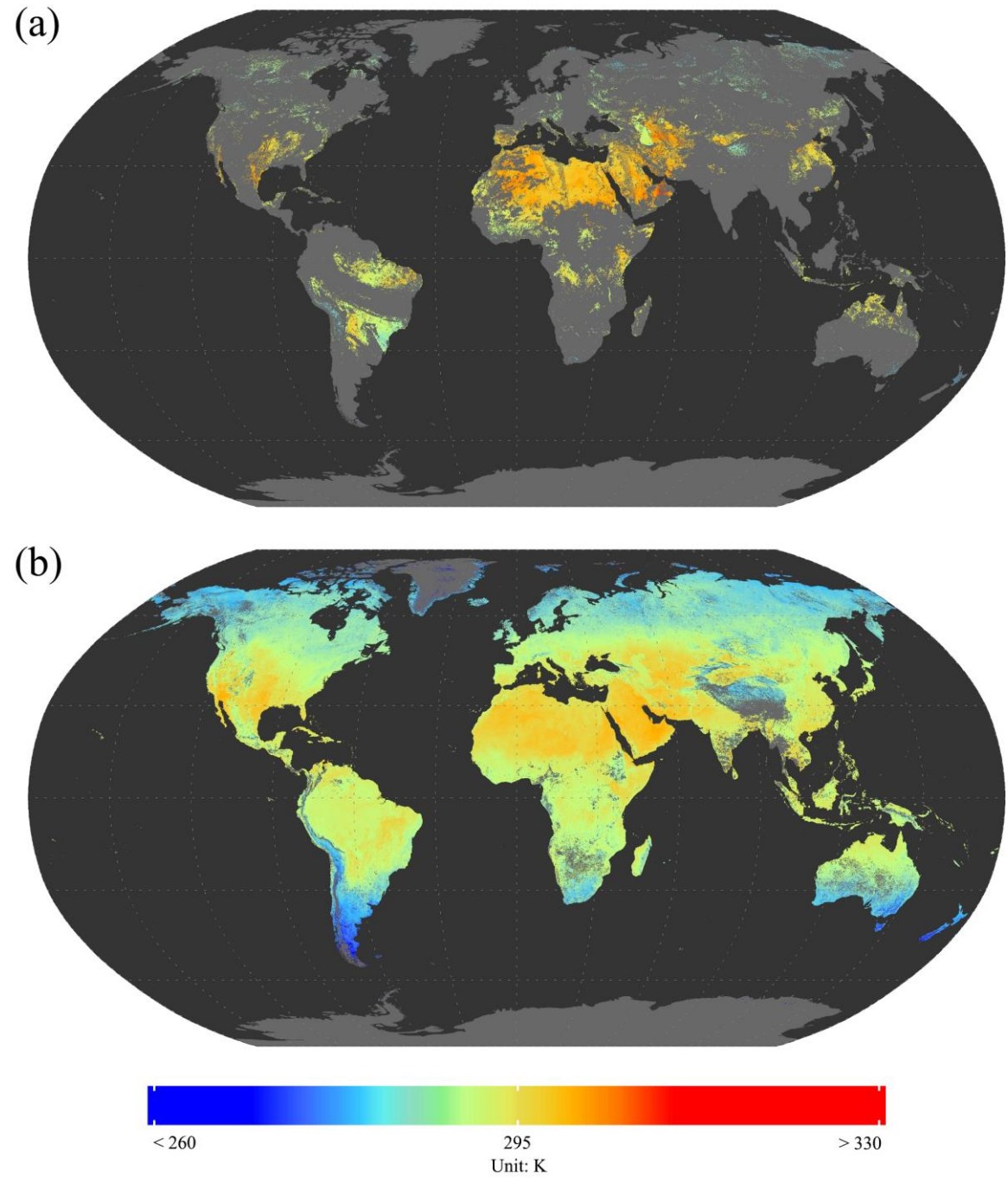


**Figure 12: Daily and monthly mean LST based on GT-LST: (a) daily mean LST for July 27, 1997; (b) monthly mean LST in August 2000.**

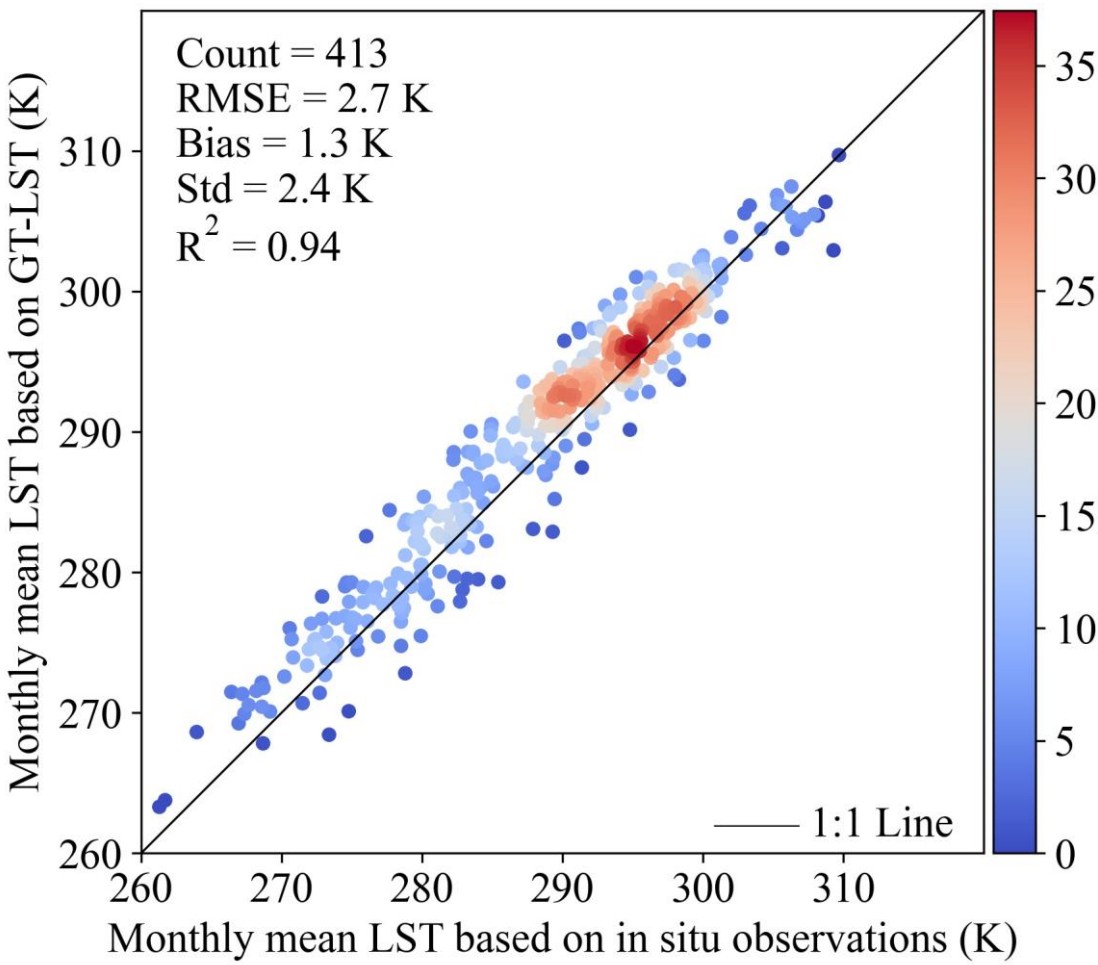

**Figure 13: Monthly mean LST based on GT-LST versus monthly mean LST based on in situ LST from 1994 to 2005.**

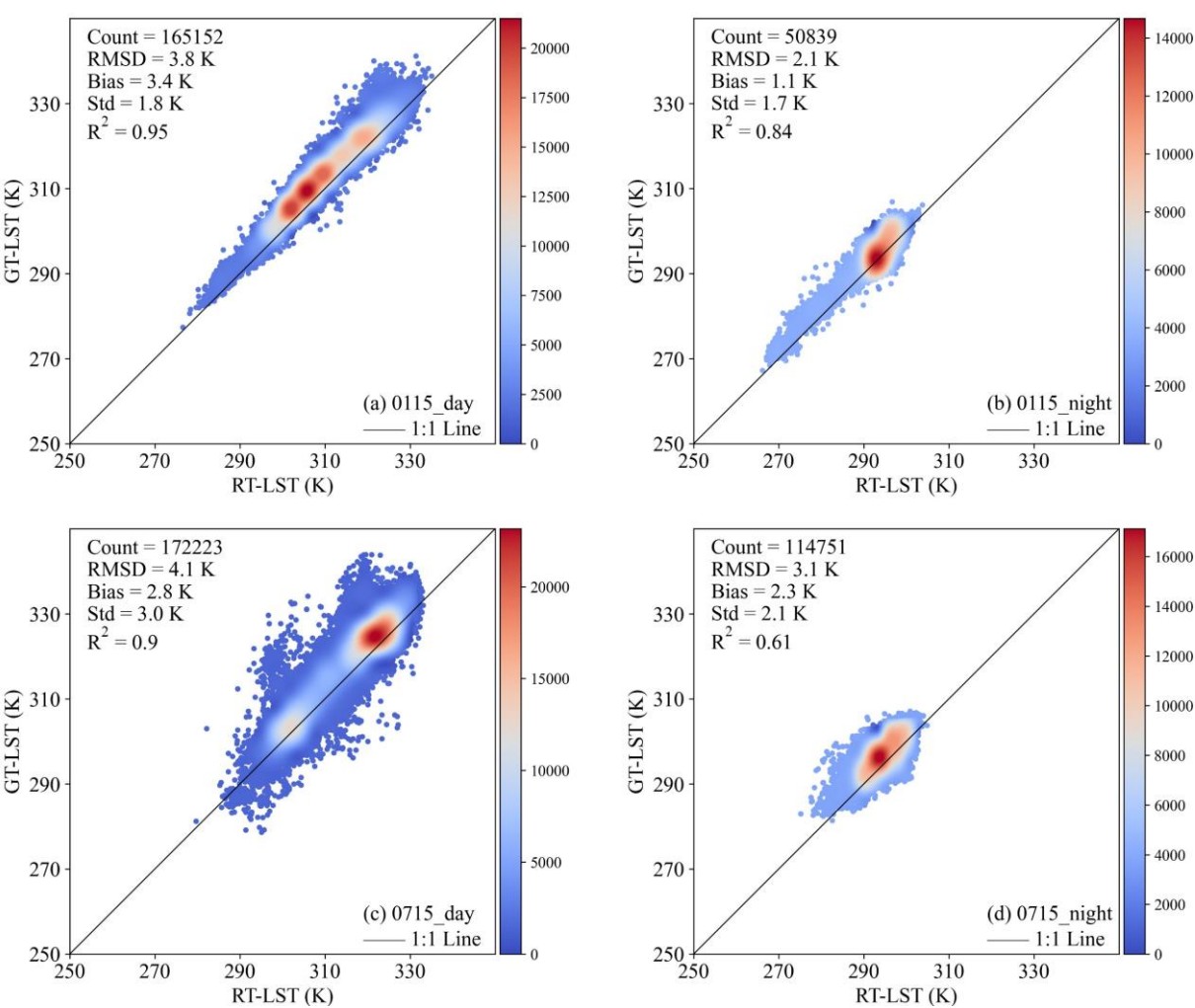

**Figure 14: GT-LST versus RT-LST during daytime and nighttime on January 15 and July 15, 1997: (a) daytime of January 15, 1997; (b) nighttime of January 15, 1997; (c) daytime of July 15, 1997; (d) nighttime of July 15, 1997.**

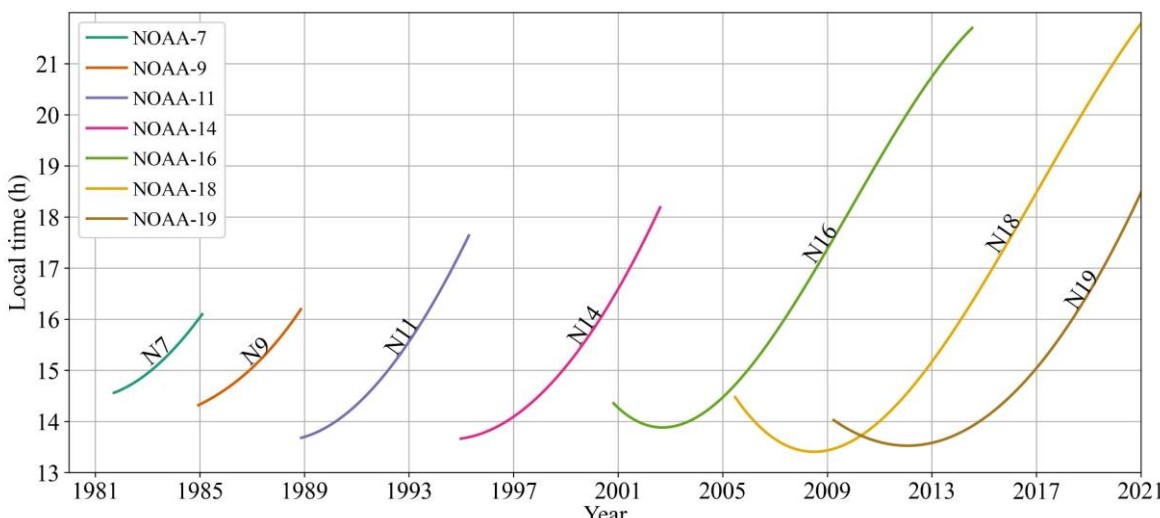

**Figure 15: Equatorial crossing time of NOAA afternoon satellites (Adapted from https://www.star.nesdis.noaa.gov/smcd/emb/vci/VH/vh_avhrr_ect.php).**