# Peer review of "A global historical twice-daily (daytime and nighttime) land surface temperature dataset produced by AVHRR observations from 1981 to 2021"

_Earth System Science Data, 2022_

## Author Comment (AC1)

Response to Reviewer #1

We appreciate a lot for your efforts in providing detailed comments and recommendation. They are very helpful to improve the quality of the manuscript. We have revised the manuscript according to your comments. The comments from the reviewers are kept in regular font, our responses use blue highlighting, and the revised sentences or words in the revised manuscript are highlighted with red color.

This manuscript proposes a long-term (1981-2005) AVHRR land surface temperature (LST) dataset that includes outcomes at both daytime and nighttime. The algorithm is the generalized split-window (GSW) algorithm while in the production, this dataset also considered annual land cover change. Overall, the accuracy of the proposed dataset is promising, and it filled the gaps regarding long-term global LST datasets, especially at nighttime. Therefore, I would recommend it be published on ESSD after a major revision.

**Q1.** Positive bias issue. Based on site validation and inter-comparison with MYD11 and the other two AVHRR LST products, the proposed GT-LST shows a clear positive bias (>1 K) nearly in all results. The authors claim the bias is due to the emissivity difference (Line 370), however, the proposed GT-LST has a clear bias than the other three products, and it seems that the emissivity used by GT-LST is not accurate. The authors mention that the dataset will be calibrated to remove the bias in the future (Line 436). I am thinking if it would be better to solve this issue in this paper as it doesn't need to be done in a separate paper.

**Response**: Thanks a lot for your valuable comments. First, we would like to make some explanations on the positive bias issue as follows:

(1) The GT-LST product and the global daytime AVHRR LST (GD-LST) used a dynamic emissivity method to retrieve LST. We compared GT-LST with GD-LST on January 15, April 15, July 15, and October 16, 1999, with low positive bias of 0.6 K (Figure R1).

(2) The MYD11A1 LST product and the regional twice-daily LST product over

Africa (RT-LST) are generated by the spilt-window (SW) algorithm. Land surface emissivities of these two products are assigned according to classification-based method that produces emissivities with fixed values for a limited number of land cover types. This method works well over densely vegetated areas and water where emissivities are relatively stable. However, cold biases of 3-5 K are often found over semi-arid and arid regions because these regions have much higher emissivity variability, and only one fixed overestimated emissivity inferred from land cover types is assigned to these regions (Coll et al., 2009; Hulley and Hook 2009; Wan et al., 2002). In order to represent the natural variation in emissivity, we used an improved NDVI threshold method to dynamically retrieve daily emissivity. Based on the analysis above, emissivity derived from dynamic methods is lower than emissivity according to classification-based method, which makes the proposed GT-LST is higher than MYD11A1 LST and RT-LST (i.e., positive bias). We note that earlier researches on this issue had similar results. Reiners et al. (2022) compared AVHRR LST product of the TIMELINE project with MYD11_L2 LST product from 2003 to 2014, the result shows that the TIMELINE dynamic emissivity is lower than the MYD11_L2 fixed emissivity and a general positive bias (i.e., bias=2.2 K) of TIMELINE LST towards MYD11_L2 LST. Martins et al. (2019) compared MSG LST and GOES-16 LST and revealed that a positive bias (MSG > GOES) of around 1.6 K persists due to the overestimation of the fixed emissivity of GOES. Mao et al. (2007) analyzed the retrieval result by radiative transfer model with neural network algorithm and MODIS product algorithm, indicating that MOD11_L2 LST product overestimates the emissivity, resulting in an underestimation of LST.

(3) To further illustrate the positive bias issue, we present an intercomparison exercise between MxD11A1 LST products (Terra and Aqua/MODIS using SW algorithm, Collection 6) with fixed emissivity and MSG LST products (MSG/SEVIRI using SW algorithm) with dynamic emissivity for 4 days (January 15, April, 15, July 15, and October 15, 2020). The criteria in Sec 3.2 were used to guarantee the reliability of the intercomparison results. The result is shown in Figure R2, indicating that a general positive bias (daytime ranges from 0.7 K to 3.3 K, nighttime ranges from 0.2 K

to 1.4 K) of MSG LST towards MxD11A1 LST for each land cover types.

(4) The comparison with in situ LST showed that a positive bias was found for all SURFRAD sites. However, only the bias of BND and FPK are large than 1 K. Similar results were obtained by Reiners et al. (2022) and Liu et al. (2019).

Therefore, we think that positive biases obtained for GT-LST and other LST products are relatively reasonable.

Next, many LST products can provide global twice-daily LST after 2000, such as ASTER LST, MODIS LST, VIIRS LST, AATSR LST and SLSTR LST. Users can obtain a relatively long-term twice-daily LST product by combining GT-LST with these LST products. However, integration of LST from different sensors is complicated. Due to the different LST inversion methods, air conditions, viewing geometries, etc., the sensors bias between GT-LST and other LST products is not constant. Therefore, developing a general method to utilize for sensor normalization is difficult and is not the key point of this paper.

[Figure]

Figure R1. GT-LST versus GD-LST during the daytime on January 15, April, 15, July 15, and October 16, 1999.

a

[Figure]

b

[Figure]

Figure R2. The bias (SEVIRI LST minus MxD11A1 LST) and RMSD between the MxD11A1 product and SEVIRI during daytime (a) and nighttime (b) over various land cover types.

**Q2.** Large RMSE (4.1 K) of the monthly mean LST result. The GT-LST is claimed to have the strength to generate gap-free monthly mean LST; however, the outcome has an RMSE of 4.1 K which is too large at a monthly scale compared to other studies (Line 395). This part weakened the statement of the advantage of GT-LST for temporal upscaling based on the logic chain. I would suggest either removing this part or quantifying the impact of orbit drift, in other words, comparing the accuracies of samples that have not and have suffered from orbit drift, and then claiming the potential of this data after orbit drift.

**Response**: Thanks for your suggestion. We used simple linear combinations of monthly mean daytime and nighttime LST values to estimate MMLST. The detailed revisions are listed as follows.

"…*To estimate MMLST, first obtain the mean instantaneous clear-sky LST at daytime and nighttime, and then use these mean values to estimate MMLST according to the* simple linear regression method (see Appendix B). *In order to validate the accuracy of MMLST results, we compared MMLST based on GT-LST with that of in situ LST observations from SURFRAD sites for* 1995–2005. *All in situ LST measurements are all-sky and complete on a certain month, which means that the in situ MMLST is true MMLST. Fig. 15 showed that MMLST derived from GT-LST are related to the true MMLST, with an R$^2$ value of* 0.94 *and an RMSE value of* 2.7 K. *This result is* similar to *that of Chen et al. (2017), who compared MMLST from MODIS day and night instantaneous clear-sky LST with actual MMLST from 156 flux tower stations, and reported RMSE bias values of approximately 2.7 K.*"

We have redrawn Fig. 15 according to the simple linear regression method. For your convenience, we listed it below.

[Figure]

*Figure 15. Monthly mean LST based on GT-LST versus monthly mean LST based on in situ LST from 1995 to 2005.*

We have added the following descriptions in Appendix B.

*"Impacting of the NOAA satellite orbital drift, daytime and nighttime observations of NOAA afternoon satellites cannot represent maximum and minimum temperatures well. Therefore, the MMLST according to the simple average method has a significantly lower accuracy than other studies (Figure A3). Xing et al. (2021) proposed to use 9 combinations of two to four MODIS instantaneous retrievals of which at least one daytime LST and one nighttime LST to estimate mean LSTs, and determined the weight for every moment. Inspired by the work of Xing et al. (2021), we determined to use simple linear combinations of monthly mean daytime and nighttime LST values that were observed at observation times for NOAA to estimate MMLST with ground-based measurement. For the combinations of two valid monthly mean LSTs (one daytime and*

*one nighttime LST), the regression models can be written as follows*:

$$MMLST = a_1 * MMLST_{day} + a_2 * MMLST_{night} + b \qquad (B1)$$

*where MMLST is the ground-based monthly mean LST, $a_1$, $a_2$ and $b$ are the fitting coefficients, $MMLST_{day}$ is the monthly mean in situ LST at the NOAA daytime observation, $MMLST_{night}$ is the monthly mean in situ LST at the NOAA nighttime observation.*

*Taking into account the observed times of NOAA satellites with orbital drift effect since 1981, combinations of two observations from these satellites contain eight cases: 13:30–17:00/01:30–05:00 local solar time in 0.5-hour interval. Based on the in situ LST measurements during the period 2003 to 2018 at 227 flux stations operating in globally diverse regions, we obtained the fitting coefficients (Table A1). Then, we calculated the MMLST of GT-LST using GT-LST monthly mean daytime and nighttime LSTs, Eq. (B1), and the fitting coefficients listed in Table A1.*"

Table A1. Statistics for the relationship between the regressions of the eight combinations and actual monthly mean LST.

| Case | Time | $a_1$ | $a_2$ | b | RMSE | $R^2$ | Number |
|---|---|---|---|---|---|---|---|
| 1 | 13:30/01:30 | 0.3844 | 0.5783 | 10.3446 | 2.0 | 0.97 | 12095 |
| 2 | 14:00/02:00 | 0.4010 | 0.5621 | 10.2042 | 1.9 | 0.98 | 12241 |
| 3 | 14:30/02:30 | 0.4235 | 0.5451 | 8.6172 | 1.9 | 0.98 | 12381 |
| 4 | 15:00/03:00 | 0.4490 | 0.5211 | 8.2652 | 1.8 | 0.98 | 12303 |
| 5 | 15:30/03:30 | 0.4816 | 0.4840 | 9.5710 | 1.8 | 0.98 | 12165 |
| 6 | 16:00/04:00 | 0.5250 | 0.4349 | 11.2284 | 2.0 | 0.97 | 11818 |
| 7 | 16:30/04:30 | 0.5663 | 0.3884 | 12.8572 | 2.2 | 0.96 | 10992 |
| 8 | 17:00/05:00 | 0.6040 | 0.3621 | 9.7302 | 2.4 | 0.96 | 9765 |

**Q3.** The impact of annual land cover change. This is an interesting part of the study, whereas the study didn't pay attention to the performance of such change. Traditionally people mainly utilized a land cover climatology map rather than annual changes to retrieve global LST. I would suggest including additional analysis to find some examples and compare with LST from Ma et al. (2020) to demonstrate the progress using annual land cover maps.

**Response**: This is a good suggestion! Changes in land cover have been accelerating since 1980 under the impact of climate changes and human activities. As an intrinsic property of natural materials, land surface emissivity predominantly depends on the land cover type. Therefore, using only one year of land cover data to determine longterm emissivity is not accurate. The quantitative relationship between annual land cover change and LST is rather complex because the changes of Land surface temperature were related to many factors, including changes in land cover, land surface parameters, seasonal variation, climatic condition and economic development, etc. Furthermore, GT-LST and LST from Ma et al. (2020) used different LST retrieval algorithms and data sources, which makes it harder to analyze the impact of annual land cover change between these two LST products. However, this is a meaningful research topic, and we will further analyze the impact in future work.

**Q4.** Some processes were not introduced clearly.

**Q4.1** why does not GT-LST cover 1981 to 2022? GAC raw data is still updating.

**Response**: Thanks a lot for your comment. The reasons that GT-LST only cover 1981 to 2005 are as follows:

Existing satellite-based global twice-daily LST products can only date back to 2000. Therefore, when the study began, we aimed to fill the data gap of global satellite-derived twice-daily LST before 2000. Considering global meteorology and climatology-related applications urgently need more than 30 years of daily LST products, GT-LST can be combined with the existing satellite-derived daily LST product (e.g., MODIS LST, AATSR LST and ASTER LST) after 2000 to satisfy that requirement. However, integration of LST from different sensors need to eliminate or limit the bias between the sensors. We then extend the time span of GT-LST to 2005. Benefiting from the same observation period with other LST products, these extended data can be used to calibrate the bias between GT-LST and other LST datasets. In this way, users can obtain a relatively homogeneous twice-daily LST product for 1981 to 2022. However, we will apply your suggestion to extend the time span of GT-LST to 2022 in the near future.

**Q4.2** why did the authors only employ the site observations from 1995 to 2000? If you can extend it to 2005, you can include one more SURFRAD site.

**Response:** Thank you for your suggestion. We have extended the observations of SURFRAD sites to 2005 and employed one more SURFRAD site (i.e., SXF) observations according to your suggestion. We have redrawn Fig. 8. For your convenience, we listed it below.

[Figure]

*Figure 8. GT-LST versus in situ LST for 1995–2005 at (a) BND, (b) DRA, (c) FPK, (d) GWN, (e) PSU, (f) SXF, and (g) TBL sites.*

**Q4.3** Regarding the site validation, 6 sites seem not enough to represent the accuracy of the global product. I would recommend adding some BSRN sites that also have good data quality.

**Response**: Thanks for your suggestion. Following your comments, we have added some BSRN sites to represent the accuracy of the GT-LST product in contrasting climatic zones. The following contents have been added in Section 2.5 and Section 4.1, respectively.

*"…The BSRN has 76 stations that detect important changes in the Earth's radiation field at the Earth's surface since 1992. These stations provide high-quality surface and upper-air meteorological observations, which are important in supporting the validation and confirmation of satellite. We selected four sites with measurements of upwelling and downwelling TIR radiances before 2000 (Table 3)."*

*"…We further compared GT-LST data with in situ LST data at BAR, NYA, PYA, and TAT sites for 1995–2005. Fig. 9 shows the scatterplots between GT-LST and in situ LST at these four BSRN sites. The accuracy of GT-LST product at BSRN sites is relatively worse than that at SURFRAD sites, with RMSE (bias) ranges from 3.1 K (-2.7 K) to 4.0 K (2.5 K)."*

*Table 3. Details of the validation sites used in this study.*

| | Name | Elevation(m) | Land cover type | Latitude | Longitude | Valid period |
|---|---|---|---|---|---|---|
| SURFRAD | BND | 230 | Croplands | 40.0519 | -88.3731 | 1995–2005 |
| | DRA | 1007 | Open shrublands | 36.6237 | -116.0195 | 1998–2005 |
| | FPK | 634 | Grasslands | 48.3078 | -105.1017 | 1994–2005 |
| | GWN | 98 | Cropland/natural vegetation mosaic | 34.2547 | -89.8729 | 1994–2005 |
| | PSU | 376 | Cropland/natural vegetation mosaic | 40.7201 | -77.9309 | 1998–2005 |
| | TBL | 1689 | Grasslands | 40.1250 | -105.2368 | 1995–2005 |
| | SXF | 473 | Croplands | 43.7343 | -96.6233 | 2003–2005 |
| BSRN | BAR | 8 | Tundra | 71.3230 | -156.6070 | 1995–2005 |
| | NYA | 11 | Tundra | 78.9227 | 11.9273 | 1999–2005 |
| | PAY | 491 | Cultivated | 46.8123 | 6.9422 | 1995–2005 |
| | TAT | 25 | Grass | 36.0581 | 140.1258 | 1996–2005 |

[Figure]

*Figure 9. Scatterplots between GT-LST and in situ LST at (a) BAR, (b) NYA, (c) PYA, and (d) TAT.*

**Q4.4** why Fig 9(b) has some considerable scattered samples? Those cases should be discussed in the context.

**Response:** Thanks for your comment. We have added some discussion in Section 4.2 for the revised manuscript as follows:

*"...However, as can be seen in Fig.10(b), large LST differences (GT-LST - MYD11A1 LST) more than 20 K are mostly distributed in red box. Through counting, there are 111 samples in red box, which are barren land cover type and arid climate type. Fig. A2 shows the distribution of each scattered samples in red box. 77 of 111 samples happened in Haiya, Sudan on March 31, 2004. The samples of rest happened in Taif, Saudi Arabia on April 2, 2004. For these samples, we double-checked all*

*variables that are essential parameters in GT-LST retrieval. The result show that all scope variables are reasonable except BT of TIR bands. Abnormal high BTs at these nighttime samples were found on March 31 and April 2, 2004 (Fig. A3), which leaded to extreme high LSTs. The possible reasons for abnormal high BTs are as follows: (1) These two regions may have experienced extreme events such as wars and natural disasters on March 31 or April 2, 2004. But we didn't find relevant information from historical news and documents. (2) Another factor may be instrument failure on these two days."*

[Figure]

*Figure 10. Inter-comparison of GT-LST and MYD11A1 LST in 2004: (a) daytime; (b) nighttime. Red box indicates considerable scattered samples.*

[Figure]

[Figure]

*Figure A1. Distribution of the 111 scattered samples.*

[Figure]

*Figure A2. An example of abnormal high BTs on (a) March 31, 2004 and (b) April 2, 2004.*

**Q4.5** Line 350: as MODIS has been spatially aggregated to match with GT-LST, why spatial heterogeneity is still an issue here?

**Response:** Thanks for your comment. We have deleted this erroneous expression.

**Q4.6** Fig10: I would suggest changing Fig10 to another format: consider RMSE and bias as the two dimensions of the plot, and mark each dot by their names as using color to show the bias is not easily quantified.

**Response:** Thanks for your valuable suggestion. We have redrawn Fig. 11 according to your suggestion. For your convenience, we listed it below.

[Figure]

*Figure 11. RMSD and bias between GT-LST and MYD11A1 LST in 2003 for various land cover types. ENF: evergreen needleleaf forests, EBF: evergreen broadleaf forests, DNF: deciduous needleleaf forests, DBF: deciduous broadleaf forests, MXF: mixed forests, CSR: closed shrublands, OSR: open shrublands, WDS: woody savannas, SVN: savannas, GRS: grasslands, PMW: permanent wetlands, CRP: croplands, UBL: urban and built-up lands, CNV: cropland/natural vegetation mosaics, PSI: permanent snow and ice, BRN: barren, WTB: water bodies and ALL: all land cover types.*

**Q4.7** Line 357: why do savannas and cropland show considerable bias?

**Response:** Fig. R3 shows relatively large disparities between GT-LST and MYD11A1 LST over savannas (i.e., woody savannas and savannas) and croplands (i.e., cropland/natural vegetation mosaics and croplands) for the intercomparison. We would like to make some explanations on large disparities between these two products as follows:

According to NDVI threshold method, the daily emissivity of an AVHRR pixel can be derived using the following formula:

$$\varepsilon = \varepsilon_{veg} * FVC + \varepsilon_{soil} * (1 - FVC)$$

Here, $\varepsilon$ is the emissivity, $\varepsilon_{veg}$ is the vegetation emissivity, $\varepsilon_{soil}$ is the bare soil emissivity, and $FVC$ is the fraction of vegetation cover.

For a vegetation pixel, its FVC is less than 1 due to the influence of natural and

human factors, which leads to the underestimation of emissivity comparing with fixed emissivity, resulting in an overestimation of LST. The situation is particularly evident over croplands and savannas. Specially, natural disasters (e.g., drought and pests) and agricultural activities (e.g., harvest, cropland lies fallow) can significantly decrease cropland density and result in higher exposure of the soil. It leads to a decrease in cropland emissivity, resulting in an overestimation of LST. The emissivity for savannas decreases because of the increasing proportion of soil by grazing, fire and annually a long period in which moisture inadequate, resulting in an overestimation of LST.

[Figure]

Figure R3. Scatterplots of GT-LST versus MYD11A1 LST during 2004 over WDS (a), SVN (b), CRP (c), and CNV (d). WDS: woody savannas, SVN: savannas, CRP: croplands, and CNV: cropland/natural vegetation mosaics.

Minor:

1. Line 35: Some of them used surface air temperature rather than LST to detect climate change and it should be not mixed.

**Response:** Thank you for your careful reading. We have removed the reference (i.e., Keenan and Riley, 2018) in the revised manuscript.

2. Line 71: remove 'the'

**Response:** Corrected as suggested.

3. Line 94: polar-orbiting

**Response:** Corrected as suggested.

4. line 101: the first

**Response:** Corrected as suggested.

5. Line 179: Especially

**Response:** Corrected as suggested.

6. Line 298: identifier

**Response:** Corrected as suggested.

7. Line 301: difference

**Response:** Corrected as suggested.

8. Line 317: due to -> because

**Response:** Corrected as suggested.

9. Line 327: RMSEs

**Response:** Corrected as suggested.

10. Line 403: remove 'in'

**Response:** Corrected as suggested.

11. Line 404: 'due to' should be followed by a noun rather than a sentence, suggest revising the whole manuscript for this issue.

**Response:** Thank you for your careful reading. Following your suggestion, we have checked the whole manuscript and corrected this issue.

12. Line 411: considers

**Response:** Corrected as suggested.

13. Line 446: open-source

**Response:** Corrected as suggested.

14. Line 451: cloud mask

**Response:** Corrected as suggested.

**References for the above responses are listed below:**

Chen, X., Su, Z., Ma, Y., Cleverly, J., Liddell, M.: An accurate estimate of monthly mean land surface temperatures from MODIS clear-sky retrievals, J. Hydrometeorol., 18, 2827-2847, doi:10.1175/JHM-D-17-0009.1,2017.

Coll, C., Wan, Z., Galve, J. M.: Temperature-based and radiance-based validations of the V5 MODIS land surface temperature product, J Geophys. Res-Atmos., 114, 1-15, doi:10.1029/2009jd012038, 2009.

Hulley, G. C., Hook, S. J.: The North American ASTER land surface emissivity database (NAALSED) version 2.0, Remote Sens. Environ, 113, 1967-1975, doi:10.1016/j.rse.2009.05.005, 2009.

Liu, X., Tang, B.-H., Yan, G., Li, Z.-L., Liang, S.: Retrieval of Global Orbit Drift Corrected Land Surface Temperature from Long-term AVHRR Data, Remote Sens., 11, 2843, doi:10.3390/rs11232843, 2019.

Mao, K., Shi, J., Li, Z. L., Tang, H.: An RM-NN algorithm for retrieving land surface temperature and emissivity from EOS/MODIS data, J Geophys. Res-Atmos., 112, 1-17, doi: 10.1029/2007JD008428, 2009.

Martins, J. P., Coelho e Freitas, S., Trigo, I. F., Barroso, C., Macedo, J.: Copernicus Global Land Operations-Lot I "Vegetation and Energy" Algorithm Theoretical Basis Document. Land Surface Temperature—LST, 1, 2019.

Reiners, P., Asam, S., Frey, C., Holzwarth, S., Bachmann, M., Sobrino, J., Göttsche, F., Bendix, J. Kuenzer, C.: Validation of AVHRR Land Surface Temperature with MODIS and In Situ LST—A TIMELINE Thematic Processor, Remote Sens., 13, 3473, doi:10.3390/rs13173473, 2021.

Xing, Z., Li, Z. L., Duan, S. B., Liu, X., Zheng, X., Leng, P., Gao, M., Zhang, X., Shang, G.: Estimation of daily mean land surface temperature at global scale using pairs of daytime and nighttime MODIS instantaneous observations, ISPRS J. Photogramm., 178, 51-67, doi:10.1016/j.isprsjprs.2021.05.017, 2021.

Wan, Z., Zhang, Y., Zhang, Q., Li, Z. L.: Validation of the land-surface temperature products retrieved from Terra Moderate Resolution Imaging Spectroradiometer data, Remote Sens. Environ, 83, 163-180, doi:10.1016/S0034-4257(02)00093-7, 2002.

---

## Author Comment (AC2)

Response to second comments of reviewer #1

We appreciate a lot for your efforts in providing detailed comments and recommendation. They are very helpful to improve the quality of the manuscript. We have revised the manuscript according to your comments. The comments from the reviewers are kept in regular font, our responses use blue highlighting, and the revised sentences or words in the revised manuscript are highlighted with red color.

Thank you for your response which resolved many of my concerns. However, I am still wondering if you have addressed some key issues of the AVHRR GAC data:

Q1. As the archived historical data, the AVHRR GAC raw data have a serious geolocation issue that has been criticized by Wu et al. (2020), especially when the view zenith angle is larger than 40-deg, thus I would suggest the authors deal with this issue or at least quantify the impact. Please double-check previous literature and collect such data issues and give a comprehensive discussion.

**Response**: Thanks for your valuable suggestion. Indeed, Wu et al. (2020) provides a preliminary geolocation assessment for AVHRR GAC data of NOAA-17, MetOp-A, and MetOp-B, which present shifts that stay within the range of 4 km for satellite zenith angles smaller than 40° and can reach 6 km when the satellite zenith angle is larger than 40°. To be more clearly for readers, we have added the detail of the geolocation issue according to the work of Wu et al. (2020) in Line 105-107 as follows:

"*Therefore, AVHRR Level-1b GAC data are generally treated as having a coarse resolution of 4 km at the nadir, and the pixel size increases with the satellite zenith angle (VZA).* Furthermore, as the VZA increases, the geolocation accuracy of the AVHRR GAC scene become lower, particularly when VZAs larger than 40° (Wu et al., 2020)."

In addition, considering the influence of geolocation issue, we used an open-source package, Pygac, to pre-process AVHRR GAC data. Pygac, which is based on ephemeris data, orbit model and time of onboard clock, uses correction of satellite location method, correction of scanline timestamps method, correction of geolocation method to improve the geolocation accuracy of the AVHRR GAC data. After the GAC data are treated

through above methods, we believe that their geolocation accuracy basically meets the demand of global applications at 0.05° spatial resolution. However, if users need high geolocation quality GAC data, we suggested that the GAC data less than 40° should be preferred. We have clarified this point in Line 473-476 with the expression as follows:

"*...A variety of factors such as cloud cover, orbital gaps, and instrument failure are responsible for this limitation. And finally, the geolocation accuracy of GT-LST product basically meets the demand of global applications at 0.05° spatial resolution. However, if users need very high geolocation quality GT-LST data, we suggested that the GT-LST data with VZAs less than 40° should be preferred.*"

Q2. It still doesn't make sense that the data ended in 2005 artificially and the other reviewer also agreed with my suggestion.

**Response**: Thanks for your comment. We are sorry for the previously unclear explanations. As emphasized in the introduction section, this study aims to fill the data gap of global satellite-derived twice-daily LST before 2000. However, considering global meteorology and climatology-related applications urgently need more than 30 years of daily LST products, there are two ways of satisfying that requirement based on GT-LST. One way is to combine GT-LST (1981-2000) with the existing satellite-derived daily LST product (2000-present), which depend on different products with the same observation period to eliminate or limit the bias between different sensors. Therefore, we extend the time span of GT-LST to 2005. Benefiting from the same observation period (i.e., 2000-2005) with MODIS LST, we will produce a global long-term (1981-present) LST data record according to the method of Liu et al. (2012), which will be primarily from the AVHRR (1981-2000) and MODIS (2000-present).

Indeed, as you mentioned, extending the time span of GT-LST to present is another way to address this issue. We have already started working on generating GT-LST products (2006-present). Although we have proposed a framework for generating GT-LST product, we still need spend a lot of time downloading global AVHRR GAC L1B data, handling large amounts of original Level-1B data, generating huge amounts of process variable data, and so on. After all data have been processed, we will upload GT-

LST (2006-present) to previous URL (https://doi.org/10.5281/zenodo.7134158).

Q3. The monthly mean LST still has an overall bias of 1.3 K compared to site observations, please double-check the code or provide a discussion and comparison with previous work.

**Response**: Thanks for your suggestion. According to your suggestion, we have double-checked the code but not found problems. In addition, we have added the discussion of positive bias between monthly mean GT-LST and in situ LST in Line 439-441 as follows:

"*...This result is similar to that of Chen et al. (2017), who compared MMLST from MODIS day and night instantaneous clear-sky LST with actual MMLST from 156 flux tower stations, and reported RMSE value of approximately 2.7 K. However, it should be noted that a positive bias of 1.3 K between GT-LST MMLST and in situ MMLST. One possible reason is that in situ MMLST of some sites does not represent the MMLST over the 0.05°×0.05° pixel.*"

**References for the above responses are listed below**:

Liu, Y., Liu, R., and Chen, J. M.: Retrospective retrieval of long-term consistent global leaf area index (1981–2011) from combined AVHRR and MODIS data, J. Geophys. Res-Biogeo., 117, 1-14, https://doi.org/10.1029/2012JG002084, 2012.

Wu, X., Naegeli, K., and Wunderle, S.: Geometric accuracy assessment of coarse-resolution satellite datasets: a study based on AVHRR GAC data at the sub-pixel level, Earth Syst. Sci. Data, 12, 539–553, https://doi.org/10.5194/essd-12-539-2020, 2020.

---

## Author Comment (AC3)

**Response to Reviewer #2**

We appreciate a lot for your efforts in providing detailed comments and recommendation. They are very helpful to improve the quality of the manuscript. We have revised the manuscript according to your comments. The comments from the reviewers are kept in regular font, our responses use blue highlighting, and the revised sentences or words in the revised manuscript are highlighted with red color.

In the study titled "A global historical twice-daily (daytime and nighttime) land surface temperature dataset produced by AVHRR observations from 1981 to 2005", the authors produce a global LST product from 1981 to 2005 at 0.05 degree using AVHRR observations. The study is potentially useful for understanding changes in surface climate over a longer time period than what we can currently examine using most existing LST products. However, I have several concerns that should be addressed before the paper is considered for publication.

Q1. The biggest issue I have is that the dataset is restricted to 2005. Given that AVHRR products have large biases compared to MODIS Aqua and use different inputs (such as the dynamic emissivity estimates used), one cannot combine MODIS and AVHRR to perform long-term analysis. Since the AVHRR is still operational, the dataset needs to be extended to more recent years.

**Response**: Thanks for your valuable suggestion. As emphasized in the introduction section, this study aims to fill the data gap of global satellite-derived twice-daily LST before 2000. However, considering global meteorology and climatology-related applications urgently need more than 30 years of daily LST products, there are two ways of satisfying that requirement based on GT-LST. One way is to combine GT-LST (1981-2000) with the existing satellite-derived daily LST product (2000-present), which depend on different products with the same observation period to eliminate or limit the bias between different sensors. Therefore, we extend the time span of GT-LST to 2005. Benefiting from the same observation period (i.e., 2000-2005) with MODIS LST, we will produce a global long-term (1981-present) LST data record according to

the method of Liu et al. (2012), which will be primarily from the AVHRR (1981-2000) and MODIS (2000-present).

Indeed, as you mentioned, extending the time span of GT-LST to present is another way to address this issue. We have already started working on generating GT-LST products (2006-present). Although we have proposed a framework for generating GT-LST product, we still need spend a lot of time downloading global AVHRR GAC L1B data, handling large amounts of original Level-1b data, generating huge amounts of process variable data, and so on. After all data have been processed, we will upload GT-LST (2006-present) to previous URL (https://doi.org/10.5281/zenodo.7134158).

Q2. As an addendum to the previous point, since one of the most important use cases of long-term datasets is time series analysis, the long-term changes in GT-LST should be compared against equivalent changes from MODIS products. If the orbital drift has a significant impact on long-term trends, we should be very cautious about the suitability of this data product for this use case. This issue needs to be quantified more clearly instead of just discussed in text in one section. This can potentially avoid misleading results from future uses of this dataset.

**Response**: Thanks a lot for your comments. Indeed, one of the intentions of GT-LST is providing effective supplementary data for global long-term time series analysis. The analysis requires daily, monthly or annual mean LST (i.e., DMLST, MMLST, and AMLST) more than instantaneous LST as these mean LSTs are key indicators when monitoring global LSTs over a long time series (Li et al., 2023; Liu et al., 2023; Xing et al., 2021). It is possible to derive an estimate of the global accurate DMLST, MMLST and AMLST based on twice-daily LST product. However, impacting of the NOAA satellite orbital drift, daytime and nighttime observations of NOAA afternoon satellites cannot represent maximum and minimum temperatures well. Therefore, calculating the daily and monthly mean LST by averaging daytime and nighttime LSTs derived from GT-LST has a significantly lower accuracy than other studies (Figure A4). Inspired by the work of Xing et al. (2021), we use simple linear combinations of daytime and nighttime LST values that were observed at observation times for NOAA to estimate

DMLST and MMLST. In order to validate the accuracy of DMLST and MMLST according to the simple linear regression method, we compared DMLST and MMLST derived from GT-LST with that of in situ LST observations from SURFRAD sites, and reported RMSE values of approximately 2.4 K and 2.7 K, respectively. These results are similar to that of Xing et al. (2021) and Chen et al. (2017). In this way, we still obtain accurate DMLST and MMLST without satellite orbit drift correction. Then, we rephrase the paragraph in Line 429-436 as follows:

"…*To estimate MMLST, first obtain the mean instantaneous clear-sky LST at daytime and nighttime, and then use these mean values to estimate MMLST according to the simple linear regression method (see Appendix B). In order to validate the accuracy of MMLST results, we compared MMLST based on GT-LST with that of in situ LST observations from SURFRAD sites for 1994–2005. All in situ LST measurements are all-sky and complete on a certain month, which means that the in situ MMLST is true MMLST. Fig. 13 showed that MMLST derived from GT-LST are related to the true MMLST, with an $R^2$ value of 0.94 and an RMSE value of 2.7 K. This result is similar to that of Chen et al. (2017), who compared MMLST from MODIS day and night instantaneous clear-sky LST with actual MMLST from 156 flux tower stations, and reported RMSE bias values of approximately 2.7 K.*"

We have redrawn Fig. 13 according to the simple linear regression method. For your convenience, we listed it below.

[Figure]

*Figure 13: Monthly mean LST based on GT-LST versus monthly mean LST based on in situ LST from 1994 to 2005.*

In addition, as for some details of the simple linear regression method, we have added the following descriptions in Appendix B.

"*Impacting of the NOAA satellite orbital drift, daytime and nighttime observations of NOAA afternoon satellites cannot represent maximum and minimum temperatures well. Therefore, the MMLST according to the simple average method has a significantly lower accuracy than other studies (Fig. A4). Xing et al. (2021) proposed to use 9 combinations of two to four MODIS instantaneous retrievals of which at least one daytime LST and one nighttime LST to estimate mean LSTs, and determined the weight for every moment. Inspired by the work of Xing et al. (2021), we determined to use simple linear combinations of monthly mean daytime and nighttime LST values that*

*were observed at observation times for NOAA to estimate MMLST with ground-based measurement. For the combinations of two valid monthly mean LSTs (one daytime and one nighttime LST), the regression models can be written as follows:*

$$MMLST = a_1 * MMLST_{day} + a_2 * MMLST_{night} + b \qquad \text{(B1)}$$

*where MMLST is the ground-based monthly mean LST, $a_1$, $a_2$ and $b$ are the fitting coefficients, $MMLST_{day}$ is the monthly mean in situ LST at the NOAA daytime observation, $MMLST_{night}$ is the monthly mean in situ LST at the NOAA nighttime observation.*

*Taking into account the observed times of NOAA satellites with orbital drift effect since 1981, combinations of two observations from these satellites contain eight cases: 13:30–17:00/01:30–05:00 local solar time in 0.5-hour interval. Based on the in situ LST measurements during the period 2003 to 2018 at 227 flux stations operating in globally diverse regions, we obtained the fitting coefficients (Table A1). Then, we calculated the MMLST of GT-LST using GT-LST monthly mean daytime and nighttime LSTs, Eq. (B1), and the fitting coefficients listed in Table A1."*

Table A1. Statistics for the relationship between the regressions of the eight combinations and actual monthly mean LST.

| Case | Time | $a_1$ | $a_2$ | $b$ | RMSE | $R^2$ | Number |
|------|------|-------|-------|-----|------|-------|--------|
| 1 | 13:30/01:30 | 0.3844 | 0.5783 | 10.3446 | 2.0 | 0.97 | 12095 |
| 2 | 14:00/02:00 | 0.4010 | 0.5621 | 10.2042 | 1.9 | 0.98 | 12241 |
| 3 | 14:30/02:30 | 0.4235 | 0.5451 | 8.6172 | 1.9 | 0.98 | 12381 |
| 4 | 15:00/03:00 | 0.4490 | 0.5211 | 8.2652 | 1.8 | 0.98 | 12303 |
| 5 | 15:30/03:30 | 0.4816 | 0.4840 | 9.5710 | 1.8 | 0.98 | 12165 |
| 6 | 16:00/04:00 | 0.5250 | 0.4349 | 11.2284 | 2.0 | 0.97 | 11818 |
| 7 | 16:30/04:30 | 0.5663 | 0.3884 | 12.8572 | 2.2 | 0.96 | 10992 |
| 8 | 17:00/05:00 | 0.6040 | 0.3621 | 9.7302 | 2.4 | 0.96 | 9765 |

Q3. A second major source of concern is the dynamic emissivity method used. There are several vegetation-adjusted emissivity methods available, which can give different values, different enough to account for some of the biases seen. Of note, at 0.05 degree, you would start resolving larger urban areas, which is a major use case for satellitederived LST (Voogt & Oke, 2003). Different emissivity methods perform differently over urban surfaces, which impacts this important use case (Chakraborty et al. 2021). Ideally, this issue needs to be tested further using different emissivity methods.

**Response**: This is a good suggestion! As you mentioned, to date, various land surface emissivity (LSE) estimation methods have been proposed with the same goal but different advantages, and limitations, e.g., classification-based method, NDVI threshold method, TES method, and physics-based day/night method and so on (Li et al., 2013). With their advantages and limitations, these methods have different accuracies and are applicable for various sensors and applications. To reflect the performance for the emissivity-retrieved methods and account for the positive bias between GT-LST and MYD11A1 LST, according to your suggestion, we compared GT-LST with MYD21A1 LST that uses the same observations with MYD11A1 but uses a physics-based algorithm to dynamically retrieve both LST and spectral emissivity. The intercomparison results of MYD21A1 LST showed a very lower bias. As for the comparison with MYD21A1, a brief explanation was analyzed below in Q4. There is no denying that it is important and significative to evaluate emissivity methods under different circumstances and for various applications. However, the goal of this study is to develop a global historical twice-daily LST product from 1981 to 2005, where LSE is only one of the key parameters. Therefore, we choose an improved NDVI threshold method to estimate LSEs from space for a global case by taking the sensor characteristics, the required accuracy, as well as computation time into account. Although evaluating emissivity methods under different circumstances is not be discussed in more detail in this study, this is a meaningful research topic. Inspired by your suggestion and the work of Chakraborty et al. (2021), we will evaluate these methods on an identical standard and to give the quality and accuracy on their applications in future work. After evaluating them we may attempt to generate the first estimates of LST at a global scale using AVHRR GAC data by combining all these approaches.

Q4. The comparison with MODIS MYD11 is somewhat difficult because of the

different emissivity method used. The comparison should be done against MODIS MYD21, which uses the same observations, but a temperature-emissivity separation method instead of classification-based prescribed emissivity.

**Response**: Thanks for your valuable comments. According to your comments, we have compared GT-LST with MYD21A1 LST (Aqua/MODIS LST product using the TES algorithm, Collection 6.1). Spatially, this intercomparison was conducted at the global scale. Temporally, it was performed on 4 months in 2004 (January, April, July, and October) which cover different seasons. The results of the daytime and nighttime comparison, in Fig. 9, are as follows: The daytime and nighttime RMSD values of 3.2 K and 2.5 K and that of bias of 0.1 K and 1.3 K. Compared to the result of MYD11A1, the significantly smaller bias was obtained for MYD21A1. The possible reason is attributed to the fact that the MYD21A1 LST uses the same observations with MYD11A1 but uses TES method to dynamically retrieve LSE. The following contents have been added in Line 184-187 and Line 405-412, respectively.

"*In this study, Collection-6.1 MYD11A1 of 2004 was selected for sensor-to-sensor comparison. MYD21A1 LST product, which uses the same observations with MYD11A1 but uses temperature–emissivity separation method to dynamically retrieve LST and emissivity, was also selected to make an intercomparison with GT-LST in this study. This inter-comparison was conducted on 4 months in 2004 (January, April, July, and October) which cover different seasons.*"

"*As a result, the dynamic emissivity of GT-LST is typically lower than that of MYD11A1, which leads to overestimation of the LST (Hulley et al., 2016; Guillevic et al., 2014; Reiners et al., 2021; Ren et al., 2011). To further demonstrate this point, we compared GT-LST with MYD21A1 LST. Fig. 9 shows the daytime and nighttime RMSD values of 3.2 K and 2.5 K and that of bias of 0.1 K and 1.3 K between GT-LST and MYD21A1 LST for 4 months in 2004. Compared to the result of MYD11A1, the significantly smaller bias was obtained for MYD21A1. The possible reason is attributed to the fact that the MYD21A1 LST uses the same observations with MYD11A1 but uses a physics-based method to dynamically retrieve emissivity.*"

[Figure]

Figure 9: Intercomparison of GT-LST and MYD21A1 LST in January, April, July, and October 2004: (a) daytime; (b) nighttime.

Q5. For comparison with SURFAD stations, did the authors check that the emissivity used to generate the LST in the ground observations is same as the LST in the GT-LST product? If they are different, would be good to adjust by the emissivity difference and check if that improves the accuracy.

**Response**: The GT-LST use directional measurements of AVHRR in the atmospheric window, while the SURFRAD stations provide upwelling and downwelling broadband hemispherical TIR radiances using pyrgeometers in the spectral range from 3.5 to 50 μm, from which estimates of LSTs can be derived using Stefan–Boltzmann's law. To retrieve LST using Stefan–Boltzmann's law, the surface broadband emissivity must be known a priori. In this study, these broadband emissivities were estimated from ASTER emissivity product using a spectral-to-broadband linear regression equation according to the work of Duan et al. (2019), as follows: BND(0.968), TBL(0.972), DRA(0.967), FPK(0.973), GCM(0.971), PSU(0.970), and SXF(0.970). According to the study of Liang (2005), the surface broadband emissivity of sites can be obtained from AVHRR LSE in AVHRR LSE for channel centered at 11 and 12 μm via the empirical relationship:
$$\varepsilon = 0.2489 + 0.2386\varepsilon_{11} + 0.4998\varepsilon_{12} \tag{R1}$$

According to Eq. (R1), the surface broadband emissivities are 0.976, 0.975, 0.972, 0.973, 0.973, 0.968 and 0.974 for BND, TBL, DRA, FPK, GCM, PSU, and SXF,

respectively. Different empirical relationships perform an error less than 0.01 in the broadband emissivity. According to the study of Xing et al. (2021), the emissivity changes by 0.01, and the change in in-situ LST will not exceed 0.37 K. Therefore, while this error is not negligible, it does not appear to be a dominant source of uncertainty in the ground-based validation.

Q6. Finally, given the view angle of AVHRR, a broader discussion needs to be added about thermal anisotropy (DUffour et al. 2015). Satellites only provide a 2d directional view of LST, and this is not directly comparable across satellites (Landsat vs MODIS) or against ground observations that have a downward pointing radiometer. This is of particular concern over heterogeneous terrain, such as mixed forests and over cities.

**Response**: Thanks a lot for your valuable comments. we would like to make some explanations on the thermal anisotropy issue as follows:

(1) Previous multi-sensor comparison studies (Guillevic et al., 2012; Trigo et al., 2008) found differences up to 12 K between MODIS and SEVIRI LST due to directional effects. Appropriate matchups significantly reduce the discrepancies induced by directional effects (Guillevic et al., 2014). In this study, to avoid the uncertainties induced by directional effects, a strict criterion of viewing geometry alignment was established to guarantee the reliability of the intercomparison results (Li et al., 2023): the difference in VZA between MYD11A1 and GT-LST is limited to be less than 15°.

(2) LST from satellite and ground measurements may differ according to their measurement methods. AVHRR use directional measurements in the atmospheric window, while ground-based longwave radiation measurements are hemispheric, wider spectrum derivations. If the surface is black body, the two LSTs are the same (Wang et al., 2005, Li et al., 2023). However, most natural objects are not black bodies. AVHRR view zenith angles were considered to be an important factor influencing the results when comparisons were made with in situ measurements. Figure R1 shows the relationship between view angles and bias in instantaneous LST at station pixels. Our result is that a high view angle does not necessarily show a high bias and a low view

angle does not necessarily always show a low bias. This means that view angle should not be a significant source for the bias of GT-LST at a 0.05° pixel size.

[Figure]

Figure R1: The relationship between bias of land surface temperature (GT-LST minus in situ observations) and view zenith angle.

**References for the above responses are listed below**:

Chakraborty, T. C., Lee, X., Ermida, S., and Zhan, W.: On the land emissivity assumption and Landsat-derived surface urban heat islands: A global analysis, Remote Sens. Environ., 265, 112682, https://doi.org/10.1016/j.rse.2021.112682, 2021.

Chen, X., Su, Z., Ma, Y., Cleverly, J., Liddell, M.: An accurate estimate of monthly mean land surface temperatures from MODIS clear-sky retrievals, J. Hydrometeorol., 18, 2827-2847, https://doi.org/10.1175/JHM-D-17-0009.1, 2017.

Duan, S.-B., Li, Z.-L., Li, H., Göttsche, F.-M., Wu, H., Zhao, W., Leng, P., Zhang, X., Coll, C.: Validation of Collection 6 MODIS land surface temperature product using in situ measurements, Remote Sens. Environ. 225, 16–29, https://doi.org/10.1016/j.rse.2019.02.020, 2019.

Guillevic, P. C., Biard, J. C., Hulley, G. C., Privette, J. L., Hook, S. J., Olioso, A., Göttsche F. M., Radocinski, R., Román, M. O., Yu, Y., and Csiszar, I.: Validation of

Land Surface Temperature products derived from the Visible Infrared Imaging Radiometer Suite (VIIRS) using ground-based and heritage satellite measurements, Remote Sens. Environ., 154, 19-37, https://doi.org/10.1016/j.rse.2014.08.013, 2014.

Guillevic, P. C., Privette, J. L., Coudert, B., Palecki, M. A., Demarty, J., Ottlé, C., and Augustine, J. A.: Land Surface Temperature product validation using NOAA's surface climate observation networks—Scaling methodology for the Visible Infrared Imager Radiometer Suite (VIIRS), Remote Sens. Environ., 124, 282-298, https://doi.org/10.1016/j.rse.2012.05.004, 2012.

Li, Z.-L., Wu, H., Duan, S.-B., Zhao, W., Ren, H., Liu, X., Leng, P., Tang R., Ye, X., Zhu, J., Sun, Y., Si, M., Liu, M., Li, J., Zhang, X., Shang, G., Tang, B.-H., Yan, G., and Zhou, C.: Satellite remote sensing of global land surface temperature: Definition, methods, products, and applications, Rev. Geophys., 61, e2022RG000777, https://doi.org/10.1029/2022RG000777, 2023.

Li, Z.-L., Wu, H., Wang, N., Qiu, S., Sobrino, J. A., Wan, Z., Tang, B.-H., and Yan, G.: Land surface emissivity retrieval from satellite data, Int. J. Remote Sens., 34, 3084-3127, https://doi.org/10.1080/01431161.2012.716540, 2013.

Liang, S.: Estimation of Surface Radiation Budget: I. Broadband Albedo, in Quantitative Remote Sensing of Land Surfaces, John Wiley & Sons, Inc., Hoboken, NJ, USA, 310–344, 2005.

Liu, X., Li, Z.-L., Li, J.-H., Leng, P., Liu, M., and Gao. M.: Temporal upscaling of MODIS 1-km instantaneous land surface temperature to monthly mean value: Method evaluation and product generation, IEEE Trans. Geosci. Remote Sens., https://doi.org/10.1109/TGRS.2023.3247428, 2023.

Liu, Y., Liu, R., and Chen, J. M. (2012). Retrospective retrieval of long-term consistent

global leaf area index (1981–2011) from combined AVHRR and MODIS data, J. Geophys. Res-Biogeo., 117(G4), 1-14, https://doi.org/10.1029/2012JG002084, 2012.

Trigo, I. F., Monteiro, I. T., Olesen, F., and Kabsch, E.: An assessment of remotely sensed land surface temperature. J. Geophys. Res-Atmos., 113, D17, https://doi.org/10.1029/2008JD010035, 2008.

Wang, K., Wan, Z., Wang, P., Sparrow, M., Liu, J., Zhou, X., and Haginoya, S.: Estimation of surface long wave radiation and broadband emissivity using Moderate Resolution Imaging Spectroradiometer (MODIS) land surface temperature/emissivity products, J. Geophys. Res-Atmos., 110, D11, https://doi.org/10.1029/2004JD005566, 2005.

Xing, Z., Li, Z.-L., Duan, S.-B., Liu, X., Zheng, X., Leng, P., Gao, M., Zhang, X., Shang, G.: Estimation of daily mean land surface temperature at global scale using pairs of daytime and nighttime MODIS instantaneous observations, ISPRS J. Photogramm., 178, 51–67, https://doi.org/10.1016/j.isprsjprs.2021.05.017, 2021.

---

## Author Comment (AC4)

**Response to Reviewer #3**

We appreciate a lot for your efforts in providing detailed comments and recommendation. They are very helpful to improve the quality of the manuscript. We have revised the manuscript according to your comments. The comments from the reviewers are kept in regular font, our responses use blue highlighting, and the revised sentences or words in the revised manuscript are highlighted with red color.

Q1. Why was the MERRA-2 atmospheric dataset selected considering its coarse spatial resolution?

**Response**: Thank you for your comments. We would like to make some explanations on the selection of MERRA-2 atmospheric dataset as follows:

(1) To the best of our knowledge, there are only two global reanalysis datasets, the fifth-generation European Center for Medium-Range Weather Forecasts atmospheric reanalysis dataset (ERA5) and the Second Modern-Era Retrospective Analysis for Research and Applications (MERRA-2) dataset, which could provide hourly atmospheric water vapor content (WVC) data from 1981 to 2005. The ERA5 and MERRA-2 provide hourly WVC at  $0.25^{\circ} \times 0.25^{\circ}$  and  $0.625^{\circ} \times 0.5^{\circ}$  spatial resolution, respectively. Huang et. al. (2021) systematically assessed the hourly WVC of ERA5 and MERRA-2 by a comparison with 33 Global Navigation Satellite System stations from 2017 to 2018. The results of the comparison are as follows: ① The accuracies of the ERA5- and MERRA-2-derived WVC are very high; ② The performance of ERA5 is slightly better than MERRA-2 due to its higher spatial resolution; ③ With the same grid spatial resolution, the mean root mean square difference between two reanalysis data sets is  $0.1\text{mm} (0.01\text{g/cm}^2)$ .

(2) To match the GT-LST pixels, these datasets all need to be resampled to 0.05° spatial resolution. However, selecting ERA5 will consume twice as much local storage space and memory as MERRA-2.

However, the goal of this study is to develop a global historical twice-daily LST product from 1981 to 2005, where WVC is only one of intermediate variables that

obtain the nonlinear generalized split-window (GSW) algorithm coefficients corresponding to the subrange of WVC. Considering the tradeoff between accuracy, local storage space and memory and computational burden, we choose the MERRA-2-derived WVC to estimate LST.

Q2. In my opinion, the SURFRAD measurements are not the best option for LST validation, especially in the case of evaluating medium/coarse spatial resolution LST, considering the substantial spatial heterogeneity of the sites. Moreover, the measured longwave radiations by pyrgeometers are different from the directional radiance collected by satellites, which has been reported in different studies.

**Response**: Thanks a lot for your comments. We would like to make some explanations as follows:

(1) In order to maximize the usefulness of GT-LST for research it is necessary to assess the accuracy of GT-LST using as many methods as possible. Ground-based validation is the most traditional and well-used method, and it provides suitable validation results for well-defined and dedicated sites in most cases. However, there are the limited number of high-quality sites (i.e., KIT stations and NASA JPL stations) around the world that are dedicated to LST validation due to their high cost and logistical barriers (Guillevic et al., 2014). Moreover, they could only provide measurements after 2009 (Guillevic, et al., 2018). Although the SURFRAD network was not initially designed for LST validation, SURFRAD can provide high-quality radiance measurements from 1995 to present, which are useful for validating satellite LST products. SURFRAD measurements have already been used for evaluating ASTER, GOES, MODIS, VIIRS, AVHRR and AMSR-E LST products (Wang and Liang, 2009; Yu et al., 2012; Guillevic et al., 2012; Liu et al., 2019; Jiménez et al., 2017). All SURFRAD stations are selected in this study. Fig. R1 shows the surroundings of the sites on the AVHRR scale, where all of these sites except DRA are located in large flat agricultural areas covered by crops and grass. Due to not all stations are representative of spatially homogeneous areas at GT-LST scales, we used as much in situ measurements as possible in order to characterize them correctly. Considering the limitation of the ground-based validation method, we compared GT-LST with a large number of well documented and validated LST products derived from satellites to characterize GT-LST performance.

(2) Indeed, as you mentioned, longwave radiations from satellite and ground measurements may differ according to their measurement methods. AVHRR measures directional measurements in the atmospheric window, while the measured longwave radiations by pyrgeometers are hemispheric, wider spectrum derivations. If the surface is black body, the LSTs derived from these two measurements are the same (Wang et al., 2005; Li et al., 2023). However, most natural objects are not black bodies. AVHRR view zenith angles were considered to be an important factor influencing the results when comparisons were made with in situ measurements. Fig. R2 shows the relationship between view zenith angles and bias in instantaneous LST at station pixels. Our result shows that a high view angle does not necessarily bring a high bias or a low view angle does not necessarily always bring a low bias. This means that view angle should not be a significant source for the bias of GT-LST at a 0.05° pixel size.

Figure R1: Aerial photos of the SURFRAD sites. The black dot marks the position of the site, and the blue square marks the size of an AVHRR pixel

Figure R2: The relationship between bias of land surface temperature (GT-LST minus in situ observations) and view zenith angle.

Q3. Why was the MYD11 LST product selected instead of the MYD21 LST, or geostationary LST products that have closer spatial resolutions to the GT-LST product? In the inter-comparison, the MDY11A1 LST was spatially aggregated to the spatial resolution of the GT-LST product with a simpler arithmetic mean. I doubt the validity of the MYD11 LST after the simple aggregation.

**Response**: Thank you for your comments. The reason that we selected MYD11A1 LST product instead of the MYD21A1 is that the MYD11A1 LST products have been well validated by using the temperature-based method and radiance-based method methods with an accuracy of approximately 1 K (Wan, 2014; Li et al., 2023). To reduce the discrepancies induced by viewing location, time, geometry and quality control (QC), we used five criteria to guarantee the reliability of the intercomparison results. As for the criterion of collocation in space, besides that MYD11A1 LST was aggregated to the spatial resolution of the GT-LST product by averaging all MYD11A1 pixels, it also requires all MYD11A1 pixels with QC = 0 (i.e., the highest quality) within a coarse spatial resolution pixel ( $0.05^{\circ} \times 0.05^{\circ}$ ).

In addition, according to your suggestion, we have compared GT-LST with MYD21A1 LST (Aqua/MODIS LST product using the TES algorithm, Collection 6.1). Spatially, this intercomparison was conducted at the global scale. Temporally, it was performed on 4 months in 2004 (January, April, July, and October) which cover different seasons. The result of the inter-comparison, in Fig. 9, is as follows: The daytime and nighttime RMSD values of 3.2 K and 2.5 K and that of bias of 0.1 K and 1.3 K. Compared to the result of MYD11A1, the significantly smaller bias was obtained for MYD21A1. The possible reason is attributed to the fact that the MYD21A1 LST uses the same observations with MYD11A1 but uses TES method to dynamically retrieve LSE. The following contents have been added in Line 184-187 and Line 405-412, respectively.

"In this study, Collection-6.1 MYD11A1 of 2004 was selected for sensor-to-sensor comparison. MYD21A1 LST product, which uses the same observations with MYD11A1

but uses temperature–emissivity separation method to dynamically retrieve LST and emissivity, was also selected to make an intercomparison with GT-LST in this study. This inter-comparison was conducted on 4 months in 2004 (January, April, July, and October) which cover different seasons."

"As a result, the dynamic emissivity of GT-LST is typically lower than that of MYD11A1, which leads to overestimation of the LST (Hulley et al., 2016; Guillevic et al., 2014; Reiners et al., 2021; Ren et al., 2011). To further demonstrate this point, we compared GT-LST with MYD21A1 LST. Fig. 9 shows the daytime and nighttime RMSD values of 3.2 K and 2.5 K and that of bias of 0.1 K and 1.3 K between GT-LST and MYD21A1 LST for 4 months in 2004. Compared to the result of MYD11A1, the significantly smaller bias was obtained for MYD21A1. The possible reason is attributed to the fact that the MYD21A1 LST uses the same observations with MYD11A1 but uses a physics-based method to dynamically retrieve emissivity."

---

## Author Comment (AC5)

**Response to Reviewer #4**

We appreciate a lot for your efforts in providing detailed comments and recommendation. They are very helpful to improve the quality of the manuscript. We have revised the manuscript according to your comments. The comments from the reviewers are kept in regular font, our responses use blue highlighting, and the revised sentences or words in the revised manuscript are highlighted with red color.

This paper developed a global historical twice-daily LST dataset (GT-LST) with a spatial resolution of 0.05° from 1981 to 2005. I believe this is an important study and it does make sense for earth science communities. The data and methods are clearly described, and the main results are well presented. However, there are some issues that need to be addressed or clarified before the paper can be published. Therefore, I recommend a major revision.

Some major comments:

1. This paper has inter-compared the GT-LST and MODIS LST over a variety of land cover types such as savannas and cropland/natural vegetation, permanent snow and ice, water bodies and etc., yet I wonder how much are the accuracies (such as RMSD and bias) over urban surfaces?

**Response**: Thanks a lot for your comments. To clearly quantify the RMSD and bias over various land cover types, we have redrawn Fig. 7, including urban and built-up lands (UBL). For your convenience, we listed it below. For UBL, the RMSD and bias are 3.4 K and 2.7 K, respectively.

[Figure]

Figure 7: RMSD and bias between GT-LST and MYD11A1 LST in 2003 for various land cover types. ENF: evergreen needleleaf forests, EBF: evergreen broadleaf forests, DNF: deciduous needleleaf forests, DBF: deciduous broadleaf forests, MXF: mixed forests, CSR: closed shrublands, OSR: open shrublands, WDS: woody savannas, SVN: savannas, GRS: grasslands, PMW: permanent wetlands, CRP: croplands, UBL: urban and built-up lands, CNV: cropland/natural vegetation mosaics, PSI: permanent snow and ice, BRN: barren, WTB: water bodies, and ALL: all land cover types.

2. Why did you choose January 15 and July 15, 1997 for the GT-LST and RT-LST comparison over continental Africa? Please clarify the selection criteria.

**Response**: Thanks a lot for your comments. The RT-LST is a twice-daily LST product at 8-km resolution over continental Africa, which spans of 6 years from 1995 to 2000. Two days, January 15 and July 15, 1997, were chosen because they represent the median time of different seasons (winter and summer, respectively). In addition, the number of matchups is enough to guarantee the reliability of the intercomparison. We have added

some descriptions in Line 334-335 as follows:

"Two days, January 15 and July 15, 1997, were selected to implement the comparison over continental Africa because they represent the median time of different seasons (winter and summer, respectively)"

3. I just suggest combining Figs. 3 to 7 into a single figure for clarity.

**Response**: Thank you for the suggestion. We have redrawn the schematic of the workflow according to your suggestion. For your convenience, we listed it below.

[Figure]

Figure 3: Schematic of the workflow used to generate the GT-LST product

4. To what extent the differences in the emissivity between MODIS LSTs and GT-LST will influence their inter-comparison results, can you provide some quantitative results?

**Response**: Thanks a lot for your comments. From Fig.R1, one can conclude that a

negative relationship between the GT-LST and MYD11A1 LST difference and their corresponding emissivity difference. The mean biases (GT-LST – MYD11A1) for LSTs calculated with emissivity differences less than -0.05, between -0.05 and -0.03, between -0.03 and -0.01, between -0.01 and 0.01 and more than 0.01 are 7.0, 4.3, 2.3, 0.8 and 0.7 K, respectively. We have added some descriptions in Line 406-408 as follows:

"*As a result, the dynamic emissivity of GT-LST is typically lower than that of MYD11A1, which leads to overestimation of the LST (Hulley et al., 2016; Guillevic et al., 2014; Reiners et al., 2021; Ren et al., 2011). Fig. A3 shows that the mean biases (GT-LST – MYD11A1) for LSTs calculated with emissivity differences less than -0.05, between -0.05 and -0.03, between -0.03 and -0.01, between -0.01 and 0.01 and more than 0.01 are 7.0, 4.3, 2.3, 0.8 and 0.7 K, respectively.*"

[Figure]

Figure A3: Difference between GT-LST and MYD11A1 LST stratified by the difference between GT-LST and MYD11A1 emissivity (water vapor content < 5 g/cm$^2$; satellite zenith angle < 50°)

5. As you stated, the LSTs for a long period such as > 40 years are important for monitoring and evaluating global long-term climate change. Thus, the validation of tendency consistency for the generated GT-LST products is also of vital importance in addition to its spatiotemporal pattern. Could you test the accuracy of time series GT-

LSTs over several typical regions, as I guess the orbit drift of AVHRR could also introduce uncertainty for the tendency estimation.

**Response**: Thanks a lot for your comments. Indeed, global long-term climate change requires daily, monthly or annual mean LST (i.e., DMLST, MMLST, and AMLST) more than instantaneous LST as these mean LSTs are key indicators when monitoring global LSTs over a long time series (Li et al., 2023; Liu et al., 2023; Xing et al., 2021). Impacting of the NOAA satellite orbital drift, daytime and nighttime observations of NOAA afternoon satellites cannot represent maximum and minimum temperatures well. Therefore, calculating the daily and monthly mean LST by averaging daytime and nighttime LSTs derived from GT-LST has a significantly lower accuracy than other studies (Fig. A4). Inspired by the work of Xing et al. (2021), we use simple linear combinations of daytime and nighttime LST values that were observed at observation times for NOAA to estimate DMLST and MMLST. In order to validate the accuracy of DMLST and MMLST according to the simple linear regression method, we compared DMLST and MMLST derived from GT-LST with that of in situ LST observations from SURFRAD sites, and reported RMSE values of approximately 2.4 K and 2.7 K, respectively. These results are similar to that of Xing et al. (2021) and Chen et al. (2017). In this way, we still obtain accurate DMLST and MMLST without satellite orbit drift correction. In order to demonstrate the tendency consistency of GT-LST products, Fig. R1 shows time series of MMLST using the simple linear regression method from 1981 to 2005 for two small area (5°×5°) in the Sahara Desert and the Tibetan plateau: no significant inconsistencies can be seen.

[Figure]

[Figure]

Figure R1: Monthly mean LST time series for 1981 to 2005 over two small area (5°×5°) in the Sahara Desert (a) and the Tibetan plateau (b).

Then, we rephrase the paragraph in Line 429-436 as follows:

"…*To estimate MMLST, first obtain the mean instantaneous clear-sky LST at daytime and nighttime, and then use these mean values to estimate MMLST according to the* simple linear regression method (see Appendix B). *In order to validate the accuracy of MMLST results, we compared MMLST based on GT-LST with that of in situ LST observations from SURFRAD sites for* 1994–2005. *All in situ LST measurements are all-sky and complete on a certain month, which means that the in situ MMLST is true MMLST. Fig. 15 showed that MMLST derived from GT-LST are related to the true MMLST, with an $R^2$ value of* 0.94 *and an RMSE value of* 2.7 *K. This result is* similar to *that of Chen et al. (2017), who compared MMLST from MODIS day and night instantaneous clear-sky LST with actual MMLST from 156 flux tower stations, and reported RMSE bias values of approximately 2.7 K.*"

We have redrawn Fig. 13 according to the simple linear regression method. For your convenience, we listed it below.

[Figure]

*Figure 13: Monthly mean LST based on GT-LST versus monthly mean LST based on in situ LST from 1994 to 2005.*

In addition, as for some details of the simple linear regression method, we have added the following descriptions in Appendix B.

"*Impacting of the NOAA satellite orbital drift, daytime and nighttime observations of NOAA afternoon satellites cannot represent maximum and minimum temperatures well. Therefore, the MMLST according to the simple average method has a significantly lower accuracy than other studies (Fig. A4). Xing et al. (2021) proposed to use 9 combinations of two to four MODIS instantaneous retrievals of which at least one daytime LST and one nighttime LST to estimate mean LSTs, and determined the weight for every moment. Inspired by the work of Xing et al. (2021), we determined to use simple linear combinations of monthly mean daytime and nighttime LST values that*

*were observed at observation times for NOAA to estimate MMLST with ground-based measurement. For the combinations of two valid monthly mean LSTs (one daytime and one nighttime LST), the regression models can be written as follows:*

$$MMLST = a_1 * MMLST_{day} + a_2 * MMLST_{night} + b \qquad\qquad (B1)$$

*where MMLST is the ground-based monthly mean LST, $a_1$, $a_2$ and $b$ are the fitting coefficients, $MMLST_{day}$ is the monthly mean in situ LST at the NOAA daytime observation, $MMLST_{night}$ is the monthly mean in situ LST at the NOAA nighttime observation.*

*Taking into account the observed times of NOAA satellites with orbital drift effect since 1981, combinations of two observations from these satellites contain eight cases: 13:30–17:00/01:30–05:00 local solar time in 0.5-hour interval. Based on the in situ LST measurements during the period 2003 to 2018 at 227 flux stations operating in globally diverse regions, we obtained the fitting coefficients (Table A1). Then, we calculated the MMLST of GT-LST using GT-LST monthly mean daytime and nighttime LSTs, Eq. (B1), and the fitting coefficients listed in Table A1.*"

Table A1. Statistics for the relationship between the regressions of the eight combinations and actual monthly mean LST.

| Case | Time | $a_1$ | $a_2$ | $b$ | RMSE | $R^2$ | Number |
|------|------|-------|-------|-----|------|-------|--------|
| 1 | 13:30/01:30 | 0.3844 | 0.5783 | 10.3446 | 2.0 | 0.97 | 12095 |
| 2 | 14:00/02:00 | 0.4010 | 0.5621 | 10.2042 | 1.9 | 0.98 | 12241 |
| 3 | 14:30/02:30 | 0.4235 | 0.5451 | 8.6172 | 1.9 | 0.98 | 12381 |
| 4 | 15:00/03:00 | 0.4490 | 0.5211 | 8.2652 | 1.8 | 0.98 | 12303 |
| 5 | 15:30/03:30 | 0.4816 | 0.4840 | 9.5710 | 1.8 | 0.98 | 12165 |
| 6 | 16:00/04:00 | 0.5250 | 0.4349 | 11.2284 | 2.0 | 0.97 | 11818 |
| 7 | 16:30/04:30 | 0.5663 | 0.3884 | 12.8572 | 2.2 | 0.96 | 10992 |
| 8 | 17:00/05:00 | 0.6040 | 0.3621 | 9.7302 | 2.4 | 0.96 | 9765 |

Some minor Comments:

1. Line 125: Is there a writing mistake on this sentence? "we used 54 land surface emissivity spectra to represent different land surface types, including 41 soil types, four vegetation types, four water body 125 types and five ice/snow types were selected."

**Response**: We appreciate your careful reading, the phrase "were selected" was removed.

2. Line 165: "The instrumental error of the SURFRAD station give rise to uncertainty in the retrieved LST value of less than 1 K". Should be "gives rise to".

**Response**: Corrected as suggested.

3. Line 266 to 268: "Therefore, to obtain relatively accurate emissivity values, we developed an improved method that consider annual changes in land cover from the GLASS-GLC dataset and combines ASTER GED data with the NDVI threshold method to estimate the emissivity" The verb forms need to be unified.

**Response**: Thank you for your careful reading. Following your suggestion, we have checked the whole manuscript and corrected this issue.

4. Line 313 to 315: This sentence seems redundant, please write it in a more explicit way.

**Response**: Thank you for the suggestion. We have rewritten this part according to your suggestion.:

"In contrast to the ground-based validation and satellite product inter-comparison mentioned above, the comparisons for AVHRR LST products were performed using different strategies. Concretely, GT-LST compared with GD-LST using a strategy that compares GT-LST and GD-LST with same SURFRAD measurements concurrently with the satellite overpass, to evaluate the difference in the absolute accuracy of these two products."

5. You can use either RMSE or RMSD, but keep consistency throughout the paper and all figures.

**Response**: Corrected as suggested.

6. Please unify the format of all references.

**Response**: Thank you for the suggestion. Following your suggestion, we have checked the whole manuscript and corrected this issue.

**References for the above responses are listed below**:

Chen, X., Su, Z., Ma, Y., Cleverly, J., Liddell, M.: An accurate estimate of monthly mean land surface temperatures from MODIS clear-sky retrievals, J. Hydrometeorol., 18, 2827-2847, https://doi.org/10.1175/JHM-D-17-0009.1,2017.

Li, Z.-L., Wu, H., Duan, S.-B., Zhao, W., Ren, H., Liu, X., Leng, P., Tang R., Ye, X., Zhu, J., Sun, Y., Si, M., Liu, M., Li, J., Zhang, X., Shang, G., Tang, B.-H., Yan, G., and Zhou, C.: Satellite remote sensing of global land surface temperature: Definition, methods, products, and applications, Rev. Geophys., 61, e2022RG000777, https://doi.org/10.1029/2022RG000777, 2023.

Liu, X., Li, Z.-L., Li, J.-H., Leng, P., Liu, M., and Gao. M.: Temporal upscaling of MODIS 1-km instantaneous land surface temperature to monthly mean value: Method evaluation and product generation, IEEE Trans. Geosci. Remote Sens., https://doi.org/10.1109/TGRS.2023.3247428, 2023.

Xing, Z., Li, Z. L., Duan, S. B., Liu, X., Zheng, X., Leng, P., Gao, M., Zhang, X., Shang, G.: Estimation of daily mean land surface temperature at global scale using pairs of daytime and nighttime MODIS instantaneous observations, ISPRS J. Photogramm., 178, 51-67, https://doi.org/10.1016/j.isprsjprs.2021.05.017, 2021.

---

## Author Response (AR2)

**Response to Reviewer #1**

We appreciate a lot for your efforts in providing detailed comments and recommendation. They are very helpful to improve the quality of the manuscript. We have revised the manuscript according to your comments. The comments from the reviewers are kept in regular font, our responses use blue highlighting, and the revised sentences or words in the revised manuscript are highlighted with red color.

This manuscript proposes a long-term (1981-2005) AVHRR land surface temperature (LST) dataset that includes outcomes at both daytime and nighttime. The algorithm is the generalized split-window (GSW) algorithm while in the production, this dataset also considered annual land cover change. Overall, the accuracy of the proposed dataset is promising, and it filled the gaps regarding long-term global LST datasets, especially at nighttime. Therefore, I would recommend it be published on ESSD after a major revision.

**Q1.** Positive bias issue. Based on site validation and inter-comparison with MYD11 and the other two AVHRR LST products, the proposed GT-LST shows a clear positive bias (>1 K) nearly in all results. The authors claim the bias is due to the emissivity difference (Line 370), however, the proposed GT-LST has a clear bias than the other three products, and it seems that the emissivity used by GT-LST is not accurate. The authors mention that the dataset will be calibrated to remove the bias in the future (Line 436). I am thinking if it would be better to solve this issue in this paper as it doesn't need to be done in a separate paper.

**Response**: Thanks a lot for your valuable comments. First, we would like to make some explanations on the positive bias issue as follows:

(1) The GT-LST product and the global daytime AVHRR LST (GD-LST) used a dynamic emissivity method to retrieve LST. We compared GT-LST with GD-LST on January 15, April 15, July 15, and October 16, 1999, with low positive bias of 0.6 K (Figure R1).

(2) The MYD11A1 LST product and the regional twice-daily LST product over

Africa (RT-LST) are generated by the spilt-window (SW) algorithm. Land surface emissivities of these two products are assigned according to classification-based method that produces emissivities with fixed values for a limited number of land cover types. This method works well over densely vegetated areas and water where emissivities are relatively stable. However, cold biases of 3-5 K are often found over semi-arid and arid regions because these regions have much higher emissivity variability, and only one fixed overestimated emissivity inferred from land cover types is assigned to these regions (Coll et al., 2009; Hulley and Hook 2009; Wan et al., 2002). In order to represent the natural variation in emissivity, we used an improved NDVI threshold method to dynamically retrieve daily emissivity. Based on the analysis above, emissivity derived from dynamic methods is lower than emissivity according to classification-based method, which makes the proposed GT-LST is higher than MYD11A1 LST and RT-LST (i.e., positive bias). We note that earlier researches on this issue had similar results. Reiners et al. (2022) compared AVHRR LST product of the TIMELINE project with MYD11_L2 LST product from 2003 to 2014, the result shows that the TIMELINE dynamic emissivity is lower than the MYD11_L2 fixed emissivity and a general positive bias (i.e., bias=2.2 K) of TIMELINE LST towards MYD11_L2 LST. Martins et al. (2019) compared MSG LST and GOES-16 LST and revealed that a positive bias (MSG > GOES) of around 1.6 K persists due to the overestimation of the fixed emissivity of GOES. Mao et al. (2007) analyzed the retrieval result by radiative transfer model with neural network algorithm and MODIS product algorithm, indicating that MOD11_L2 LST product overestimates the emissivity, resulting in an underestimation of LST.

(3) To further illustrate the positive bias issue, we present an intercomparison exercise between MxD11A1 LST products (Terra and Aqua/MODIS using SW algorithm, Collection 6) with fixed emissivity and MSG LST products (MSG/SEVIRI using SW algorithm) with dynamic emissivity for 4 days (January 15, April, 15, July 15, and October 15, 2020). The criteria in Sec 3.2 were used to guarantee the reliability of the intercomparison results. The result is shown in Figure R2, indicating that a general positive bias (daytime ranges from 0.7 K to 3.3 K, nighttime ranges from 0.2 K

to 1.4 K) of MSG LST towards MxD11A1 LST for each land cover types.

(4) The comparison with in situ LST showed that a positive bias was found for all SURFRAD sites. However, only the bias of BND and FPK are large than 1 K. Similar results were obtained by Reiners et al. (2022) and Liu et al. (2019).

Therefore, we think that positive biases obtained for GT-LST and other LST products are relatively reasonable.

Next, many LST products can provide global twice-daily LST after 2000, such as ASTER LST, MODIS LST, VIIRS LST, AATSR LST and SLSTR LST. Users can obtain a relatively long-term twice-daily LST product by combining GT-LST with these LST products. However, integration of LST from different sensors is complicated. Due to the different LST inversion methods, air conditions, viewing geometries, etc., the sensors bias between GT-LST and other LST products is not constant. Therefore, developing a general method to utilize for sensor normalization is difficult and is not the key point of this paper.

[Figure]

Figure R1. GT-LST versus GD-LST during the daytime on January 15, April, 15, July 15, and October 16, 1999.

a

[Figure]

b

[Figure]

Figure R2. The bias (SEVIRI LST minus MxD11A1 LST) and RMSD between the MxD11A1 product and SEVIRI during daytime (a) and nighttime (b) over various land cover types.

**Q2.** Large RMSE (4.1 K) of the monthly mean LST result. The GT-LST is claimed to have the strength to generate gap-free monthly mean LST; however, the outcome has an RMSE of 4.1 K which is too large at a monthly scale compared to other studies (Line 395). This part weakened the statement of the advantage of GT-LST for temporal upscaling based on the logic chain. I would suggest either removing this part or quantifying the impact of orbit drift, in other words, comparing the accuracies of samples that have not and have suffered from orbit drift, and then claiming the potential of this data after orbit drift.

**Response**: Thanks for your suggestion. We used simple linear combinations of monthly mean daytime and nighttime LST values to estimate MMLST. The detailed revisions are listed as follows.

"…*To estimate MMLST, first obtain the mean instantaneous clear-sky LST at daytime and nighttime, and then use these mean values to estimate MMLST according to the* simple linear regression method (see Appendix B). *In order to validate the accuracy of MMLST results, we compared MMLST based on GT-LST with that of in situ LST observations from SURFRAD sites for* 1994–2005. *All in situ LST measurements are all-sky and complete on a certain month, which means that the in situ MMLST is true MMLST. Fig. 15 showed that MMLST derived from GT-LST are related to the true MMLST, with an $R^2$ value of* 0.94 *and an RMSE value of* 2.7 K. *This result is* similar to *that of Chen et al. (2017), who compared MMLST from MODIS day and night instantaneous clear-sky LST with actual MMLST from 156 flux tower stations, and reported RMSE bias values of approximately 2.7 K.*"

We have redrawn Fig. 13 according to the simple linear regression method. For your convenience, we listed it below.

[Figure]

*Figure 13. Monthly mean LST based on GT-LST versus monthly mean LST based on in situ LST from 1995 to 2005.*

We have added the following descriptions in Appendix B.

 *"Impacting of the NOAA satellite orbital drift, daytime and nighttime observations of NOAA afternoon satellites cannot represent maximum and minimum temperatures well. Therefore, the MMLST according to the simple average method has a significantly lower accuracy than other studies (Figure A3). Xing et al. (2021) proposed to use 9 combinations of two to four MODIS instantaneous retrievals of which at least one daytime LST and one nighttime LST to estimate mean LSTs, and determined the weight for every moment. Inspired by the work of Xing et al. (2021), we determined to use simple linear combinations of monthly mean daytime and nighttime LST values that were observed at observation times for NOAA to estimate MMLST with ground-based*

*measurement. For the combinations of two valid monthly mean LSTs (one daytime and one nighttime LST), the regression models can be written as follows*:

$$MMLST = a_1 * MMLST_{day} + a_2 * MMLST_{night} + b \tag{B1}$$

*where MMLST is the ground-based monthly mean LST, $a_1$, $a_2$ and $b$ are the fitting coefficients, $MMLST_{day}$ is the monthly mean in situ LST at the NOAA daytime observation, $MMLST_{night}$ is the monthly mean in situ LST at the NOAA nighttime observation.*

*Taking into account the observed times of NOAA satellites with orbital drift effect since 1981, combinations of two observations from these satellites contain eight cases: 13:30–17:00/01:30–05:00 local solar time in 0.5-hour interval. Based on the in situ LST measurements during the period 2003 to 2018 at 227 flux stations operating in globally diverse regions, we obtained the fitting coefficients (Table A1). Then, we calculated the MMLST of GT-LST using GT-LST monthly mean daytime and nighttime LSTs, Eq. (B1), and the fitting coefficients listed in Table A1.*"

Table A1. Statistics for the relationship between the regressions of the eight combinations and actual monthly mean LST.

| Case | Time | $a_1$ | $a_2$ | b | RMSE | $R^2$ | Number |
|------|------|-------|-------|------|------|-------|--------|
| 1 | 13:30/01:30 | 0.3844 | 0.5783 | 10.3446 | 2.0 | 0.97 | 12095 |
| 2 | 14:00/02:00 | 0.4010 | 0.5621 | 10.2042 | 1.9 | 0.98 | 12241 |
| 3 | 14:30/02:30 | 0.4235 | 0.5451 | 8.6172 | 1.9 | 0.98 | 12381 |
| 4 | 15:00/03:00 | 0.4490 | 0.5211 | 8.2652 | 1.8 | 0.98 | 12303 |
| 5 | 15:30/03:30 | 0.4816 | 0.4840 | 9.5710 | 1.8 | 0.98 | 12165 |
| 6 | 16:00/04:00 | 0.5250 | 0.4349 | 11.2284 | 2.0 | 0.97 | 11818 |
| 7 | 16:30/04:30 | 0.5663 | 0.3884 | 12.8572 | 2.2 | 0.96 | 10992 |
| 8 | 17:00/05:00 | 0.6040 | 0.3621 | 9.7302 | 2.4 | 0.96 | 9765 |

**Q3.** The impact of annual land cover change. This is an interesting part of the study, whereas the study didn't pay attention to the performance of such change. Traditionally people mainly utilized a land cover climatology map rather than annual changes to retrieve global LST. I would suggest including additional analysis to find some

examples and compare with LST from Ma et al. (2020) to demonstrate the progress using annual land cover maps.

**Response**: This is a good suggestion! Changes in land cover have been accelerating since 1980 under the impact of climate changes and human activities. As an intrinsic property of natural materials, land surface emissivity predominantly depends on the land cover type. Therefore, using only one year of land cover data to determine long-term emissivity is not accurate. The quantitative relationship between annual land cover change and LST is rather complex because the changes of Land surface temperature were related to many factors, including changes in land cover, land surface parameters, seasonal variation, climatic condition and economic development, etc. Furthermore, GT-LST and LST from Ma et al. (2020) used different LST retrieval algorithms and data sources, which makes it harder to analyze the impact of annual land cover change between these two LST products. However, this is a meaningful research topic, and we will further analyze the impact in future work.

**Q4.** Some processes were not introduced clearly.

**Q4.1** why does not GT-LST cover 1981 to 2022? GAC raw data is still updating.

**Response**: Thanks a lot for your comment. The reasons that GT-LST only cover 1981 to 2005 are as follows:

Existing satellite-based global twice-daily LST products can only date back to 2000. Therefore, when the study began, we aimed to fill the data gap of global satellite-derived twice-daily LST before 2000. Considering global meteorology and climatology-related applications urgently need more than 30 years of daily LST products, GT-LST can be combined with the existing satellite-derived daily LST product (e.g., MODIS LST, AATSR LST and ASTER LST) after 2000 to satisfy that requirement. However, integration of LST from different sensors need to eliminate or limit the bias between the sensors. We then extend the time span of GT-LST to 2005. Benefiting from the same observation period with other LST products, these extended

data can be used to calibrate the bias between GT-LST and other LST datasets. In this way, users can obtain a relatively homogeneous twice-daily LST product for 1981 to 2022. However, we will apply your suggestion to extend the time span of GT-LST to 2022 in the near future.

**Q4.2** why did the authors only employ the site observations from 1995 to 2000? If you can extend it to 2005, you can include one more SURFRAD site.

**Response:** Thank you for your suggestion. We have extended the observations of SURFRAD sites to 2005 and employed one more SURFRAD site (i.e., SXF) observations according to your suggestion. We have redrawn Fig. 4. For your convenience, we listed it below.

[Figure]

*Figure 4. GT-LST versus in situ LST for 1995–2005 at (a) BND, (b) DRA, (c) FPK, (d) GWN, (e) PSU, (f) SXF, and (g) TBL sites.*

**Q4.3** Regarding the site validation, 6 sites seem not enough to represent the accuracy of the global product. I would recommend adding some BSRN sites that also have good data quality.

**Response**: Thanks for your suggestion. Following your comments, we have added some BSRN sites to represent the accuracy of the GT-LST product in contrasting climatic zones. The following contents have been added in Section 2.5 and Section 4.1, respectively.

*"…The BSRN has 76 stations that detect important changes in the Earth's radiation field at the Earth's surface since 1992. These stations provide high-quality surface and upper-air meteorological observations, which are important in supporting the validation and confirmation of satellite. We selected four sites with measurements of upwelling and downwelling TIR radiances before 2000 (Table 3)."*

*"…We further compared GT-LST data with in situ LST data at BAR, NYA, PYA, and TAT sites for 1995–2005. Fig. 5 shows the scatterplots between GT-LST and in situ LST at these four BSRN sites. The accuracy of GT-LST product at BSRN sites is relatively worse than that at SURFRAD sites, with RMSE (bias) ranges from 3.1 K (-2.7 K) to 4.0 K (2.5 K)."*

*Table 3. Details of the validation sites used in this study.*

|  | Name | Elevation(m) | Land cover type | Latitude | Longitude | Valid period |
|---|---|---|---|---|---|---|
| SURFRAD | BND | 230 | Croplands | 40.0519 | -88.3731 | 1995–2005 |
| | DRA | 1007 | Open shrublands | 36.6237 | -116.0195 | 1998–2005 |
| | FPK | 634 | Grasslands | 48.3078 | -105.1017 | 1994–2005 |
| | GWN | 98 | Cropland/natural vegetation mosaic | 34.2547 | -89.8729 | 1994–2005 |
| | PSU | 376 | Cropland/natural vegetation mosaic | 40.7201 | -77.9309 | 1998–2005 |
| | TBL | 1689 | Grasslands | 40.1250 | -105.2368 | 1995–2005 |
| | SXF | 473 | Croplands | 43.7343 | -96.6233 | 2003–2005 |
| BSRN | BAR | 8 | Tundra | 71.3230 | -156.6070 | 1995–2005 |
| | NYA | 11 | Tundra | 78.9227 | 11.9273 | 1999–2005 |
| | PAY | 491 | Cultivated | 46.8123 | 6.9422 | 1995–2005 |
| | TAT | 25 | Grass | 36.0581 | 140.1258 | 1996–2005 |

[Figure]

*Figure 5. Scatterplots between GT-LST and in situ LST at (a) BAR, (b) NYA, (c) PYA, and (d) TAT.*

**Q4.4** why Fig 9(b) has some considerable scattered samples? Those cases should be discussed in the context.

**Response:** Thanks for your comment. We have added some discussion in Section 4.2 for the revised manuscript as follows:

*"...However, as can be seen in Fig.10(b), large LST differences (GT-LST - MYD11A1 LST) more than 20 K are mostly distributed in red box. Through counting, there are 111 samples in red box, which are barren land cover type and arid climate*

*type. Fig. A1 shows the distribution of each scattered samples in red box. 77 of 111 samples happened in Haiya, Sudan on March 31, 2004. The samples of rest happened in Taif, Saudi Arabia on April 2, 2004. For these samples, we double-checked all variables that are essential parameters in GT-LST retrieval. The result show that all scope variables are reasonable except BT of TIR bands. Abnormal high BTs at these nighttime samples were found on March 31 and April 2, 2004 (Fig. A2), which leaded to extreme high LSTs. The possible reasons for abnormal high BTs are as follows: (1) These two regions may have experienced extreme events such as wars and natural disasters on March 31 or April 2, 2004. But we didn't find relevant information from historical news and documents. (2) Another factor may be instrument failure on these two days."*

[Figure]

*Figure 6. Inter-comparison of GT-LST and MYD11A1 LST in 2004: (a) daytime; (b) nighttime. Red box indicates considerable scattered samples.*

[Figure]

*Figure A1. Distribution of the 111 scattered samples.*

[Figure]

*Figure A2. An example of abnormal high BTs on (a) March 31, 2004 and (b) April 2, 2004.*

**Q4.5** Line 350: as MODIS has been spatially aggregated to match with GT-LST, why spatial heterogeneity is still an issue here?

**Response:** Thanks for your comment. We have deleted this erroneous expression.

**Q4.6** Fig10: I would suggest changing Fig10 to another format: consider RMSE and bias as the two dimensions of the plot, and mark each dot by their names as using color to show the bias is not easily quantified.

**Response:** Thanks for your valuable suggestion. We have redrawn Fig. 11 according to your suggestion. For your convenience, we listed it below.

[Figure]

*Figure 7. RMSD and bias between GT-LST and MYD11A1 LST in 2003 for various land cover types. ENF: evergreen needleleaf forests, EBF: evergreen broadleaf forests, DNF: deciduous needleleaf forests, DBF: deciduous broadleaf forests, MXF: mixed forests, CSR: closed shrublands, OSR: open shrublands, WDS: woody savannas, SVN: savannas, GRS: grasslands, PMW: permanent wetlands, CRP: croplands, UBL: urban and built-up lands, CNV: cropland/natural vegetation mosaics, PSI: permanent snow and ice, BRN: barren, WTB: water bodies, and ALL: all land cover types.*

**Q4.7** Line 357: why do savannas and cropland show considerable bias?

**Response:** Fig. R3 shows relatively large disparities between GT-LST and MYD11A1 LST over savannas (i.e., woody savannas and savannas) and croplands (i.e., cropland/natural vegetation mosaics and croplands) for the intercomparison. We would like to make some explanations on large disparities between these two products as follows:

According to NDVI threshold method, the daily emissivity of an AVHRR pixel can be derived using the following formula:

$$\varepsilon = \varepsilon_{veg} * FVC + \varepsilon_{soil} * (1 - FVC) \qquad \qquad (R1)$$

Here, $\varepsilon$ is the emissivity, $\varepsilon_{veg}$ is the vegetation emissivity, $\varepsilon_{soil}$ is the bare soil emissivity, and $FVC$ is the fraction of vegetation cover.

For a vegetation pixel, its FVC is less than 1 due to the influence of natural and human factors, which leads to the underestimation of emissivity comparing with fixed emissivity, resulting in an overestimation of LST. The situation is particularly evident over croplands and savannas. Specially, natural disasters (e.g., drought and pests) and agricultural activities (e.g., harvest, cropland lies fallow) can significantly decrease cropland density and result in higher exposure of the soil. It leads to a decrease in cropland emissivity, resulting in an overestimation of LST. The emissivity for savannas decreases because of the increasing proportion of soil by grazing, fire and annually a long period in which moisture inadequate, resulting in an overestimation of LST.

[Figure]

Figure R3. Scatterplots of GT-LST versus MYD11A1 LST during 2004 over WDS (a), SVN (b), CRP (c), and CNV (d). WDS: woody savannas, SVN: savannas, CRP: croplands, and CNV: cropland/natural vegetation mosaics.

Minor:

1. Line 35: Some of them used surface air temperature rather than LST to detect climate change and it should be not mixed.

**Response:** Thank you for your careful reading. We have removed the reference (i.e., Keenan and Riley, 2018) in the revised manuscript.

2. Line 71: remove 'the'

**Response:** Corrected as suggested.

3. Line 94: polar-orbiting

**Response:** Corrected as suggested.

4. line 101: the first

**Response:** Corrected as suggested.

5. Line 179: Especially

**Response:** Corrected as suggested.

6. Line 298: identifier

**Response:** Corrected as suggested.

7. Line 301: difference

**Response:** Corrected as suggested.

8. Line 317: due to -> because

**Response:** Corrected as suggested.

9. Line 327: RMSEs

**Response:** Corrected as suggested.

10. Line 403: remove 'in'

**Response:** Corrected as suggested.

11. Line 404: 'due to' should be followed by a noun rather than a sentence, suggest revising the whole manuscript for this issue.

**Response:** Thank you for your careful reading. Following your suggestion, we have checked the whole manuscript and corrected this issue.

12. Line 411: considers

**Response:** Corrected as suggested.

13. Line 446: open-source

**Response:** Corrected as suggested.

14. Line 451: cloud mask

**Response:** Corrected as suggested.

**References for the above responses are listed below:**

Chen, X., Su, Z., Ma, Y., Cleverly, J., Liddell, M.: An accurate estimate of monthly

mean land surface temperatures from MODIS clear-sky retrievals, J. Hydrometeorol., 18, 2827-2847, doi:10.1175/JHM-D-17-0009.1,2017.

Coll, C., Wan, Z., Galve, J. M.: Temperature-based and radiance-based validations of the V5 MODIS land surface temperature product, J Geophys. Res-Atmos., 114, 1-15, doi:10.1029/2009jd012038, 2009.

Hulley, G. C., Hook, S. J.: The North American ASTER land surface emissivity database (NAALSED) version 2.0, Remote Sens. Environ, 113, 1967-1975, doi:10.1016/j.rse.2009.05.005, 2009.

Liu, X., Tang, B.-H., Yan, G., Li, Z.-L., Liang, S.: Retrieval of Global Orbit Drift Corrected Land Surface Temperature from Long-term AVHRR Data, Remote Sens., 11, 2843, doi:10.3390/rs11232843, 2019.

Mao, K., Shi, J., Li, Z. L., Tang, H.: An RM-NN algorithm for retrieving land surface temperature and emissivity from EOS/MODIS data, J Geophys. Res-Atmos., 112, 1-17, doi: 10.1029/2007JD008428, 2009.

Martins, J. P., Coelho e Freitas, S., Trigo, I. F., Barroso, C., Macedo, J.: Copernicus Global Land Operations-Lot I "Vegetation and Energy" Algorithm Theoretical Basis Document. Land Surface Temperature—LST, 1, 2019.

Reiners, P., Asam, S., Frey, C., Holzwarth, S., Bachmann, M., Sobrino, J., Göttsche, F., Bendix, J. Kuenzer, C.: Validation of AVHRR Land Surface Temperature with MODIS and In Situ LST—A TIMELINE Thematic Processor, Remote Sens., 13, 3473, doi:10.3390/rs13173473, 2021.

Xing, Z., Li, Z. L., Duan, S. B., Liu, X., Zheng, X., Leng, P., Gao, M., Zhang, X., Shang, G.: Estimation of daily mean land surface temperature at global scale using pairs of

daytime and nighttime MODIS instantaneous observations, ISPRS J. Photogramm., 178, 51-67, doi:10.1016/j.isprsjprs.2021.05.017, 2021.

Wan, Z., Zhang, Y., Zhang, Q., Li, Z. L.: Validation of the land-surface temperature products retrieved from Terra Moderate Resolution Imaging Spectroradiometer data, Remote Sens. Environ, 83, 163-180, doi:10.1016/S0034-4257(02)00093-7, 2002.

Response to the second comments of Reviewer #1

We appreciate a lot for your efforts in providing detailed comments and recommendation. They are very helpful to improve the quality of the manuscript. We have revised the manuscript according to your comments. The comments from the reviewers are kept in regular font, our responses use blue highlighting, and the revised sentences or words in the revised manuscript are highlighted with red color.

Thank you for your response which resolved many of my concerns. However, I am still wondering if you have addressed some key issues of the AVHRR GAC data:

Q1. As the archived historical data, the AVHRR GAC raw data have a serious geolocation issue that has been criticized by Wu et al. (2020), especially when the view zenith angle is larger than 40-deg, thus I would suggest the authors deal with this issue or at least quantify the impact. Please double-check previous literature and collect such data issues and give a comprehensive discussion.

**Response**: Thanks for your valuable suggestion. Indeed, Wu et al. (2020) provides a preliminary geolocation assessment for AVHRR GAC data of NOAA-17, MetOp-A, and MetOp-B, which present shifts that stay within the range of 4 km for satellite zenith angles smaller than 40° and can reach 6 km when the satellite zenith angle is larger than 40°. To be more clearly for readers, we have added the detail of the geolocation issue according to the work of Wu et al. (2020) in Line 105-107 as follows:

"*Therefore, AVHRR Level-1b GAC data are generally treated as having a coarse resolution of 4 km at the nadir, and the pixel size increases with the satellite zenith angle (VZA). Furthermore, as the VZA increases, the geolocation accuracy of the AVHRR GAC scene become lower, particularly when VZAs larger than 40° (Wu et al., 2020).*".

In addition, considering the influence of geolocation issue, we used an open-source package, Pygac, to pre-process AVHRR GAC data. Pygac, which is based on ephemeris data, orbit model and time of onboard clock, uses correction of satellite location method, correction of scanline timestamps method, correction of geolocation method to improve the geolocation accuracy of the AVHRR GAC data. After the GAC data are treated

through above methods, we believe that their geolocation accuracy basically meets the demand of global applications at 0.05° spatial resolution. However, if users need high geolocation quality GAC data, we suggested that the GAC data less than 40° should be preferred. We have clarified this point in Line 473-476 with the expression as follows:

"*...A variety of factors such as cloud cover, orbital gaps, and instrument failure are responsible for this limitation. And finally, the geolocation accuracy of GT-LST product basically meets the demand of global applications at 0.05° spatial resolution. However, if users need very high geolocation quality GT-LST data, we suggested that the GT-LST data with VZAs less than 40° should be preferred.*"

Q2. It still doesn't make sense that the data ended in 2005 artificially and the other reviewer also agreed with my suggestion.

**Response**: Thanks for your comment. We are sorry for the previously unclear explanations. As emphasized in the introduction section, this study aims to fill the data gap of global satellite-derived twice-daily LST before 2000. However, considering global meteorology and climatology-related applications urgently need more than 30 years of daily LST products, there are two ways of satisfying that requirement based on GT-LST. One way is to combine GT-LST (1981-2000) with the existing satellite-derived daily LST product (2000-present), which depend on different products with the same observation period to eliminate or limit the bias between different sensors. Therefore, we extend the time span of GT-LST to 2005. Benefiting from the same observation period (i.e., 2000-2005) with MODIS LST, we will produce a global long-term (1981-present) LST data record according to the method of Liu et al. (2012), which will be primarily from the AVHRR (1981-2000) and MODIS (2000-present).

Indeed, as you mentioned, extending the time span of GT-LST to present is another way to address this issue. We have already started working on generating GT-LST products (2006-present). Although we have proposed a framework for generating GT-LST product, we still need spend a lot of time downloading global AVHRR GAC L1B data, handling large amounts of original Level-1B data, generating huge amounts of process variable data, and so on. After all data have been processed, we will upload GT-

LST (2006-present) to previous URL (https://doi.org/10.5281/zenodo.7134158).

Q3. The monthly mean LST still has an overall bias of 1.3 K compared to site observations, please double-check the code or provide a discussion and comparison with previous work.

**Response**: Thanks for your suggestion. According to your suggestion, we have double-checked the code but not found problems. In addition, we have added the discussion of positive bias between monthly mean GT-LST and in situ LST in Line 439-441 as follows:

"*…This result is similar to that of Chen et al. (2017), who compared MMLST from MODIS day and night instantaneous clear-sky LST with actual MMLST from 156 flux tower stations, and reported RMSE value of approximately 2.7 K. However, it should be noted that a positive bias of 1.3 K between GT-LST MMLST and in situ MMLST. One possible reason is that in situ MMLST of some sites does not represent the MMLST over the 0.05°×0.05° pixel.*"

**References for the above responses are listed below**:

Liu, Y., Liu, R., and Chen, J. M.: Retrospective retrieval of long-term consistent global leaf area index (1981–2011) from combined AVHRR and MODIS data, J. Geophys. Res-Biogeo., 117, 1-14, https://doi.org/10.1029/2012JG002084, 2012.

Wu, X., Naegeli, K., and Wunderle, S.: Geometric accuracy assessment of coarse-resolution satellite datasets: a study based on AVHRR GAC data at the sub-pixel level, Earth Syst. Sci. Data, 12, 539–553, https://doi.org/10.5194/essd-12-539-2020, 2020.

**Response to the third comments of Reviewer #1**

Q1. Line 70: space missed after 0.5°

**Response**: Thank you for your careful reading. We have corrected this issue.

Q2. Based on Fig 12, the MMLST (monthly mean LST) still has clear gaps, so the statement 'all-sky' or 'gap-free' (line 28) is not completely accurate, even though the improvement for minimizing the cloud gaps is quite significant.

**Response**: We appreciate your careful reading, the phrase "under all-sky conditions" in Line 28 was removed.

Q3. Obtaining monthly mean could be a considerable contribution to the community while as the authors mentioned in the manuscript, the data are still limited by the orbital drift issue. I would recommend the authors provide some additional statistics to verify how much the RMSE of MMLST would change due to orbital drift after several years, probably temporal accuracy variation analysis for N14 would be enough.

This would help to assess the impact of the orbital drift issue on the accuracy of the MMLST product over time, providing motivation for future studies. By analyzing how the RMSE changes over time, the authors could provide a better understanding of the limitations and potential biases of the data, which would be useful for the community using this product.

**Response**: Thanks for your suggestion. Many studies have shown that two satellite observations that are separated by approximately 12 h can be used to estimate a relatively accurate daily and monthly mean LST (i.e., DMLST and MMLST) by weighted average method (Chen et al., 2017; Liu et al., 2023; Xing et al., 2021). Based on the in situ LST measurements, the weight for every moment in every combination (one daytime and one nighttime) is determined. Fig. R1 displays the density scatter plots that depict the true in situ DMLST against the weighted average of the in situ day/night observations (one daytime and one nighttime) at the time of NOAA satellite overpass. The results show that the DMLST obtained by the proposed method is not

influenced by the orbital drift of the satellite. We applied the same methodology to compute the MMLST, as described in Appendix B. Theoretically, the MMLST is also independent of the satellite orbit drift, resulting in a constant RMSE over time.

[Figure]

Figure R1. Comparison of the actual in situ DMLST and the in situ DMLST estimated using a weighted average method based on NOAA satellites with orbital drift effect at 76 stations from 2003 to 2018: (a) 13:30/01:30, (b) 14:00/02:00, (c) 14:30/02:30, (d) 15:00/3:00, (e) 15:30/03:30, (f) 16:00/04:00, (g) 16:30/04:30, and (h) 17:00/05:00 (adapted from Li et al., 2023).

**References for the above responses are listed below**:

Chen, X., Su, Z., Ma, Y., Cleverly, J., and Liddell, M.: An accurate estimate of monthly mean land surface temperatures from MODIS clear-sky retrievals, J. Hydrometeorol., 18, 2827-2847, https://doi.org/10.1175/JHM-D-17-0009.1, 2017.

Li, J.-H., Li, Z.-L., Liu, X., Duan, S.-B., Si, M., Shang, G., and Zhang, X.: A generalized method for retrieving global daily mean land surface temperature from polar-orbiting thermal infrared sensor instantaneous observations, Int. J. Remote Sens., Under review.

Liu, X., Li, Z.-L., Li, J.-H., Leng, P., Liu, M., and Gao. M.: Temporal upscaling of MODIS 1-km instantaneous land surface temperature to monthly mean value: Method evaluation and product generation, IEEE Trans. Geosci. Remote Sens., https://doi.org/10.1109/TGRS.2023.3247428, 2023.

Xing, Z., Li, Z.-L., Duan, S.-B., Liu, X., Zheng, X., Leng, P., Gao, M., Zhang, X., and Shang, G.: Estimation of daily mean land surface temperature at global scale using pairs of daytime and nighttime MODIS instantaneous observations, ISPRS J. Photogramm., 178, 51–67, https://doi.org/10.1016/j.isprsjprs.2021.05.017, 2021.

Response to Reviewer #2

We appreciate a lot for your efforts in providing detailed comments and recommendation. They are very helpful to improve the quality of the manuscript. We have revised the manuscript according to your comments. The comments from the reviewers are kept in regular font, our responses use blue highlighting, and the revised sentences or words in the revised manuscript are highlighted with red color.

In the study titled "A global historical twice-daily (daytime and nighttime) land surface temperature dataset produced by AVHRR observations from 1981 to 2005", the authors produce a global LST product from 1981 to 2005 at 0.05 degree using AVHRR observations. The study is potentially useful for understanding changes in surface climate over a longer time period than what we can currently examine using most existing LST products. However, I have several concerns that should be addressed before the paper is considered for publication.

Q1. The biggest issue I have is that the dataset is restricted to 2005. Given that AVHRR products have large biases compared to MODIS Aqua and use different inputs (such as the dynamic emissivity estimates used), one cannot combine MODIS and AVHRR to perform long-term analysis. Since the AVHRR is still operational, the dataset needs to be extended to more recent years.

Response: Thanks for your valuable suggestion. As emphasized in the introduction section, this study aims to fill the data gap of global satellite-derived twice-daily LST before 2000. However, considering global meteorology and climatology-related applications urgently need more than 30 years of daily LST products, there are two ways of satisfying that requirement based on GT-LST. One way is to combine GT-LST (1981-2000) with the existing satellite-derived daily LST product (2000-present), which depend on different products with the same observation period to eliminate or limit the bias between different sensors. Therefore, we extend the time span of GT-LST to 2005. Benefiting from the same observation period (i.e., 2000-2005) with MODIS LST, we will produce a global long-term (1981-present) LST data record according to

the method of Liu et al. (2012), which will be primarily from the AVHRR (1981-2000) and MODIS (2000-present).

Indeed, as you mentioned, extending the time span of GT-LST to present is another way to address this issue. We have already started working on generating GT-LST products (2006-present). Although we have proposed a framework for generating GT-LST product, we still need spend a lot of time downloading global AVHRR GAC L1B data, handling large amounts of original Level-1b data, generating huge amounts of process variable data, and so on. After all data have been processed, we will upload GT-LST (2006-present) to previous URL (https://doi.org/10.5281/zenodo.7134158).

Q2. As an addendum to the previous point, since one of the most important use cases of long-term datasets is time series analysis, the long-term changes in GT-LST should be compared against equivalent changes from MODIS products. If the orbital drift has a significant impact on long-term trends, we should be very cautious about the suitability of this data product for this use case. This issue needs to be quantified more clearly instead of just discussed in text in one section. This can potentially avoid misleading results from future uses of this dataset.

**Response**: Thanks a lot for your comments. Indeed, one of the intentions of GT-LST is providing effective supplementary data for global long-term time series analysis. The analysis requires daily, monthly or annual mean LST (i.e., DMLST, MMLST, and AMLST) more than instantaneous LST as these mean LSTs are key indicators when monitoring global LSTs over a long time series (Li et al., 2023; Liu et al., 2023; Xing et al., 2021). It is possible to derive an estimate of the global accurate DMLST, MMLST and AMLST based on twice-daily LST product. However, impacting of the NOAA satellite orbital drift, daytime and nighttime observations of NOAA afternoon satellites cannot represent maximum and minimum temperatures well. Therefore, calculating the daily and monthly mean LST by averaging daytime and nighttime LSTs derived from GT-LST has a significantly lower accuracy than other studies (Figure A4). Inspired by the work of Xing et al. (2021), we use simple linear combinations of daytime and nighttime LST values that were observed at observation times for NOAA to estimate

DMLST and MMLST. In order to validate the accuracy of DMLST and MMLST according to the simple linear regression method, we compared DMLST and MMLST derived from GT-LST with that of in situ LST observations from SURFRAD sites, and reported RMSE values of approximately 2.4 K and 2.7 K, respectively. These results are similar to that of Xing et al. (2021) and Chen et al. (2017). In this way, we still obtain accurate DMLST and MMLST without satellite orbit drift correction. Then, we rephrase the paragraph in Line 429-436 as follows:

"…*To estimate MMLST, first obtain the mean instantaneous clear-sky LST at daytime and nighttime, and then use these mean values to estimate MMLST according to the simple linear regression method (see Appendix B). In order to validate the accuracy of MMLST results, we compared MMLST based on GT-LST with that of in situ LST observations from SURFRAD sites for 1994–2005. All in situ LST measurements are all-sky and complete on a certain month, which means that the in situ MMLST is true MMLST. Fig. 13 showed that MMLST derived from GT-LST are related to the true MMLST, with an $R^2$ value of 0.94 and an RMSE value of 2.7 K. This result is similar to that of Chen et al. (2017), who compared MMLST from MODIS day and night instantaneous clear-sky LST with actual MMLST from 156 flux tower stations, and reported RMSE bias values of approximately 2.7 K.*"

We have redrawn Fig. 13 according to the simple linear regression method. For your convenience, we listed it below.

[Figure]

*Figure 13: Monthly mean LST based on GT-LST versus monthly mean LST based on in situ LST from 1994 to 2005.*

In addition, as for some details of the simple linear regression method, we have added the following descriptions in Appendix B.

*"Impacting of the NOAA satellite orbital drift, daytime and nighttime observations of NOAA afternoon satellites cannot represent maximum and minimum temperatures well. Therefore, the MMLST according to the simple average method has a significantly lower accuracy than other studies (Fig. A4). Xing et al. (2021) proposed to use 9 combinations of two to four MODIS instantaneous retrievals of which at least one daytime LST and one nighttime LST to estimate mean LSTs, and determined the weight for every moment. Inspired by the work of Xing et al. (2021), we determined to use simple linear combinations of monthly mean daytime and nighttime LST values that*

*were observed at observation times for NOAA to estimate MMLST with ground-based measurement. For the combinations of two valid monthly mean LSTs (one daytime and one nighttime LST), the regression models can be written as follows*:

*$MMLST = a_1 * MMLST_{day} + a_2 * MMLST_{night} + b$* (B1)

*where MMLST is the ground-based monthly mean LST, $a_1$, $a_2$ and $b$ are the fitting coefficients, $MMLST_{day}$ is the monthly mean in situ LST at the NOAA daytime observation, $MMLST_{night}$ is the monthly mean in situ LST at the NOAA nighttime observation.*

*Taking into account the observed times of NOAA satellites with orbital drift effect since 1981, combinations of two observations from these satellites contain eight cases: 13:30–17:00/01:30–05:00 local solar time in 0.5-hour interval. Based on the in situ LST measurements during the period 2003 to 2018 at 227 flux stations operating in globally diverse regions, we obtained the fitting coefficients (Table A1). Then, we calculated the MMLST of GT-LST using GT-LST monthly mean daytime and nighttime LSTs, Eq. (B1), and the fitting coefficients listed in Table A1.*"

Table A1. Statistics for the relationship between the regressions of the eight combinations and actual monthly mean LST.

| Case | Time | $a_1$ | $a_2$ | b | RMSE | $R^2$ | Number |
|---|---|---|---|---|---|---|---|
| 1 | 13:30/01:30 | 0.3844 | 0.5783 | 10.3446 | 2.0 | 0.97 | 12095 |
| 2 | 14:00/02:00 | 0.4010 | 0.5621 | 10.2042 | 1.9 | 0.98 | 12241 |
| 3 | 14:30/02:30 | 0.4235 | 0.5451 | 8.6172 | 1.9 | 0.98 | 12381 |
| 4 | 15:00/03:00 | 0.4490 | 0.5211 | 8.2652 | 1.8 | 0.98 | 12303 |
| 5 | 15:30/03:30 | 0.4816 | 0.4840 | 9.5710 | 1.8 | 0.98 | 12165 |
| 6 | 16:00/04:00 | 0.5250 | 0.4349 | 11.2284 | 2.0 | 0.97 | 11818 |
| 7 | 16:30/04:30 | 0.5663 | 0.3884 | 12.8572 | 2.2 | 0.96 | 10992 |
| 8 | 17:00/05:00 | 0.6040 | 0.3621 | 9.7302 | 2.4 | 0.96 | 9765 |

Q3. A second major source of concern is the dynamic emissivity method used. There are several vegetation-adjusted emissivity methods available, which can give different values, different enough to account for some of the biases seen. Of note, at 0.05 degree, you would start resolving larger urban areas, which is a major use case for satellitederived LST (Voogt & Oke, 2003). Different emissivity methods perform differently over urban surfaces, which impacts this important use case (Chakraborty et al. 2021). Ideally, this issue needs to be tested further using different emissivity methods.

**Response**: This is a good suggestion! As you mentioned, to date, various land surface emissivity (LSE) estimation methods have been proposed with the same goal but different advantages, and limitations, e.g., classification-based method, NDVI threshold method, TES method, and physics-based day/night method and so on (Li et al., 2013). With their advantages and limitations, these methods have different accuracies and are applicable for various sensors and applications. To reflect the performance for the emissivity-retrieved methods and account for the positive bias between GT-LST and MYD11A1 LST, according to your suggestion, we compared GT-LST with MYD21A1 LST that uses the same observations with MYD11A1 but uses a physics-based algorithm to dynamically retrieve both LST and spectral emissivity. The intercomparison results of MYD21A1 LST showed a very lower bias. As for the comparison with MYD21A1, a brief explanation was analyzed below in Q4. There is no denying that it is important and significative to evaluate emissivity methods under different circumstances and for various applications. However, the goal of this study is to develop a global historical twice-daily LST product from 1981 to 2005, where LSE is only one of the key parameters. Therefore, we choose an improved NDVI threshold method to estimate LSEs from space for a global case by taking the sensor characteristics, the required accuracy, as well as computation time into account. Although evaluating emissivity methods under different circumstances is not be discussed in more detail in this study, this is a meaningful research topic. Inspired by your suggestion and the work of Chakraborty et al. (2021), we will evaluate these methods on an identical standard and to give the quality and accuracy on their applications in future work. After evaluating them we may attempt to generate the first estimates of LST at a global scale using AVHRR GAC data by combining all these approaches.

Q4. The comparison with MODIS MYD11 is somewhat difficult because of the

different emissivity method used. The comparison should be done against MODIS MYD21, which uses the same observations, but a temperature-emissivity separation method instead of classification-based prescribed emissivity.

**Response**: Thanks for your valuable comments. According to your comments, we have compared GT-LST with MYD21A1 LST (Aqua/MODIS LST product using the TES algorithm, Collection 6.1). Spatially, this intercomparison was conducted at the global scale. Temporally, it was performed on 4 months in 2004 (January, April, July, and October) which cover different seasons. The results of the daytime and nighttime comparison, in Fig. 9, are as follows: The daytime and nighttime RMSD values of 3.2 K and 2.5 K and that of bias of 0.1 K and 1.3 K. Compared to the result of MYD11A1, the significantly smaller bias was obtained for MYD21A1. The possible reason is attributed to the fact that the MYD21A1 LST uses the same observations with MYD11A1 but uses TES method to dynamically retrieve LSE. The following contents have been added in Line 184-187 and Line 405-412, respectively.

"*In this study, Collection-6.1 MYD11A1 of 2004 was selected for sensor-to-sensor comparison. MYD21A1 LST product, which uses the same observations with MYD11A1 but uses temperature–emissivity separation method to dynamically retrieve LST and emissivity, was also selected to make an intercomparison with GT-LST in this study. This inter-comparison was conducted on 4 months in 2004 (January, April, July, and October) which cover different seasons.*"

"*As a result, the dynamic emissivity of GT-LST is typically lower than that of MYD11A1, which leads to overestimation of the LST (Hulley et al., 2016; Guillevic et al., 2014; Reiners et al., 2021; Ren et al., 2011). To further demonstrate this point, we compared GT-LST with MYD21A1 LST. Fig. 9 shows the daytime and nighttime RMSD values of 3.2 K and 2.5 K and that of bias of 0.1 K and 1.3 K between GT-LST and MYD21A1 LST for 4 months in 2004. Compared to the result of MYD11A1, the significantly smaller bias was obtained for MYD21A1. The possible reason is attributed to the fact that the MYD21A1 LST uses the same observations with MYD11A1 but uses a physics-based method to dynamically retrieve emissivity.*"

[Figure]

Figure 9: Intercomparison of GT-LST and MYD21A1 LST in January, April, July, and October 2004: (a) daytime; (b) nighttime.

Q5. For comparison with SURFAD stations, did the authors check that the emissivity used to generate the LST in the ground observations is same as the LST in the GT-LST product? If they are different, would be good to adjust by the emissivity difference and check if that improves the accuracy.

**Response**: The GT-LST use directional measurements of AVHRR in the atmospheric window, while the SURFRAD stations provide upwelling and downwelling broadband hemispherical TIR radiances using pyrgeometers in the spectral range from 3.5 to 50 μm, from which estimates of LSTs can be derived using Stefan–Boltzmann's law. To retrieve LST using Stefan–Boltzmann's law, the surface broadband emissivity must be known a priori. In this study, these broadband emissivities were estimated from ASTER emissivity product using a spectral-to-broadband linear regression equation according to the work of Duan et al. (2019), as follows: BND(0.968), TBL(0.972), DRA(0.967), FPK(0.973), GCM(0.971), PSU(0.970), and SXF(0.970). According to the study of Liang (2005), the surface broadband emissivity of sites can be obtained from AVHRR LSE in AVHRR LSE for channel centered at 11 and 12 μm via the empirical relationship:
$$\varepsilon = 0.2489 + 0.2386\varepsilon_{11} + 0.4998\varepsilon_{12} \tag{R1}$$

According to Eq. (R1), the surface broadband emissivities are 0.976, 0.975, 0.972, 0.973, 0.973, 0.968 and 0.974 for BND, TBL, DRA, FPK, GCM, PSU, and SXF,

respectively. Different empirical relationships perform an error less than 0.01 in the broadband emissivity. According to the study of Xing et al. (2021), the emissivity changes by 0.01, and the change in in-situ LST will not exceed 0.37 K. Therefore, while this error is not negligible, it does not appear to be a dominant source of uncertainty in the ground-based validation.

Q6. Finally, given the view angle of AVHRR, a broader discussion needs to be added about thermal anisotropy (DUffour et al. 2015). Satellites only provide a 2d directional view of LST, and this is not directly comparable across satellites (Landsat vs MODIS) or against ground observations that have a downward pointing radiometer. This is of particular concern over heterogeneous terrain, such as mixed forests and over cities.

**Response**: Thanks a lot for your valuable comments. we would like to make some explanations on the thermal anisotropy issue as follows:

(1) Previous multi-sensor comparison studies (Guillevic et al., 2012; Trigo et al., 2008) found differences up to 12 K between MODIS and SEVIRI LST due to directional effects. Appropriate matchups significantly reduce the discrepancies induced by directional effects (Guillevic et al., 2014). In this study, to avoid the uncertainties induced by directional effects, a strict criterion of viewing geometry alignment was established to guarantee the reliability of the intercomparison results (Li et al., 2023): the difference in VZA between MYD11A1 and GT-LST is limited to be less than 15°.

(2) LST from satellite and ground measurements may differ according to their measurement methods. AVHRR use directional measurements in the atmospheric window, while ground-based longwave radiation measurements are hemispheric, wider spectrum derivations. If the surface is black body, the two LSTs are the same (Wang et al., 2005, Li et al., 2023). However, most natural objects are not black bodies. AVHRR view zenith angles were considered to be an important factor influencing the results when comparisons were made with in situ measurements. Figure R1 shows the relationship between view angles and bias in instantaneous LST at station pixels. Our result is that a high view angle does not necessarily show a high bias and a low view

angle does not necessarily always show a low bias. This means that view angle should not be a significant source for the bias of GT-LST at a 0.05° pixel size.

[Figure]

Figure R1: The relationship between bias of land surface temperature (GT-LST minus in situ observations) and view zenith angle.

**References for the above responses are listed below**:

Chakraborty, T. C., Lee, X., Ermida, S., and Zhan, W.: On the land emissivity assumption and Landsat-derived surface urban heat islands: A global analysis, Remote Sens. Environ., 265, 112682, https://doi.org/10.1016/j.rse.2021.112682, 2021.

Chen, X., Su, Z., Ma, Y., Cleverly, J., Liddell, M.: An accurate estimate of monthly mean land surface temperatures from MODIS clear-sky retrievals, J. Hydrometeorol., 18, 2827-2847, https://doi.org/10.1175/JHM-D-17-0009.1, 2017.

Duan, S.-B., Li, Z.-L., Li, H., Göttsche, F.-M., Wu, H., Zhao, W., Leng, P., Zhang, X., Coll, C.: Validation of Collection 6 MODIS land surface temperature product using in situ measurements, Remote Sens. Environ. 225, 16–29, https://doi.org/10.1016/j.rse.2019.02.020, 2019.

Guillevic, P. C., Biard, J. C., Hulley, G. C., Privette, J. L., Hook, S. J., Olioso, A., Göttsche F. M., Radocinski, R., Román, M. O., Yu, Y., and Csiszar, I.: Validation of

Land Surface Temperature products derived from the Visible Infrared Imaging Radiometer Suite (VIIRS) using ground-based and heritage satellite measurements, Remote Sens. Environ., 154, 19-37, https://doi.org/10.1016/j.rse.2014.08.013, 2014.

Guillevic, P. C., Privette, J. L., Coudert, B., Palecki, M. A., Demarty, J., Ottlé, C., and Augustine, J. A.: Land Surface Temperature product validation using NOAA's surface climate observation networks—Scaling methodology for the Visible Infrared Imager Radiometer Suite (VIIRS), Remote Sens. Environ., 124, 282-298, https://doi.org/10.1016/j.rse.2012.05.004, 2012.

Li, Z.-L., Wu, H., Duan, S.-B., Zhao, W., Ren, H., Liu, X., Leng, P., Tang R., Ye, X., Zhu, J., Sun, Y., Si, M., Liu, M., Li, J., Zhang, X., Shang, G., Tang, B.-H., Yan, G., and Zhou, C.: Satellite remote sensing of global land surface temperature: Definition, methods, products, and applications, Rev. Geophys., 61, e2022RG000777, https://doi.org/10.1029/2022RG000777, 2023.

Li, Z.-L., Wu, H., Wang, N., Qiu, S., Sobrino, J. A., Wan, Z., Tang, B.-H., and Yan, G.: Land surface emissivity retrieval from satellite data, Int. J. Remote Sens., 34, 3084-3127, https://doi.org/10.1080/01431161.2012.716540, 2013.

Liang, S.: Estimation of Surface Radiation Budget: I. Broadband Albedo, in Quantitative Remote Sensing of Land Surfaces, John Wiley & Sons, Inc., Hoboken, NJ, USA, 310–344, 2005.

Liu, X., Li, Z.-L., Li, J.-H., Leng, P., Liu, M., and Gao. M.: Temporal upscaling of MODIS 1-km instantaneous land surface temperature to monthly mean value: Method evaluation and product generation, IEEE Trans. Geosci. Remote Sens., https://doi.org/10.1109/TGRS.2023.3247428, 2023.

Liu, Y., Liu, R., and Chen, J. M. (2012). Retrospective retrieval of long-term consistent

global leaf area index (1981–2011) from combined AVHRR and MODIS data, J. Geophys. Res-Biogeo., 117(G4), 1-14, https://doi.org/10.1029/2012JG002084, 2012.

Trigo, I. F., Monteiro, I. T., Olesen, F., and Kabsch, E.: An assessment of remotely sensed land surface temperature. J. Geophys. Res-Atmos., 113, D17, https://doi.org/10.1029/2008JD010035, 2008.

Wang, K., Wan, Z., Wang, P., Sparrow, M., Liu, J., Zhou, X., and Haginoya, S.: Estimation of surface long wave radiation and broadband emissivity using Moderate Resolution Imaging Spectroradiometer (MODIS) land surface temperature/emissivity products, J. Geophys. Res-Atmos., 110, D11, https://doi.org/10.1029/2004JD005566, 2005.

Xing, Z., Li, Z.-L., Duan, S.-B., Liu, X., Zheng, X., Leng, P., Gao, M., Zhang, X., Shang, G.: Estimation of daily mean land surface temperature at global scale using pairs of daytime and nighttime MODIS instantaneous observations, ISPRS J. Photogramm., 178, 51–67, https://doi.org/10.1016/j.isprsjprs.2021.05.017, 2021.

**Response to Reviewer #3**

We appreciate a lot for your efforts in providing detailed comments and recommendation. They are very helpful to improve the quality of the manuscript. We have revised the manuscript according to your comments. The comments from the reviewers are kept in regular font, our responses use blue highlighting, and the revised sentences or words in the revised manuscript are highlighted with red color.

Q1. Why was the MERRA-2 atmospheric dataset selected considering its coarse spatial resolution?

**Response**: Thank you for your comments. We would like to make some explanations on the selection of MERRA-2 atmospheric dataset as follows:

(1) To the best of our knowledge, there are only two global reanalysis datasets, the fifth-generation European Center for Medium-Range Weather Forecasts atmospheric reanalysis dataset (ERA5) and the Second Modern-Era Retrospective Analysis for Research and Applications (MERRA-2) dataset, which could provide hourly atmospheric water vapor content (WVC) data from 1981 to 2005. The ERA5 and MERRA-2 provide hourly WVC at 0.25°×0.25° and 0.625°×0.5° spatial resolution, respectively. Huang et. al. (2021) systematically assessed the hourly WVC of ERA5 and MERRA-2 by a comparison with 33 Global Navigation Satellite System stations from 2017 to 2018. The results of the comparison are as follows: ① The accuracies of the ERA5- and MERRA-2-derived WVC are very high; ② The performance of ERA5 is slightly better than MERRA-2 due to its higher spatial resolution; ③ With the same grid spatial resolution, the mean root mean square difference between two reanalysis data sets is 0.1mm (0.01g/cm$^2$).

(2) To match the GT-LST pixels, these datasets all need to be resampled to 0.05° spatial resolution. However, selecting ERA5 will consume twice as much local storage space and memory as MERRA-2.

However, the goal of this study is to develop a global historical twice-daily LST product from 1981 to 2005, where WVC is only one of intermediate variables that

obtain the nonlinear generalized split-window (GSW) algorithm coefficients corresponding to the subrange of WVC. Considering the tradeoff between accuracy, local storage space and memory and computational burden, we choose the MERRA-2-derived WVC to estimate LST.

Q2. In my opinion, the SURFRAD measurements are not the best option for LST validation, especially in the case of evaluating medium/coarse spatial resolution LST, considering the substantial spatial heterogeneity of the sites. Moreover, the measured longwave radiations by pyrgeometers are different from the directional radiance collected by satellites, which has been reported in different studies.

**Response**: Thanks a lot for your comments. We would like to make some explanations as follows:

(1) In order to maximize the usefulness of GT-LST for research it is necessary to assess the accuracy of GT-LST using as many methods as possible. Ground-based validation is the most traditional and well-used method, and it provides suitable validation results for well-defined and dedicated sites in most cases. However, there are the limited number of high-quality sites (i.e., KIT stations and NASA JPL stations) around the world that are dedicated to LST validation due to their high cost and logistical barriers (Guillevic et al., 2014). Moreover, they could only provide measurements after 2009 (Guillevic, et al., 2018). Although the SURFRAD network was not initially designed for LST validation, SURFRAD can provide high-quality radiance measurements from 1995 to present, which are useful for validating satellite LST products. SURFRAD measurements have already been used for evaluating ASTER, GOES, MODIS, VIIRS, AVHRR and AMSR-E LST products (Wang and Liang, 2009; Yu et al., 2012; Guillevic et al., 2012; Liu et al., 2019; Jiménez et al., 2017). All SURFRAD stations are selected in this study. Fig. R1 shows the surroundings of the sites on the AVHRR scale, where all of these sites except DRA are located in large flat agricultural areas covered by crops and grass. Due to not all stations are representative of spatially homogeneous areas at GT-LST scales, we used as much in situ measurements as possible in order to characterize them correctly. Considering

the limitation of the ground-based validation method, we compared GT-LST with a large number of well documented and validated LST products derived from satellites to characterize GT-LST performance.

(2) Indeed, as you mentioned, longwave radiations from satellite and ground measurements may differ according to their measurement methods. AVHRR measures directional measurements in the atmospheric window, while the measured longwave radiations by pyrgeometers are hemispheric, wider spectrum derivations. If the surface is black body, the LSTs derived from these two measurements are the same (Wang et al., 2005; Li et al., 2023). However, most natural objects are not black bodies. AVHRR view zenith angles were considered to be an important factor influencing the results when comparisons were made with in situ measurements. Fig. R2 shows the relationship between view zenith angles and bias in instantaneous LST at station pixels. Our result shows that a high view angle does not necessarily bring a high bias or a low view angle does not necessarily always bring a low bias. This means that view angle should not be a significant source for the bias of GT-LST at a 0.05° pixel size.

[Figure]

Figure R1: Aerial photos of the SURFRAD sites. The black dot marks the position of the site, and the blue square marks the size of an AVHRR pixel

[Figure]

Figure R2: The relationship between bias of land surface temperature (GT-LST minus in situ observations) and view zenith angle.

Q3. Why was the MYD11 LST product selected instead of the MYD21 LST, or geostationary LST products that have closer spatial resolutions to the GT-LST product? In the inter-comparison, the MDY11A1 LST was spatially aggregated to the spatial resolution of the GT-LST product with a simpler arithmetic mean. I doubt the validity of the MYD11 LST after the simple aggregation.

**Response**: Thank you for your comments. The reason that we selected MYD11A1 LST product instead of the MYD21A1 is that the MYD11A1 LST products have been well validated by using the temperature-based method and radiance-based method methods with an accuracy of approximately 1 K (Wan, 2014; Li et al., 2023). To reduce the discrepancies induced by viewing location, time, geometry and quality control (QC), we used five criteria to guarantee the reliability of the intercomparison results. As for the criterion of collocation in space, besides that MYD11A1 LST was aggregated to the spatial resolution of the GT-LST product by averaging all MYD11A1 pixels, it also requires all MYD11A1 pixels with QC = 0 (i.e., the highest quality) within a coarse spatial resolution pixel (0.05°×0.05°).

In addition, according to your suggestion, we have compared GT-LST with MYD21A1 LST (Aqua/MODIS LST product using the TES algorithm, Collection 6.1). Spatially, this intercomparison was conducted at the global scale. Temporally, it was performed on 4 months in 2004 (January, April, July, and October) which cover different seasons. The result of the inter-comparison, in Fig. 9, is as follows: The daytime and nighttime RMSD values of 3.2 K and 2.5 K and that of bias of 0.1 K and 1.3 K. Compared to the result of MYD11A1, the significantly smaller bias was obtained for MYD21A1. The possible reason is attributed to the fact that the MYD21A1 LST uses the same observations with MYD11A1 but uses TES method to dynamically retrieve LSE. The following contents have been added in Line 184-187 and Line 405-412, respectively.

*"In this study, Collection-6.1 MYD11A1 of 2004 was selected for sensor-to-sensor comparison. MYD21A1 LST product, which uses the same observations with MYD11A1*

*but uses temperature–emissivity separation method to dynamically retrieve LST and emissivity, was also selected to make an intercomparison with GT-LST in this study. This inter-comparison was conducted on 4 months in 2004 (January, April, July, and October) which cover different seasons.*"

"*As a result, the dynamic emissivity of GT-LST is typically lower than that of MYD11A1, which leads to overestimation of the LST (Hulley et al., 2016; Guillevic et al., 2014; Reiners et al., 2021; Ren et al., 2011). To further demonstrate this point, we compared GT-LST with MYD21A1 LST. Fig. 9 shows the daytime and nighttime RMSD values of 3.2 K and 2.5 K and that of bias of 0.1 K and 1.3 K between GT-LST and MYD21A1 LST for 4 months in 2004. Compared to the result of MYD11A1, the significantly smaller bias was obtained for MYD21A1. The possible reason is attributed to the fact that the MYD21A1 LST uses the same observations with MYD11A1 but uses a physics-based method to dynamically retrieve emissivity.*"

[Figure]

Figure 9: Intercomparison of GT-LST and MYD21A1 LST in January, April, July, and October 2004: (a) daytime; (b) nighttime.

Q4. Typo in Eq. 5. It should be 'AVH' rather than 'AST' on the left side of the equation.

**Response**: Thank you for your careful reading. We have corrected this issue.

Q5. The description of the emissivity retrieval process is unclear. How were the soil

type data used?

**Response**: Sorry for the unclear expression. To make it clear, the paragraph in Line 254-258 has been revised to as follows:

"...where $\varepsilon_{i,s}^{AST}$ is the bare soil emissivity in ASTER channel $i$ (i=10, ..., 14), and $\varepsilon_{i,v}^{AST}$ is the emissivity of dense vegetation in ASTER channel $i$. Because the emissivity spectra of dense vegetation are similar and vary slightly in the TIR region, we used the dense vegetation emissivity of ASTER channel $i$ provided by Meng et al. (2016). $\varepsilon_i^{AST}$ is the emissivity of the ASTER GED product in channel $i$. $P_v$ is calculated from the NDVI of the ASTER GED product according to Eq. (3). *For long-term cloud cover pixels and dense vegetation pixels ($P_v = 1$), the bare soil emissivity of these ASTER pixels are null values. To generate a global gap-free bare soil emissivity map of ASTER, we used the average emissivity of the same soil type within 5×5 neighborhood pixels to fill these null values. Because of some pixels with no valid neighbor pixel for averaging we needed to enlarge the neighborhood until all null values are filled.* Soil-type data are described in Section 2.3."

Q6. Why were the RMSEs over savannas and croplands the largest amongst different land surface types?

**Response**: Fig. R3 shows the intercomparison results between GT-LST and MYD11A1 LST over savannas (i.e., woody savannas and savannas) and croplands (i.e., cropland/natural vegetation mosaics and croplands). We would like to make some explanations on relatively large disparities over savannas and croplands between these two products as follows:

According to NDVI threshold method, the daily emissivity of an AVHRR pixel can be derived using the following formula:

$$\varepsilon = \varepsilon_{veg} * FVC + \varepsilon_{soil} * (1 - FVC) \qquad (R1)$$

Here, $\varepsilon$ is the emissivity, $\varepsilon_{veg}$ is the vegetation emissivity, $\varepsilon_{soil}$ is the bare soil emissivity, and $FVC$ is the fraction of vegetation cover.

For a vegetation pixel, its FVC varies greatly due to the influence of natural and human factors, which leads to the underestimation of emissivity comparing with fixed emissivity, resulting in an overestimation of LST. The situation is particularly evident

over croplands and savannas. Specially, natural disasters (e.g., drought and pests) and agricultural activities (e.g., harvest, cropland lies fallow) can significantly decrease cropland density and result in higher exposure of the soil. It leads to a decrease in cropland emissivity, resulting in an overestimation of LST. The emissivity for savannas decreases because of the increasing proportion of soil by grazing, fire and annually a long period in which moisture inadequate, resulting in an overestimation of LST. We have added the following descriptions in Line 391-393:

"*Savannas and croplands, including woody savannas and savannas, croplands and cropland/natural vegetation mosaics, respectively, had the largest RMSD. The possible reason is that the fraction of vegetation cover of savannas and croplands vary greatly due to the influence of natural and human factors, which leads to the underestimation of emissivity comparing with fixed emissivity of MYD11A1, resulting in an overestimation of LST. Snow and ice and water bodies had the smallest RMSD.*"

[Figure]

Figure R3. Scatterplots of GT-LST versus MYD11A1 LST during 2004 over WDS (a), SVN (b), CRP (c), and CNV (d). WDS: woody savannas, SVN: savannas, CRP:

croplands, and CNV: cropland/natural vegetation mosaics.

Q7. Any explanations for the higher uncertainties in spring and summer?

**Response**: Thanks a lot for your comments. A reasonable explanation could be that differences seasons associated with different atmospheric conditions: cool and dry in autumn and winter, hot and wet in spring and summer. Generally, large differences between different LST products were found under high temperatures and high atmospheric water vapor content conditions. Some literatures got similar result. For example, Reiners et al. (2021) compared AVHRR LST product of the TIMELINE project with MYD11_L2 LST product from 2003 to 2014. The result shows the seasonal pattern, with lower accordance and higher bias in summer and higher accordance and lower bias in winter.

Q8. Line 370, why is the ASTER GED-based emissivity retrieval used in the GT-LST product lower than the classification-based emissivity used in the MYD11 product?

**Response**: Thanks a lot for your valuable comments. We would like to make some explanations on the lower emissivity of GT-LST product as follows:

(1) The MYD11A1 LST product is generated by the spilt-window (SW) algorithm. Land surface emissivities of this product are assigned according to classification-based method that produces emissivities with fixed values for a limited number of land cover types. This method works well over densely vegetated areas and water where emissivities are relatively stable. However, cold biases of 3-5 K are often found over semi-arid and arid regions because these regions have much higher emissivity variability, and only one fixed overestimated emissivity inferred from land cover types is assigned to these regions (Coll et al., 2009; Hulley and Hook 2009; Wan et al., 2002). In order to represent the natural variation in emissivity, we used an improved NDVI threshold method to dynamically retrieve daily emissivity. Based on the analysis above, emissivity derived from dynamic methods is lower than emissivity according to classification-based method, which makes the proposed GT-LST is higher than

MYD11A1 LST (i.e., positive bias). We note that earlier researches on this issue had similar results. Reiners et al. (2021) compared AVHRR LST product of the TIMELINE project with MYD11_L2 LST product from 2003 to 2014, the result shows that the TIMELINE dynamic emissivity is lower than the MYD11_L2 fixed emissivity and a general positive bias (i.e., bias=2.2 K) of TIMELINE LST towards MYD11_L2 LST. Martins et al. (2019) compared MSG LST and GOES-16 LST and revealed that a positive bias (MSG > GOES) of around 1.6 K persists due to the overestimation of the fixed emissivity of GOES. Mao et al. (2007) analyzed the retrieval result by radiative transfer model with neural network algorithm and MODIS product algorithm, indicating that MOD11_L2 LST product overestimates the emissivity, resulting in an underestimation of LST.

(2) To further illustrate the MYD11A1 LST product overestimates the emissivity, we present an intercomparison exercise between MxD11A1 LST products (Terra and Aqua/MODIS using SW algorithm, Collection 6) with fixed emissivity and MSG LST products (MSG/SEVIRI using SW algorithm) with dynamic emissivity for 4 days (January 15, April, 15, July 15, and October 15, 2020). The criteria in Sec 3.2 were used to guarantee the reliability of the intercomparison results. The result is shown in Fig. R4, indicating that a general positive bias (daytime ranges from 0.7 K to 3.3 K, nighttime ranges from 0.2 K to 1.4 K) of MSG LST towards MxD11A1 LST for each land cover types.

Based on the analysis above, it is reasonable to conclude that the GT-LST dynamic emissivity is lower than the MYD11A1 classification-based emissivity.

a

[Figure]

b

[Figure]

Figure R4: The bias (SEVIRI LST minus MxD11A1 LST) and RMSD between the MxD11A1 product and SEVIRI during daytime (a) and nighttime (b) over various land cover types.

Q9. Line 400, why is the emissivity of GT-LST lower than that of RT-LST? The analysis is too simple to understand the intercomparison between different LST products. More in-depth investigations are needed for the comparison with the existing AVHRR LST data.

**Response**: RT-LST product provided only a rough estimation of emissivity using a land cover classification map, the FAO soil map of Africa and additional maps of tree, herbaceous, and bare soil percent cover. All of these data sets are static and therefore authors do not account for local phenological, environmental changes, and human factors in time. Based on the analysis above, emissivity derived from dynamic methods is lower than emissivity according to static method.

Q10. There are quite some minor grammatical errors, e.g., Line 411, 'an improved method that consider annual changes'. Please check them carefully.

**Response**: Thank you for your careful reading. Following your suggestion, we have checked the whole manuscript and corrected these issues.

Q11. The authors mentioned the increased uncertainties of AVHRR LST with time due to the orbital drift. It would be useful to add some analyses of the variation in the accuracy of the GT-LST product in the time series.

**Response**: Thanks a lot for your comments. Indeed, one of the intentions of GT-LST is providing effective supplementary data for global long-term time series analysis. The analysis requires daily, monthly or annual mean LST (i.e., DMLST, MMLST, and AMLST) more than instantaneous LST as these mean LSTs are key indicators when monitoring global LSTs over a long time series (Li et al., 2023; Liu et al., 2023; Xing et al., 2021). It is possible to derive an estimate of the global accurate DMLST, MMLST and AMLST based on twice-daily LST product. However, impacting of the NOAA satellite orbital drift, daytime and nighttime observations of NOAA afternoon satellites cannot represent maximum and minimum temperatures well. Therefore, calculating the daily and monthly mean LST by averaging daytime and nighttime LSTs derived from GT-LST has a significantly lower accuracy than other studies (Fig. A4). Inspired by the work of Xing et al. (2021), we use simple linear combinations of daytime and nighttime LST values that were observed at observation times for NOAA to estimate DMLST and MMLST. In order to validate the accuracy of DMLST and MMLST according to the simple linear regression method, we compared DMLST and MMLST derived from GT-

LST with that of in situ LST observations from SURFRAD sites, and reported RMSE values of approximately 2.4 K and 2.7 K, respectively. These results are similar to that of Xing et al. (2021) and Chen et al. (2017). In this way, we still obtain accurate DMLST and MMLST without satellite orbit drift correction. Then, we rephrase the paragraph in Line 429-436 as follows:

"…*To estimate MMLST, first obtain the mean instantaneous clear-sky LST at daytime and nighttime, and then use these mean values to estimate MMLST according to the* simple linear regression method (see Appendix B)*. In order to validate the accuracy of MMLST results, we compared MMLST based on GT-LST with that of in situ LST observations from SURFRAD sites for* 1994–2005*. All in situ LST measurements are all-sky and complete on a certain month, which means that the in situ MMLST is true MMLST. Fig. 15 showed that MMLST derived from GT-LST are related to the true MMLST, with an R$^2$ value of* 0.94 *and an RMSE value of* 2.7 K*. This result is* similar to *that of Chen et al. (2017), who compared MMLST from MODIS day and night instantaneous clear-sky LST with actual MMLST from 156 flux tower stations, and reported RMSE bias values of approximately 2.7 K.*"

We have redrawn Fig. 13 according to the simple linear regression method. For your convenience, we listed it below.

[Figure]

*Figure 13: Monthly mean LST based on GT-LST versus monthly mean LST based on in situ LST from 1994 to 2005.*

In addition, as for some details of the simple linear regression method, we have added the following descriptions in Appendix B.

*"Impacting of the NOAA satellite orbital drift, daytime and nighttime observations of NOAA afternoon satellites cannot represent maximum and minimum temperatures well. Therefore, the MMLST according to the simple average method has a significantly lower accuracy than other studies (Fig. A4). Xing et al. (2021) proposed to use 9 combinations of two to four MODIS instantaneous retrievals of which at least one daytime LST and one nighttime LST to estimate mean LSTs, and determined the weight for every moment. Inspired by the work of Xing et al. (2021), we determined to use simple linear combinations of monthly mean daytime and nighttime LST values that were observed at observation times for NOAA to estimate MMLST with ground-based*

*measurement. For the combinations of two valid monthly mean LSTs (one daytime and one nighttime LST), the regression models can be written as follows*:

$$MMLST = a_1 * MMLST_{day} + a_2 * MMLST_{night} + b \qquad (B1)$$

*where MMLST is the ground-based monthly mean LST, $a_1$, $a_2$ and $b$ are the fitting coefficients, $MMLST_{day}$ is the monthly mean in situ LST at the NOAA daytime observation, $MMLST_{night}$ is the monthly mean in situ LST at the NOAA nighttime observation.*

*Taking into account the observed times of NOAA satellites with orbital drift effect since 1981, combinations of two observations from these satellites contain eight cases: 13:30–17:00/01:30–05:00 local solar time in 0.5-hour interval. Based on the in situ LST measurements during the period 2003 to 2018 at 227 flux stations operating in globally diverse regions, we obtained the fitting coefficients (Table A1). Then, we calculated the MMLST of GT-LST using GT-LST monthly mean daytime and nighttime LSTs, Eq. (B1), and the fitting coefficients listed in Table A1.*"

Table A1. Statistics for the relationship between the regressions of the eight combinations and actual monthly mean LST.

| Case | Time | $a_1$ | $a_2$ | $b$ | RMSE | $R^2$ | Number |
|------|------|-------|-------|-----|------|-------|--------|
| 1 | 13:30/01:30 | 0.3844 | 0.5783 | 10.3446 | 2.0 | 0.97 | 12095 |
| 2 | 14:00/02:00 | 0.4010 | 0.5621 | 10.2042 | 1.9 | 0.98 | 12241 |
| 3 | 14:30/02:30 | 0.4235 | 0.5451 | 8.6172 | 1.9 | 0.98 | 12381 |
| 4 | 15:00/03:00 | 0.4490 | 0.5211 | 8.2652 | 1.8 | 0.98 | 12303 |
| 5 | 15:30/03:30 | 0.4816 | 0.4840 | 9.5710 | 1.8 | 0.98 | 12165 |
| 6 | 16:00/04:00 | 0.5250 | 0.4349 | 11.2284 | 2.0 | 0.97 | 11818 |
| 7 | 16:30/04:30 | 0.5663 | 0.3884 | 12.8572 | 2.2 | 0.96 | 10992 |
| 8 | 17:00/05:00 | 0.6040 | 0.3621 | 9.7302 | 2.4 | 0.96 | 9765 |

**References for the above responses are listed below**:

Chen, X., Su, Z., Ma, Y., Cleverly, J., Liddell, M.: An accurate estimate of monthly mean land surface temperatures from MODIS clear-sky retrievals, J. Hydrometeorol., 18, 2827-2847, https://doi.org/10.1175/JHM-D-17-0009.1,2017.

Coll, C., Wan, Z., Galve, J. M.: Temperature-based and radiance-based validations of

the V5 MODIS land surface temperature product, J Geophys. Res-Atmos., 114, 1-15, https://doi.org/10.1029/2009jd012038, 2009.

Guillevic, P. C., Biard, J. C., Hulley, G. C., Privette, J. L., Hook, S. J., Olioso, A., Göttsche F. M., Radocinski, R., Román, M. O., Yu, Y., and Csiszar, I.: Validation of Land Surface Temperature products derived from the Visible Infrared Imaging Radiometer Suite (VIIRS) using ground-based and heritage satellite measurements, Remote Sens. Environ., 154, 19-37, https://doi.org/10.1016/j.rse.2014.08.013, 2014.

Guillevic, P., Göttsche, F., Nickeson, J., Hulley, G., Ghent, D., Yu, Y., Trigo, I., Hook, S., Sobrino, J. A., Remedios, J., Román, M., and Camacho, F.: Land surface temperature product validation best practice protocol, https://lpvs.gsfc.nasa.gov/PDF/CEOS_LST_PROTOCOL_Feb2018_v1.1.0_light.pdf, 2018.

Guillevic, P. C., Privette, J. L., Coudert, B., Palecki, M. A., Demarty, J., Ottlé, C., and Augustine, J. A.: Land Surface Temperature product validation using NOAA's surface climate observation networks—Scaling methodology for the Visible Infrared Imager Radiometer Suite (VIIRS), Remote Sens. Environ., 124, 282-298, https://doi.org/10.1016/j.rse.2012.05.004, 2012.

Huang, L., Mo, Z., Liu, L., Zeng, Z., Chen, J., Xiong, S., and He, H.: Evaluation of hourly PWV products derived from ERA5 and MERRA-2 over the Tibetan Plateau using ground-based GNSS observations by two enhanced models, Earth Space Sci., 8, e2020EA001516, https://doi.org/10.1029/2020EA001516, 2021.

Hulley, G. C., Hook, S. J.: The North American ASTER land surface emissivity database (NAALSED) version 2.0, Remote Sens. Environ, 113, 1967-1975, https://doi.org/10.1016/j.rse.2009.05.005, 2009.

Li, Z.-L., Wu, H., Duan, S.-B., Zhao, W., Ren, H., Liu, X., Leng, P., Tang R., Ye, X., Zhu, J., Sun, Y., Si, M., Liu, M., Li, J., Zhang, X., Shang, G., Tang, B.-H., Yan, G., and Zhou, C.: Satellite remote sensing of global land surface temperature: Definition, methods, products, and applications, Rev. Geophys., 61, e2022RG000777,

https://doi.org/10.1029/2022RG000777, 2023.

Liu, X., Li, Z.-L., Li, J.-H., Leng, P., Liu, M., and Gao. M.: Temporal upscaling of MODIS 1-km instantaneous land surface temperature to monthly mean value: Method evaluation and product generation, IEEE Trans. Geosci. Remote Sens., https://doi.org/10.1109/TGRS.2023.3247428, 2023.

Liu, X., Tang, B.-H., Yan, G., Li, Z.-L., Liang, S.: Retrieval of Global Orbit Drift Corrected Land Surface Temperature from Long-term AVHRR Data, Remote Sens., 11, 2843, https://doi.org/10.3390/rs11232843, 2019.

Jiménez, C., Prigent, C., Ermida, S. L., and Moncet, J. L.: Inversion of AMSR-E observations for land surface temperature estimation: 1. Methodology and evaluation with station temperature, J. Geophys. Res-Atmos., 122, 3330-3347, https://doi.org/10.1002/2016JD026144, 2017.

Mao, K., Shi, J., Li, Z. L., Tang, H.: An RM-NN algorithm for retrieving land surface temperature and emissivity from EOS/MODIS data, J Geophys. Res-Atmos., 112, 1-17, https://doi.org/10.1029/2007JD008428, 2009.

Martins, J. P., Coelho e Freitas, S., Trigo, I. F., Barroso, C., Macedo, J.: Copernicus Global Land Operations-Lot I "Vegetation and Energy" Algorithm Theoretical Basis Document. Land Surface Temperature—LST, 1, 2019.

Reiners, P., Asam, S., Frey, C., Holzwarth, S., Bachmann, M., Sobrino, J., Göttsche, F., Bendix, J. Kuenzer, C.: Validation of AVHRR Land Surface Temperature with MODIS and In Situ LST—A TIMELINE Thematic Processor, Remote Sens., 13, 3473, https://doi.org/10.3390/rs13173473, 2021.

Wan, Z., Zhang, Y., Zhang, Q., Li, Z. L.: Validation of the land-surface temperature products retrieved from Terra Moderate Resolution Imaging Spectroradiometer data, Remote Sens. Environ, 83, 163-180, https://doi.org/10.1016/S0034-4257(02)00093-7, 2002.

Wan, Z.: New refinements and validation of the collection-6 MODIS land-surface temperature/emissivity product, Remote Sens. Environ., 140, 36–45, https://doi.org/10.1016/j.rse.2013.08.027, 2014.

Wang, K., and Liang, S.: Evaluation of ASTER and MODIS land surface temperature and emissivity products using long-term surface longwave radiation observations at SURFRAD sites, Remote Sens. Environ., 113, 1556–1565, https://doi.org/10.1016/j.rse.2009.03.009, 2009.

Xing, Z., Li, Z. L., Duan, S. B., Liu, X., Zheng, X., Leng, P., Gao, M., Zhang, X., Shang, G.: Estimation of daily mean land surface temperature at global scale using pairs of daytime and nighttime MODIS instantaneous observations, ISPRS J. Photogramm., 178, 51-67, https://doi.org/10.1016/j.isprsjprs.2021.05.017, 2021.

Yu, Y., Tarpley, D., Privette, J. L., Flynn, L. E., Xu, H., Chen, M., Vinnikov, K. Y., Sun, D., and Tian, Y.: Validation of GOES-R satellite land surface temperature algorithm using SURFRAD ground measurements and statistical estimates of error properties, IEEE Trans. Geosci. Remote Sens., 50, 704-713, https://doi.org/10.1109/TGRS.2011.2162338, 2012.

We appreciate a lot for your efforts in providing detailed comments and recommendation. They are very helpful to improve the quality of the manuscript. We have revised the manuscript according to your comments. The comments from the reviewers are kept in regular font, our responses use blue highlighting, and the revised sentences or words in the revised manuscript are highlighted with red color.

This paper developed a global historical twice-daily LST dataset (GT-LST) with a spatial resolution of 0.05° from 1981 to 2005. I believe this is an important study and it does make sense for earth science communities. The data and methods are clearly described, and the main results are well presented. However, there are some issues that need to be addressed or clarified before the paper can be published. Therefore, I recommend a major revision.

Some major comments:

1. This paper has inter-compared the GT-LST and MODIS LST over a variety of land cover types such as savannas and cropland/natural vegetation, permanent snow and ice, water bodies and etc., yet I wonder how much are the accuracies (such as RMSD and bias) over urban surfaces?

**Response**: Thanks a lot for your comments. To clearly quantify the RMSD and bias over various land cover types, we have redrawn Fig. 7, including urban and built-up lands (UBL). For your convenience, we listed it below. For UBL, the RMSD and bias are 3.4 K and 2.7 K, respectively.

[Figure]

Figure 7: RMSD and bias between GT-LST and MYD11A1 LST in 2003 for various land cover types. ENF: evergreen needleleaf forests, EBF: evergreen broadleaf forests, DNF: deciduous needleleaf forests, DBF: deciduous broadleaf forests, MXF: mixed forests, CSR: closed shrublands, OSR: open shrublands, WDS: woody savannas, SVN: savannas, GRS: grasslands, PMW: permanent wetlands, CRP: croplands, UBL: urban and built-up lands, CNV: cropland/natural vegetation mosaics, PSI: permanent snow and ice, BRN: barren, WTB: water bodies, and ALL: all land cover types.

2. Why did you choose January 15 and July 15, 1997 for the GT-LST and RT-LST comparison over continental Africa? Please clarify the selection criteria.

**Response**: Thanks a lot for your comments. The RT-LST is a twice-daily LST product at 8-km resolution over continental Africa, which spans of 6 years from 1995 to 2000. Two days, January 15 and July 15, 1997, were chosen because they represent the median time of different seasons (winter and summer, respectively). In addition, the number of matchups is enough to guarantee the reliability of the intercomparison. We have added some descriptions in Line 334-335 as follows:

"Two days, January 15 and July 15, 1997, were selected to implement the comparison over continental Africa because they represent the median time of different seasons (winter and summer, respectively)"

3. I just suggest combining Figs. 3 to 7 into a single figure for clarity.

**Response**: Thank you for the suggestion. We have redrawn the schematic of the workflow according to your suggestion. For your convenience, we listed it below.

[Figure]

Figure 3: Schematic of the workflow used to generate the GT-LST product

4. To what extent the differences in the emissivity between MODIS LSTs and GT-LST will influence their inter-comparison results, can you provide some quantitative results?

**Response**: Thanks a lot for your comments. From Fig.R1, one can conclude that a negative relationship between the GT-LST and MYD11A1 LST difference and their corresponding emissivity difference. The mean biases (GT-LST – MYD11A1) for LSTs calculated with emissivity differences less than -0.05, between -0.05 and -0.03, between -0.03 and -0.01, between -0.01 and 0.01 and more than 0.01 are 7.0, 4.3, 2.3, 0.8 and 0.7 K, respectively. We have added some descriptions in Line 406-408 as follows:

"*As a result, the dynamic emissivity of GT-LST is typically lower than that of MYD11A1, which leads to overestimation of the LST (Hulley et al., 2016; Guillevic et al., 2014;*

*Reiners et al., 2021; Ren et al., 2011). Fig. A3 shows that the mean biases (GT-LST –*
*MYD11A1) for LSTs calculated with emissivity differences less than -0.05, between -*
*0.05 and -0.03, between -0.03 and -0.01, between -0.01 and 0.01 and more than 0.01*
*are 7.0, 4.3, 2.3, 0.8 and 0.7 K, respectively.*"

[Figure]

Figure A3: Difference between GT-LST and MYD11A1 LST stratified by the difference
between GT-LST and MYD11A1 emissivity (water vapor content < 5 g/cm$^2$; satellite
zenith angle < 50°)

5. As you stated, the LSTs for a long period such as > 40 years are important for
monitoring and evaluating global long-term climate change. Thus, the validation of
tendency consistency for the generated GT-LST products is also of vital importance in
addition to its spatiotemporal pattern. Could you test the accuracy of time series GT-
LSTs over several typical regions, as I guess the orbit drift of AVHRR could also
introduce uncertainty for the tendency estimation.

**Response**: Thanks a lot for your comments. Indeed, global long-term climate change
requires daily, monthly or annual mean LST (i.e., DMLST, MMLST, and AMLST)
more than instantaneous LST as these mean LSTs are key indicators when monitoring
global LSTs over a long time series (Li et al., 2023; Liu et al., 2023; Xing et al., 2021).
Impacting of the NOAA satellite orbital drift, daytime and nighttime observations of
NOAA afternoon satellites cannot represent maximum and minimum temperatures well.
Therefore, calculating the daily and monthly mean LST by averaging daytime and
nighttime LSTs derived from GT-LST has a significantly lower accuracy than other

studies (Fig. A4). Inspired by the work of Xing et al. (2021), we use simple linear combinations of daytime and nighttime LST values that were observed at observation times for NOAA to estimate DMLST and MMLST. In order to validate the accuracy of DMLST and MMLST according to the simple linear regression method, we compared DMLST and MMLST derived from GT-LST with that of in situ LST observations from SURFRAD sites, and reported RMSE values of approximately 2.4 K and 2.7 K, respectively. These results are similar to that of Xing et al. (2021) and Chen et al. (2017). In this way, we still obtain accurate DMLST and MMLST without satellite orbit drift correction. In order to demonstrate the tendency consistency of GT-LST products, Fig. R1 shows time series of MMLST using the simple linear regression method from 1981 to 2005 for two small area (5°×5°) in the Sahara Desert and the Tibetan plateau: no significant inconsistencies can be seen.

[Figure]

Figure R1: Monthly mean LST time series for 1981 to 2005 over two small area (5°×5°) in the Sahara Desert (a) and the Tibetan plateau (b).

Then, we rephrase the paragraph in Line 429-436 as follows:

"…To estimate MMLST, first obtain the mean instantaneous clear-sky LST at daytime and nighttime, and then use these mean values to estimate MMLST according to the simple linear regression method (see Appendix B). In order to validate the accuracy of MMLST results, we compared MMLST based on GT-LST with that of in situ LST observations from SURFRAD sites for 1994–2005. All in situ LST measurements are all-sky and complete on a certain month, which means that the in situ MMLST is true MMLST. Fig. 15 showed that MMLST derived from GT-LST are related to the true MMLST, with an $R^2$ value of 0.94 and an RMSE value of 2.7 K. This result is similar to that of Chen et al. (2017), who compared MMLST from MODIS day and night

*instantaneous clear-sky LST with actual MMLST from 156 flux tower stations, and reported RMSE bias values of approximately 2.7 K.*"

We have redrawn Fig. 13 according to the simple linear regression method. For your convenience, we listed it below.

[Figure]

*Figure 13: Monthly mean LST based on GT-LST versus monthly mean LST based on in situ LST from 1994 to 2005.*

In addition, as for some details of the simple linear regression method, we have added the following descriptions in Appendix B.

"*Impacting of the NOAA satellite orbital drift, daytime and nighttime observations of NOAA afternoon satellites cannot represent maximum and minimum temperatures well. Therefore, the MMLST according to the simple average method has a significantly lower accuracy than other studies (Fig. A4). Xing et al. (2021) proposed to use 9 combinations of two to four MODIS instantaneous retrievals of which at least one*

*daytime LST and one nighttime LST to estimate mean LSTs, and determined the weight for every moment. Inspired by the work of Xing et al. (2021), we determined to use simple linear combinations of monthly mean daytime and nighttime LST values that were observed at observation times for NOAA to estimate MMLST with ground-based measurement. For the combinations of two valid monthly mean LSTs (one daytime and one nighttime LST), the regression models can be written as follows:*

$$MMLST = a_1 * MMLST_{day} + a_2 * MMLST_{night} + b \qquad \text{(B1)}$$

*where MMLST is the ground-based monthly mean LST, $a_1$, $a_2$ and $b$ are the fitting coefficients, $MMLST_{day}$ is the monthly mean in situ LST at the NOAA daytime observation, $MMLST_{night}$ is the monthly mean in situ LST at the NOAA nighttime observation.*

*Taking into account the observed times of NOAA satellites with orbital drift effect since 1981, combinations of two observations from these satellites contain eight cases: 13:30–17:00/01:30–05:00 local solar time in 0.5-hour interval. Based on the in situ LST measurements during the period 2003 to 2018 at 227 flux stations operating in globally diverse regions, we obtained the fitting coefficients (Table A1). Then, we calculated the MMLST of GT-LST using GT-LST monthly mean daytime and nighttime LSTs, Eq. (B1), and the fitting coefficients listed in Table A1.*"

Table A1. Statistics for the relationship between the regressions of the eight combinations and actual monthly mean LST.

| Case | Time | $a_1$ | $a_2$ | b | RMSE | $R^2$ | Number |
|------|------|-------|-------|---|------|-------|--------|
| 1 | 13:30/01:30 | 0.3844 | 0.5783 | 10.3446 | 2.0 | 0.97 | 12095 |
| 2 | 14:00/02:00 | 0.4010 | 0.5621 | 10.2042 | 1.9 | 0.98 | 12241 |
| 3 | 14:30/02:30 | 0.4235 | 0.5451 | 8.6172 | 1.9 | 0.98 | 12381 |
| 4 | 15:00/03:00 | 0.4490 | 0.5211 | 8.2652 | 1.8 | 0.98 | 12303 |
| 5 | 15:30/03:30 | 0.4816 | 0.4840 | 9.5710 | 1.8 | 0.98 | 12165 |
| 6 | 16:00/04:00 | 0.5250 | 0.4349 | 11.2284 | 2.0 | 0.97 | 11818 |
| 7 | 16:30/04:30 | 0.5663 | 0.3884 | 12.8572 | 2.2 | 0.96 | 10992 |
| 8 | 17:00/05:00 | 0.6040 | 0.3621 | 9.7302 | 2.4 | 0.96 | 9765 |

Some minor Comments:

1. Line 125: Is there a writing mistake on this sentence? "we used 54 land surface emissivity spectra to represent different land surface types, including 41 soil types, four vegetation types, four water body 125 types and five ice/snow types were selected."

**Response**: We appreciate your careful reading, the phrase "were selected" was removed.

2. Line 165: "The instrumental error of the SURFRAD station give rise to uncertainty in the retrieved LST value of less than 1 K". Should be "gives rise to".

**Response**: Corrected as suggested.

3. Line 266 to 268: "Therefore, to obtain relatively accurate emissivity values, we developed an improved method that consider annual changes in land cover from the GLASS-GLC dataset and combines ASTER GED data with the NDVI threshold method to estimate the emissivity" The verb forms need to be unified.

**Response**: Thank you for your careful reading. Following your suggestion, we have checked the whole manuscript and corrected this issue.

4. Line 313 to 315: This sentence seems redundant, please write it in a more explicit way.

**Response**: Thank you for the suggestion. We have rewritten this part according to your suggestion.:

"In contrast to the ground-based validation and satellite product inter-comparison mentioned above, the comparisons for AVHRR LST products were performed using different strategies. Concretely, GT-LST compared with GD-LST using a strategy that compares GT-LST and GD-LST with same SURFRAD measurements concurrently with the satellite overpass, to evaluate the difference in the absolute accuracy of these two products."

5. You can use either RMSE or RMSD, but keep consistency throughout the paper and all figures.

**Response**: Corrected as suggested.

6. Please unify the format of all references.

**Response**: Thank you for the suggestion. Following your suggestion, we have checked the whole manuscript and corrected this issue.

**References for the above responses are listed below**:

Chen, X., Su, Z., Ma, Y., Cleverly, J., Liddell, M.: An accurate estimate of monthly mean land surface temperatures from MODIS clear-sky retrievals, J. Hydrometeorol., 18, 2827-2847, https://doi.org/10.1175/JHM-D-17-0009.1,2017.

Li, Z.-L., Wu, H., Duan, S.-B., Zhao, W., Ren, H., Liu, X., Leng, P., Tang R., Ye, X., Zhu, J., Sun, Y., Si, M., Liu, M., Li, J., Zhang, X., Shang, G., Tang, B.-H., Yan, G., and Zhou, C.: Satellite remote sensing of global land surface temperature: Definition, methods, products, and applications, Rev. Geophys., 61, e2022RG000777, https://doi.org/10.1029/2022RG000777, 2023.

Liu, X., Li, Z.-L., Li, J.-H., Leng, P., Liu, M., and Gao. M.: Temporal upscaling of MODIS 1-km instantaneous land surface temperature to monthly mean value: Method evaluation and product generation, IEEE Trans. Geosci. Remote Sens., https://doi.org/10.1109/TGRS.2023.3247428, 2023.

Xing, Z., Li, Z. L., Duan, S. B., Liu, X., Zheng, X., Leng, P., Gao, M., Zhang, X., Shang, G.: Estimation of daily mean land surface temperature at global scale using pairs of daytime and nighttime MODIS instantaneous observations, ISPRS J. Photogramm., 178, 51-67, https://doi.org/10.1016/j.isprsjprs.2021.05.017, 2021.

---

## Author Response (AR3)

**Response to Reviewer #1**

The authors have addressed my concerns in the previous discussions, and I only have several comments on technical corrections.

Q1. Line 70: space missed after 0.5°

**Response**: Thank you for your careful reading. We have corrected this issue.

Q2. Based on Fig 12, the MMLST (monthly mean LST) still has clear gaps, so the statement 'all-sky' or 'gap-free' (line 28) is not completely accurate, even though the improvement for minimizing the cloud gaps is quite significant.

**Response**: We appreciate your careful reading, the phrase "under all-sky conditions" in Line 28 has been removed.

Q3. Obtaining monthly mean could be a considerable contribution to the community while as the authors mentioned in the manuscript, the data are still limited by the orbital drift issue. I would recommend the authors provide some additional statistics to verify how much the RMSE of MMLST would change due to orbital drift after several years, probably temporal accuracy variation analysis for N14 would be enough.

This would help to assess the impact of the orbital drift issue on the accuracy of the MMLST product over time, providing motivation for future studies. By analyzing how the RMSE changes over time, the authors could provide a better understanding of the limitations and potential biases of the data, which would be useful for the community using this product.

**Response**: Thanks for your suggestion. Many studies have shown that two satellite observations that are separated by approximately 12 h can be used to estimate a relatively accurate daily and monthly mean LST (i.e., DMLST and MMLST) by weighted average method (Chen et al., 2017; Liu et al., 2023; Xing et al., 2021). Based on the in situ LST measurements, the weight for every moment in every combination (one daytime and one nighttime) is determined. Fig. R1 displays the density scatter plots that depict the true in situ DMLST against the weighted average of the in situ

day/night observations (one daytime and one nighttime) at the time of NOAA satellite overpass. The results show that the DMLST obtained by the proposed method is not influenced by the orbital drift of the satellite. We applied the same methodology to compute the MMLST, as described in Appendix B. Theoretically, the MMLST is also independent of the satellite orbit drift, resulting in a constant RMSE over time.

[Figure]

Figure R1. Comparison of the actual in situ DMLST and the in situ DMLST estimated using a weighted average method based on NOAA satellites with orbital drift effect at 76 stations from 2003 to 2018: (a) 13:30/01:30, (b) 14:00/02:00, (c) 14:30/02:30, (d)

15:00/3:00, (e) 15:30/03:30, (f) 16:00/04:00, (g) 16:30/04:30, and (h) 17:00/05:00 (Li et al., under review).

**References for the above responses are listed below**:

Chen, X., Su, Z., Ma, Y., Cleverly, J., and Liddell, M.: An accurate estimate of monthly mean land surface temperatures from MODIS clear-sky retrievals, J. Hydrometeorol., 18, 2827-2847, https://doi.org/10.1175/JHM-D-17-0009.1, 2017.

Li, J.-H., Li, Z.-L., Liu, X., Duan, S.-B., Si, M., Shang, G., and Zhang, X.: A generalized method for retrieving global daily mean land surface temperature from polar-orbiting thermal infrared sensor instantaneous observations, Int. J. Remote Sens., Under review.

Liu, X., Li, Z.-L., Li, J.-H., Leng, P., Liu, M., and Gao. M.: Temporal upscaling of MODIS 1-km instantaneous land surface temperature to monthly mean value: Method evaluation and product generation, IEEE Trans. Geosci. Remote Sens., https://doi.org/10.1109/TGRS.2023.3247428, 2023.

Xing, Z., Li, Z.-L., Duan, S.-B., Liu, X., Zheng, X., Leng, P., Gao, M., Zhang, X., and Shang, G.: Estimation of daily mean land surface temperature at global scale using pairs of daytime and nighttime MODIS instantaneous observations, ISPRS J. Photogramm., 178, 51–67, https://doi.org/10.1016/j.isprsjprs.2021.05.017, 2021.

**Response to Reviewer #2**

Overall, I think the authors have done a good job with the revision.

With a journal such as ESSD however, I am concerned about the dataset being artificially restricted to the year 2005, which will limit its use for examining long-term LST trends, the primary goal of releasing the dataset. The authors have mentioned in the response letter that they are processing the data till the year 2021. It may have been helpful to just include those years of analysis to this paper for a more complete dataset paper.

However, this is not a fundamental flaw with the science behind the study. As such, I don't have a strong opinion and will leave the decision up to the editor.

**Response**: Thanks for your valuable comments. We have followed your suggestion and generated the GT-LST product for the period 2006-2021, which is now publicly available at https://doi.org/10.5281/zenodo.7813607 (Li et al., 2023).

**References for the above responses are listed below**:

Li, J. H., Liu, X., Li, Z. L., and Duan, S. B.: A global historical twice-daily (daytime and nighttime) land surface temperature dataset produced by AVHRR observations from 1981 to 2021 (2006-2021), https://doi.org/10.5281/zenodo.7813607, 2023.

**Response to Editor**

Overall all the four reviewers are satisfied with the authors' revision. However, I agree with the Reviewer #4's comments on the covered perids of the dataset. I strongly recommend the authors to include the recent 20 years' data in this paper, considering that the authors are already processing other years' data. Extending the time series to cover the nearly 40 years will make this dataset more useful and valuable. Restricting the data to the year 2005 which limit its use and is not reasonable, considering that the same methods can be applied to other years.

**Response**: Thank you for your valuable suggestion. According to your suggestion, we have extended the time span of the GT-LST product to 2021. The GT-LST product for the period 2006-2021 can be freely available at https://doi.org/10.5281/zenodo.7813607 (Li et al., 2023). Our revised article is titled "A global historical twice-daily (daytime and nighttime) land surface temperature dataset produced by AVHRR observations from 1981 to 2021", which more accurately reflects the scope of our study. Furthermore, we have updated several figures in our manuscript, including Fig. 1, Fig. 2, Fig. 3, and Fig. 15.

**References for the above responses are listed below**:

Li, J. H., Liu, X., Li, Z. L., and Duan, S. B.: A global historical twice-daily (daytime and nighttime) land surface temperature dataset produced by AVHRR observations from 1981 to 2021 (2006-2021), https://doi.org/10.5281/zenodo.7813607, 2023.